# On Federated Learning of Deep Networks from Non-IID Data: Parameter Divergence and the Effects of Hyperparametric Methods

## Abstract

*Federated learning*, where a global model is trained by iterative *parameter averaging* of locally-computed updates, is a promising approach for distributed training of deep networks; it provides high communication-efficiency and privacy-preservability, which allows to fit well into decentralized data environments, e.g., mobile-cloud ecosystems. However, despite the advantages, the federated learning-based methods still have a challenge in dealing with non-IID training data of local devices (i.e., learners). In this regard, we study the effects of a variety of hyperparametric conditions under the non-IID environments, to answer important concerns in practical implementations: (i) We first investigate *parameter divergence* of local updates to explain performance degradation from non-IID data. The origin of the parameter divergence is also found both empirically and theoretically. (ii) We then revisit the effects of optimizers, network depth/width, and regularization techniques; our observations show that the well-known advantages of the hyperparameter optimization strategies could rather yield diminishing returns with non-IID data. (iii) We finally provide the reasons of the failure cases in a categorized way, mainly based on metrics of the parameter divergence.

## 1 Introduction

Over the recent years, *federated learning* (McMahan et al., 2017) has been a huge success to reduce the communication overhead in distributed training of deep networks. Guaranteeing competitive performance, the federated learning permits each learner to compute their *local updates* of each round for relatively many iterations (e.g., 1 epoch, 10 epochs, etc.), which provides much higher communication-efficiency compared to the conventional data parallelism approaches (for intra-datacenter environments, e.g., Dean et al. (2012); Chen et al. (2016)) that generally require very frequent *gradient* aggregation. Furthermore, the federated learning can also significantly reduce data privacy and security risks by enabling to conceal on-device data of each learner from the server or other learners; thus the approach can be applied well to environments with highly private data (e.g., personal medical data), it is now emerging as a promising methodology for privacy-preserving distributed learning along with differential privacy-based methods (Hard et al., 2018; Yang et al., 2018; Bonawitz et al., 2019; Chen et al., 2019).

On this wise, the federated learning takes a simple approach that performs iterative *parameter averaging* of local updates computed from each learners' own dataset, which suggests an efficient way to learn a shared model without centralizing training data from multiple sources; but hereby, since the local data of each device is created based on their usage pattern, the heterogeneity of training data distributions across the learners might be naturally assumed in real-world cases. Hence, each local dataset would not follow the population distribution, and handling the decentralized non-IID data still remains a statistical challenge in the field of federated learning (Smith et al., 2017). For instance, Zhao et al. (2018) observed severe performance degradation in multi-class classification accuracy under highly skewed non-IID data; it was reported that more diminishing returns could be yielded as the probabilistic distance of learners' local data from the population distribution increases.

---

**Algorithm 1** Federated learning. $\mathbf{w}_k^t$ is the model parameters updated by learner $k$ at communication round $t$, $\overline{\mathbf{w}}^t$ is the global model parameters at round $t$, $\mathcal{K} = \{1, 2, \cdots, K\}$ is the universal set of learners, $\mathcal{P}_k$ is the local training dataset of learner $k$, $B$ is the local minibatch size, $E$ is the number of the local epochs per round, $\eta$ is the learning rate, and $\ell(\cdot)$ is the loss function.

---

**Server executes:**
    initialize $\overline{\mathbf{w}}^0$
    **for** each round $t = 1, 2, \cdots, T$ **do**
        **for** each learner $k \in \mathcal{K}$ **in parallel do**
            $\mathbf{w}_k^t \leftarrow \text{LearnerUpdate}(k, \overline{\mathbf{w}}^{t-1})$
        **end for**
        $\overline{\mathbf{w}}^t \leftarrow \sum_{k \in \mathcal{K}} \frac{|\mathcal{P}_k|}{\sum_{j \in \mathcal{K}} |\mathcal{P}_j|} \mathbf{w}_k^t$
    **end for**

**LearnerUpdate**$(k, \mathbf{w})$: // *Run on learner $k$*
    $\mathcal{B} \leftarrow$ (split $\mathcal{P}_k$ into batches of size $B$)
    **for** each local epoch $\varepsilon$ from 1 to $E$ **do**
        **for** each batch $\mathbf{b} \in \mathcal{B}$ **do**
            $\mathbf{w} \leftarrow \mathbf{w} - \eta \nabla \ell(\mathbf{w}; \mathbf{b})$
        **end for**
    **end for**
    return $\mathbf{w}$ to server

---

**Contributions.** To address the non-IID issue under federated learning, there have been a variety of recent works[1]; nevertheless, in this paper we explore more fundamental factors, the effects of various hyperparameters. The optimization for the number of local iterations per round or learning rates has been handled in several literatures (e.g., Huang et al. (2018); Li et al. (2019c); Wang et al. (2019)); by extension we discuss, for the first time to the best of our knowledge, the effects of optimizers, network depth/width, and regularization techniques.

Our contributions are summarized as follows: First, as a root cause of performance degradation from non-IID data, we investigate *parameter divergence* of local updates at each round. The parameter divergence can be regarded as a direct response to learners' local data being non-IID sampled from the population distribution, of which the excessive magnitude could disturb the performance of the consequent parameter averaging. We also investigate the origin of the parameter divergence in both empirical and theoretical ways. Second, we observe the effects of well-known hyperparameter optimization methods[2] under the non-IID data environments; interestingly, some of our findings show highly conflicted aspects with their positive outcomes under "vanilla" training[3] or the IID data setting. Third, we analyze the internal reasons of our observations in a unified way, mainly using the parameter divergence metrics; it is identified that the rationale of the failures under non-IID data lies in some or all of (i) inordinate magnitude of parameter divergence, (ii) its *steep fall phenomenon* (described in Section 4.2), and (iii) excessively high training loss of local updates.

## 2 PRELIMINARIES

### 2.1 ALGORITHM

In this study, Algorithm 1 is considered as a federated learning method, and it is written based on `FedAvg` (McMahan et al., 2017).[4] We note that this kind of parameter averaging-based approach has been widely discussed in the literature, under various names, e.g., *parallel (restarted) SGD* (Zhang et al., 2016; Yu et al., 2019) and *local SGD* (Lin et al., 2018; Stich, 2019).

### 2.2 EXPERIMENTAL SETUP

In our experiments with Tensorflow (Abadi et al., 2016),[5] we consider the multi-class classification tasks on CIFAR-10 (Krizhevsky & Hinton, 2009) and SVHN (Netzer et al., 2011) datasets.

---

[1]For instance, Smith et al. (2017); Caldas et al. (2018); Huang et al. (2018); Jeong et al. (2018); Zhao et al. (2018); Zhu & Jin (2018); Corinzia & Buhmann (2019); Duan (2019); Li et al. (2019a;b;c); Liu et al. (2019); Mohri et al. (2019); Sattler et al. (2019); Wang et al. (2019); Yonetani et al. (2018); Yoshida et al. (2019).

[2]we use the term *hyperparameter optimization methods* and *hyperparametric methods* interchangeably.

[3]This term refers to the non-distributed training with a single machine, using the whole training examples.

[4]Regarding the significance of the algorithm, we additionally note that Google is currently employing it on their mobile keyboard application (Gboard) (Hard et al., 2018; Yang et al., 2018; Bonawitz et al., 2019; Chen et al., 2019). In this study we deal with image classification, which is also considered as the main applications of federated learning along with the language models (McMahan et al., 2017).

[5]Our source code is available at `https://github.com/fl-noniid/fl-noniid`

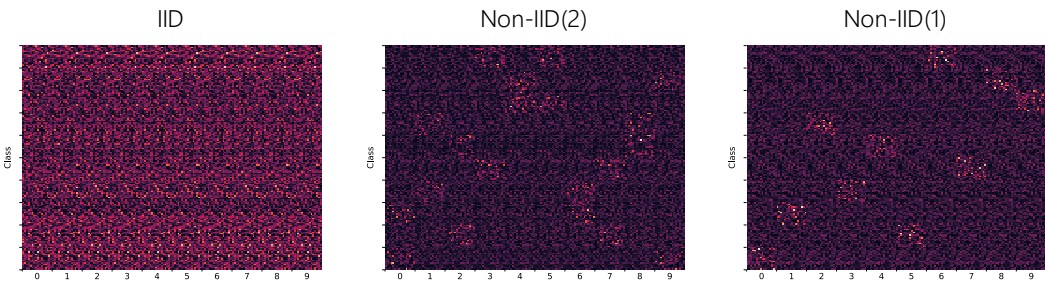

Figure 1: Magnitude of neurons in the last fully-connected layer of the NetA-Baseline model, at the first round of Algorithm 1 on CIFAR-10. Brighter ones illustrates having greater magnitude.

**Baseline network model:** For the baseline deep network, we consider a CNN model that has three $3 \times 3$ convolutional layers with 64, 128, 256 output channels, respectively; and then three fully-connected layers with 512, 256, 10 output sizes, respectively (see Appendix A.1 for more detailed description). We use the term NetA-Baseline to denote this baseline model throughout this paper. **Regularization configuration:** For weight decay, we apply the method of decoupled weight decay regularization (Loshchilov & Hutter, 2019) based on the fact that weight decay is equivalent to $L_2$ regularization *only* for pure SGD (Loshchilov & Hutter, 2019; Zhang et al., 2019). The baseline value of the weight decay factor is set to 0.00005. As our regularization baseline, we consider not to apply any other regularization techniques additionally. We importantly note that if without any particular comments, the results described in the following sections are ones obtained using the above baseline configurations of the network model and regularization.

**Environmental configuration:** We consider 10 learners to have each 5000 non-overlapping training examples; Table 1 summarizes our configuration of data settings; Non-IID(N) denotes a data setting that lets each learner to have training examples only for N class(es). The data settings in the Table 1 deal with data balanced cases where learners have the same amount of local data, and they are mainly considered in the follow-

Table 1: Configuration of balanced data settings. #Cls/L: the number of classes in each learner's local training dataset; #Exs/Cl/L: the number of training examples per class in each local dataset.

| Data Setting | #Learners | #Cls/L | #Exs/Cl/L |
|---|---|---|---|
| IID | 10 | 10 | 500 |
| Non-IID(N) | 10 | N | 5000/N |

ing sections; we additionally note that one can refer to Appendix C.8 for the experiments with data unbalanced cases. For the IID and the non-IID data settings, $T = 200$ and 300 are used respectively, while $E = 1$ and minibatch size of 50 are considered commonly for the both.[6] One can find the remaining configurations for the experiments in Appendix A.

## 3 PARAMETER DIVERGENCE

*Parameter divergence* is recently being regarded as a strong cause of diminishing returns from decentralized non-IID data in federated learning (Zhao et al., 2018) (it is sometimes expressed in another way, *gradient/loss divergence* (Li et al., 2019b;c; Liu et al., 2019; Wang et al., 2019)). For the divergence metrics, many of the literatures usually handle the difference of each learner's local model parameters from one computed with the population distribution; it eventually also causes parameter diversity *between the local updates* as the data distributions become heterogeneous across learners. A pleasant level of parameter divergence could rather imply exploiting rich decentralized data (IID cases); however, if the local datasets are far from the population distribution, the consequent parameter averaging of the highly diverged local updates could lead to bad solutions away from the global optimum (non-IID cases).

---

[6]Here, *one epoch* is used as one of the standard local iterations per round in McMahan et al. (2017) and its large number of descendants; furthermore, more extreme cases such as $E = 5, 20$ are also considered. In addition, in view of the number of local iterations, we use 100 for it during each round; note that $\gg 100$ iterations are even considered (e.g., up to 200 iterations is used on CIFAR-10 in McMahan et al. (2017)).

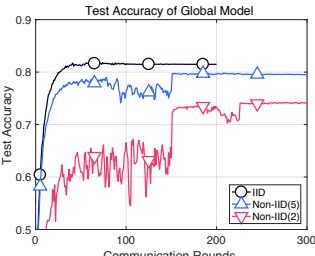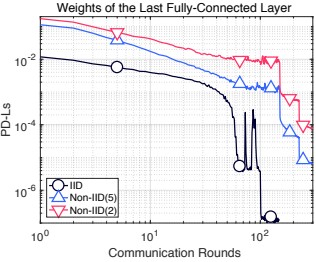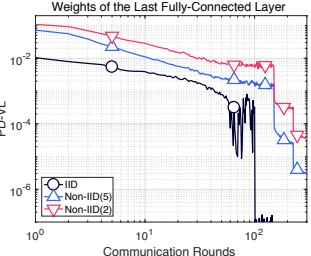

Figure 2: Comparison of test accuracy and parameter divergence with respect to the degree of data non-IIDness in the training of NetA-Baseline on CIFAR-10. Note that EMD values of Non-IID(5) and Non-IID(2) from the population distribution are 1.0 and 1.6, respectively.

**The origin of parameter divergence.** In relation, it has been theoretically proven that the parameter divergence (between the global model parameters under FedAvg and those computed by vanilla SGD training) is directly related to the probabilistic distance of local datasets from the population distribution (see Proposition 3.1 in Zhao et al. (2018)). In addition to it, for multi-class classification tasks, we here identify in lower level, that *if data distributions in each local dataset are highly skewed and heterogeneous over classes, subsets of neurons, which have especially big magnitudes of the gradients in back propagation, become significantly different across learners*; this leads to inordinate parameter divergence between them. As illustrated in Figure 1, under the IID data setting, the weight values in the output layer are evenly distributed relatively evenly across classes if the neurons of the model are initialized uniformly. However, we can observe under the non-IID data settings that the magnitudes of the gradients are distributed depending on each learner's data distribution. We also provide the corresponding theoretical analysis in Appendix B.

**Metrics.** To capture parameter divergence under federated learning, we define the following two metrics using the notations in Algorithm 1. Since in our analysis we compare different network architectures or training settings together in a set, the number of neurons in the probed layers can become different, and values of model parameters can highly depend on the experimental manipulations; thus instead of Euclidean distance, in the two divergence metrics we use cosine distance that enables normalized (qualitative) measures. We also note that PD-VL is defined assuming the balancedness of data amount between learners, i.e., the same numbers of local iterations per round.

The reason of probing parameter divergence being important is that the federated learning are performed based on iterative parameter averaging. That is, investigating how local updates are diverged can give a clue whether the subsequent parameter averaging yields positive returns; the proposed divergence metrics provide two ways for it.

**Definition 1.** *For* $\mathbf{z}_1^t, \mathbf{z}_2^t, \cdots, \mathbf{z}_K^t \in \mathbb{R}^d$ *where* $\mathbf{z}_k^t$ *is a subset (or the universal set) of* $\mathbf{w}_k^t$, *we define parameter divergence between local updates as*

$$\text{PD-Ls}: \zeta\left(\mathbf{z}_1^t, \mathbf{z}_2^t, \cdots, \mathbf{z}_K^t\right) = \binom{K}{2}^{-1} \sum_{i,j \in \mathcal{K}; i < j} \left(1 - \frac{\mathbf{z}_i^t \cdot \mathbf{z}_j^t}{\left\|\mathbf{z}_i^t\right\| \left\|\mathbf{z}_j^t\right\|}\right).$$

*In addition, assume that* $\left|\mathcal{P}_k\right|$ *is identical* $\forall k \in \mathcal{K}$, *and let* $\mathbf{w}_{-1}^t$ *be the vanilla-updated parameters, that is, the model parameters updated on the global parameters (i.e.,* $\overline{\mathbf{w}}^{t-1}$*) using IID training data during the same number of iterations with the actual learners (i.e.,* $\left|\mathcal{P}_k\right|/B$*). Then, for* $\mathbf{z}_1^t, \mathbf{z}_2^t, \cdots, \mathbf{z}_K^t$, *and* $\mathbf{z}_{-1}^t \in \mathbb{R}^d$ *where* $\mathbf{z}_{-1}^t$ *is a subset (or the universal set) of* $\mathbf{w}_{-1}^t$, *we define parameter divergence between the vanilla update and local updates as*

$$\text{PD-VL}: \xi\left(\mathbf{z}_{-1}^t; \mathbf{z}_1^t, \mathbf{z}_2^t, \cdots, \mathbf{z}_K^t\right) = \frac{1}{K} \sum_{k \in \mathcal{K}} \left(1 - \frac{\mathbf{z}_{-1}^t \cdot \mathbf{z}_k^t}{\left\|\mathbf{z}_{-1}^t\right\| \left\|\mathbf{z}_k^t\right\|}\right).$$

**Relationship among probabilistic distance, parameter divergence, and learning performance.** We consider Non-IID(5) and Non-IID(2) for non-IID data settings. Here we use earth mover's distance (EMD), also known as Wasserstein distance, to measure probabilistic distance of each data

settings from the population distribution; the value becomes 1.0 and 1.6 for Non-IID(5) and Non-IID(2), respectively. From the middle and right panels of Figure 2, it is seen that greater EMDs lead to bigger parameter divergence (refer to also Figure 9 in the appendix). Also, together with the left panel, we can observe the positive correlation between parameter divergence and learning performance. Therefore, we believe the parameter divergence metrics can help to reveal the missing link between data non-IIDness and the consequent learning performance. Note that one can also refer to the similar analysis with more various EMD in Zhao et al. (2018).

## 4 THE EFFECTS OF HYPERPARAMETRIC METHODS

### 4.1 SUMMARY OF OUR OBSERVATIONS

From now on we describe our findings for the effects of various hyperparameter optimization methods with non-IID data on the federated learning algorithm. The considered hyperparametric methods have been a huge success to improve performance in deep learning; however, here we newly identify that under non-IID data settings, they could give negative/diminishing effects on performance of the federated learning algorithm. The following is the summary of our findings; we provide the complete experimental results and further discussion in the next subsection and the appendix.

**Effects of optimizers.** Unlike non-adaptive optimizers such as pure SGD and momentum SGD (Polyak, 1964; Nesterov, 1983), Adam (Kingma & Ba, 2015) could give poor performance from non-IID data if the parameter averaging is performed only for weights and biases, compared to all the model variables (including the first and second moment) being averaged.

Here we importantly note that both momentum SGD and Adam require the additional variables related to momentum as well as weights and biases; throughout the rest of the paper, the terms *(optimizer name)-A* and *(optimizer name)-WB* are used to refer to the parameter averaging being performed for all the variables[7] and only for weights & biases, respectively.

**Effects of network depth/width.** It is also known that deepening "plain" networks (which simply stacks layers, without techniques such as *information highways* (Srivastava et al., 2015) and *shortcut connection* (He et al., 2016)) yields performance degradation at a certain depth, even under vanilla training; however this phenomenon gets much worse under non-IID data environments. On the contrary, widening networks could help to achieve better outcomes; in that sense, *the global average pooling* (Lin et al., 2014) could fail in this case since it significantly reduces the channel dimension of the (last) fully-connected layer, compared to using the max pooling.

**Effects of Batch Normalization.** The well-known strength of Batch Normalization (Ioffe & Szegedy, 2015), the dependence of hidden activations in the minibatch (Ioffe, 2017), could become a severe drawback in non-IID data environments. Batch Renormalization (Ioffe, 2017) helps to mitigate this, but it also does not resolve the problem completely.

**Effects of regularization techniques.** With non-IID data, regularizations techniques such as weight decay and data augmentation could give excessively high training loss of local updates even in a modest level, which offsets the generalization gain.

### 4.2 DISCUSSION

We now explain the internal reasons of the observations in the previous subsection. Through the experimental results, we were able to classify the causes of the failures under non-IID data into

---

[7]To the best of our knowledge, so far there have been no studies about Adam to synchronize all the three sets of variables (i.e., weights & biases, the first moment, and the second moment) under federated learning. However, in the momentum SGD case, there have been some literatures; for instance, Lin et al. (2018) presented the synchronization methods with *local momentum*, *global momentum*, and *hybrid momentum*. Here our simple averaging strategy has the similar philosophy with the local momentum method; one can see from Table 4 in Lin et al. (2018) that the simple averaging strategy can yield still competitive results compared to global momentum or hybrid momentum method.

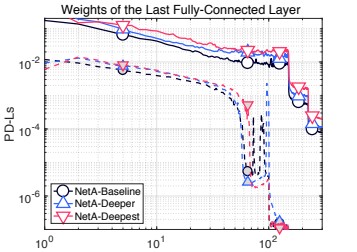 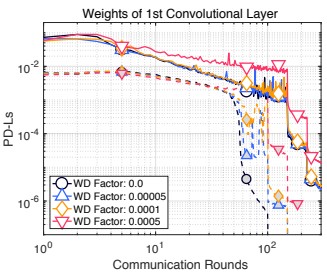 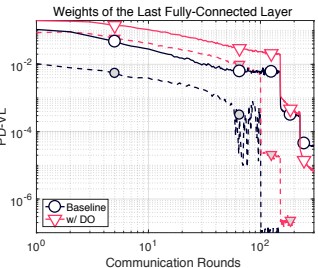

Figure 3: Inordinate magnitude of parameter divergence in the CIFAR-10 training with NetA-Deepest (left), weight decay factor of 0.0005 (middle), and Dropout (right), respectively, under the non-IID data setting. Dotted and solid lines indicate the results under IID and Non-IID(2) setting, respectively. WD: weight decay; DO: dropout.

Table 2: Test accuracy (%) of the trials in Figure 3. Values outside/inside brackets denote one measured after the whole training and the highest one during the rounds, respectively. NMom: Nesterov momentum SGD; BS: minibatch size. Note that weight decay factor of 0.00005 is the baseline configuration.

| Network | Method | Vanilla Training | | Federated Learning | |
|---|---|---|---|---|---|
| | | BS: 50 | BS: 500 | IID | Non-IID(2) |
| NetA-Baseline | NMom-A | 83.89 (83.90) | 79.39 (79.49) | 81.49 (81.82) | 74.11 (74.23) |
| NetA-Deeper | NMom-A | 85.94 (86.00) | 81.23 (81.29) | 83.40 (83.63) | 73.67 (73.89) |
| NetA-Deepest | NMom-A | 86.20 (86.33) | 79.48 (79.53) | 83.12 (83.58) | **68.98 (69.64)** |
| NetA-Baseline | NMom-A + WD: 0.0 | 81.13 (81.15) | 79.09 (79.25) | 81.68 (81.90) | 73.95 (64.27) |
| NetA-Baseline | NMom-A + WD: 0.0001 | 84.29 (84.46) | 79.13 (79.74) | 82.22 (82.25) | 72.65 (72.79) |
| NetA-Baseline | NMom-A + WD: 0.0005 | 83.66 (84.60) | 80.48 (80.79) | 82.69 (83.14) | **54.11 (54.15)** |
| NetA-Baseline | NMom-A + DO | 86.55 (86.63) | 84.29 (84.32) | 84.34 (84.53) | **75.80 (75.89)** |

three categories; the following discussions are described based on this.[8] Note that our discussion in this subsection is mostly made from the results under Nesterov momentum SGD and on CIFAR-10; the complete results including other optimizers (e.g., pure SGD, Polyak momentum SGD, and Adam) and datasets (e.g., SVHN) are given in Appendix C.

**Inordinate magnitude of parameter divergence.** As mentioned before, bigger parameter divergence is the root cause of diminishing returns under federated learning methods with non-IID data. By extension, here we observe that even under the same non-IID data setting, some of the considered hyperparametric methods yield greater parameter divergence than when they are not applied.

For example, from the left plot of Figure 3, we see that under the Non-IID(2) setting, the parameter divergence values (in the last fully-connected layer) become greater as the network depth increases (note that NetA-Baseline, NetA-Deeper, and NetA-Deepest have 3, 6, and 9 convolutional layers, respectively; see also Appendix A.1 for their detailed architecture). The corresponding final test accuracy was found to be 74.11%, 73.67%, and 68.98%, respectively, in order of the degree of shallowness; this fits well into the parameter divergence results. Since the NetA-Deeper and NetA-Deepest have twice and three times as many model parameters as NetA-Baseline, it can be expected enough that the deeper models yield bigger parameter divergence in the whole model; but our results also show its qualitative increase in a layer level. In relation, we also provide the results using the modern network architecture (e.g., ResNet (He et al., 2016)) in Table 8 of the appendix.

---

[8]From the figures of the experimental results in Appendix C, we can identify that in most cases the parameter divergence values of the first convolutional layer and the last fully-connected layer would be more dominant than those of the other layers, judging from their difference of magnitude between under the IID and the non-IID data setting (please also note that log scale is used for the y-axis). We additionally remark that the results of other related studies also show the dominance of the first convolutional layer and the last fully-connected layer (e.g., see Figure 2 in Zhao et al. (2018)). Therefore, our discussion here was primarily described based on the results of the first convolutional layer and the last fully-connected layer.

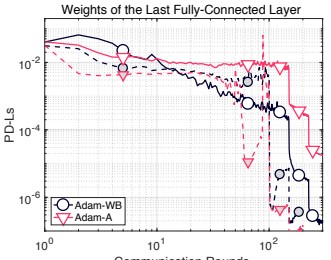 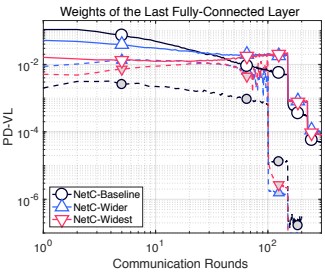 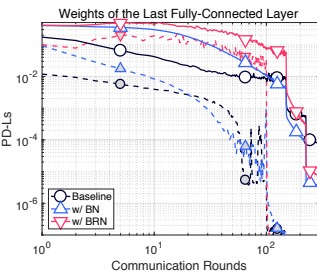

Figure 4: Steep fall phenomenon in the CIFAR-10 training with Adam-WB (left), NetC-Baseline (middle), and Batch Normalization (right), respectively, under the non-IID data setting. Dotted and solid lines indicate the results under IID and Non-IID(2) setting, respectively. BN: Batch Normalization; BRN: Batch Renormalization.

Table 3: Test accuracy (%) of the trials in Figure 4.

| Network | Method | Vanilla Training | | Federated Learning | |
|---|---|---|---|---|---|
| | | BS: 50 | BS: 500 | IID | Non-IID(2) |
| NetA-Baseline | Adam-WB | 81.97 (82.35) | 80.73 (80.76) | 80.99 (81.00) | **67.51 (67.82)** |
| NetA-Baseline | Adam-A | 81.97 (82.35) | 80.73 (80.76) | 81.18 (81.45) | 75.32 (75.45) |
| NetC-Baseline | NMom-A | 84.25 (84.50) | 76.70 (76.82) | 80.60 (80.75) | **64.06 (64.55)** |
| NetC-Wider | NMom-A | 81.48 (81.53) | 77.00 (77.14) | 79.59 (79.85) | 72.61 (72.97) |
| NetC-Widest | NMom-A | 83.16 (83.36) | 78.39 (78.56) | 80.90 (81.14) | 73.64 (73.91) |
| NetA-Baseline | NMom-A | 83.89 (83.90) | 79.39 (79.49) | 81.49 (81.82) | 74.11 (74.23) |
| NetA-Baseline | NMom-A + BN | 85.22 (85.25) | 79.58 (80.14) | 83.54 (83.80) | **50.46 (59.31)** |
| NetA-Baseline | NMom-A + BRN | 86.36 (86.43) | 82.03 (82.46) | 84.24 (84.70) | **70.32 (70.38)** |

From the middle plot of the figure, we can also observe bigger parameter divergence in a high level of weight decay under the Non-IID(2) setting. Under the non-IID data setting, the test accuracy of about $72 \sim 74\%$ was achieved in the low levels ($\leq 0.0001$), but weight decay factor of 0.0005 yielded only that of 54.11%. Hence, this suggests that with non-IID data we should apply much smaller weight decay to federated learning-based methods. Here we note that if a single iteration is considered for each learner's local update per round, the corresponding parameter divergence will be of course the same without regard to degree of weight decay. However, in our experiments, the great number of local iterations per round (i.e., 100) made a big difference of the divergence values under the non-IID data setting; this eventually yielded the accuracy gap. We additionally observe for the non-IID cases that even with weight decay factor of 0.0005, the parameter divergence values are similar to those with the smaller factors at very early rounds in which the norms of the weights are relatively very small.

In addition, it is observed from the right plot of the figure that Dropout (Hinton et al., 2012; Srivatava et al., 2014) also yields bigger parameter divergence under the non-IID data setting. The corresponding test accuracy was seen to be a diminishing return with Nesterov momentum SGD (i.e., using Dropout we can achieve $+2.85\%$ under IID, but only $+1.69\%$ is obtained under non-IID(2), compared to when it is not applied; see Table 2); however, it was observed that the generalization effect of the Dropout is still valid in test accuracy for the pure SGD and the Adam (refer to also Table 13 in the appendix).

**Steep fall phenomenon.** As we see previously, inordinate magnitude of parameter divergence is one of the notable characteristics for failure cases under federated learning with non-IID data. However, under the non-IID data setting, some of the failure cases have been observed where the test accuracy is still low but the parameter divergence values of the last fully-connected layer decrease (rapidly) over rounds; as the round goes, even the values were sometimes seen to be lower than those of the comparison targets. We refer to this phenomenon as *steep fall phenomenon*. It is inferred that these (unexpected abnormal) sudden drops of parameter divergence values indicate going into poor local minima (or saddles); this can be supported by the behaviors that test accuracy increases

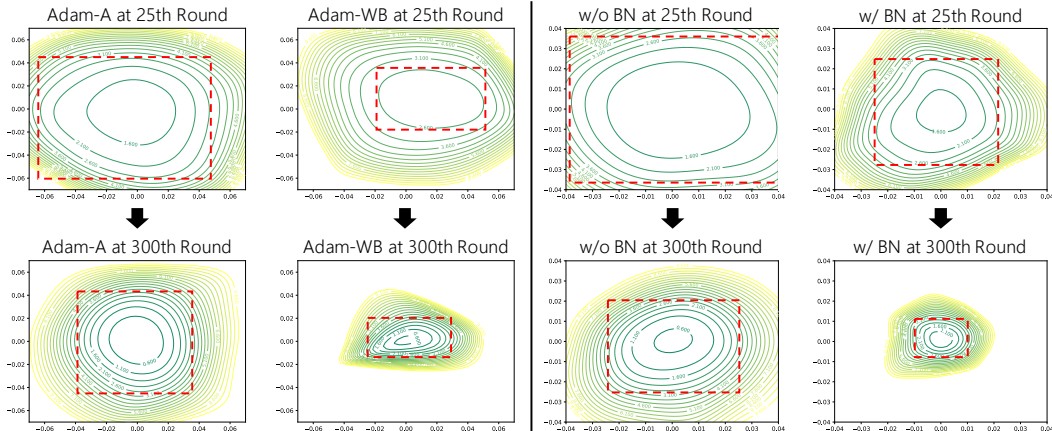

Figure 5: Loss surface of the global model parameters under Non-IID(2) setting. The red dashed boxes indicate the same level of loss value.

plausibly at very early rounds, but the growth rate quickly stagnates and eventually becomes much lower than the comparison targets.

The left plot of Figure 4 shows the effect of the Adam optimizer with respect to its implementations. Through the experiments, we identified that under non-IID data environments, the performance of Adam is very sensitive to the range of model variables to be averaged, unlike the non-adaptive optimizers (e.g., momentum SGD); its moment variables should be also considered in the parameter averaging together with weights and biases (see also Table 3). The poor performance of the Adam-WB under the Non-IID(2) setting would be from twice as many momentum variables as the momentum SGD, which indicates the increased number of them affected by the non-IIDness; thus, originally we had thought that extreme parameter divergence could appear if the momentum variables are not averaged together with weights and biases. However, it was seen that the parameter divergence values under the Adam-WB was seen to be similar or even smaller than under Adam-A (see also Figure 11 in the appendix). Instead, from the left panel we can observe that the parameter divergence of Adam-WB in the last fully-connected layer is bigger than that of Adam-A at the very early rounds (as we expected), but soon it is abnormally sharply reduced over rounds; this is considered the steep fall phenomenon.

The middle and the right plots of the figure also show the steep fall phenomenon in the last fully-connected layer, with respect to network width and whether to use Batch Normalization, respectively. In the case of the NetC models, NetC-Baseline, NetC-Wider, and NetC-Widest use the global average pooling, the max pooling with stride 4, and the max pooling with stride 2, respectively, after the last convolutional layer; the number of neurons in the output layer becomes 2560, 10240, and 40960, respectively (see also Appendix A.1 for their detailed architecture). Under the Non-IID(2) setting, the corresponding test accuracy was found to be 64.06%, 72.61%, and 73.64%, respectively, in order of the degree of wideness. In addition, we can see that under Non-IID(2), Batch Normalization[9] yields not only big parameter divergence (especially before the first learning rate drop) but also the steep fall phenomenon; the corresponding test accuracy was seen to be very low (see Table 3). The failure of the Batch Normalization stems from that the dependence of batch-normalized hidden activations makes each learner's update too overfitted to the distribution of their local training data. Batch Renormalization, by relaxing the dependence, yields a better outcome; however, it still fails to exceed the performance of the baseline due to the significant parameter divergence.

To explain the impact of the steep fall phenomenon in test accuracy, we provide Figure 5, which indicates that the loss landscapes for the failure cases (e.g., Adam-WB and with Batch Normalization) commonly show *sharper* minima that leads to poorer generalization (Hochreiter & Schmidhuber,

---

[9]For its implementations into the considered federated learning algorithm, we let the server get the proper moving variance by $\frac{1}{K} \sum_{k \in \mathcal{K}} \left( \mathbb{E}\left[\phi^2\right] \right)_k - \mathbb{E}\left[\phi\right]^2$ at each round, by allowing each learner $k$ collect $\left( \mathbb{E}\left[\phi^2\right] \right)_k$ as well as the existing moving statistics of Batch Normalization ($\phi$ denotes activations). This can be regarded as a federated version of distributed Batch Normazliation methods (e.g., Qin et al. (2018); Zhang et al. (2018)).

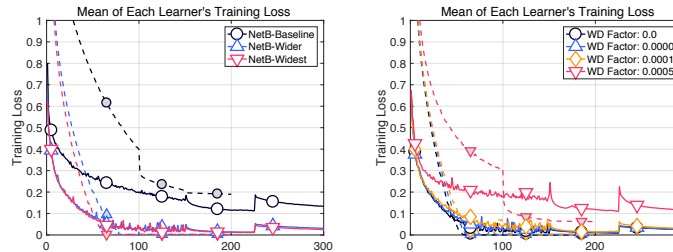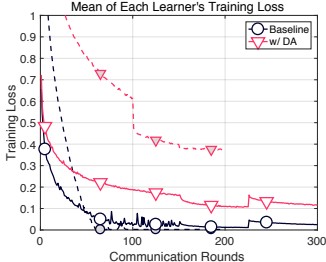

Figure 6: Excessively high training loss in the CIFAR-10 training with NetB-Baseline (left), the weight decay factor of 0.0005 (middle), and data augmentation (right), respectively, under the non-IID data setting. Dotted and solid lines indicate the results under IID and Non-IID(2) setting, respectively. WD: weight decay; DA: data augmentation. Note that training loss values of each learner were calculated on their local data before each synchronization.

Table 4: Test accuracy (%) of the trials in Figure 6. Note that weight decay factor of 0.00005 is the baseline configuration.

| Network | Method | Vanilla Training | | Federated Learning | |
|---|---|---|---|---|---|
| | | BS: 50 | BS: 500 | IID | Non-IID(2) |
| NetB-Baseline | NMom-A | 85.05 (85.13) | 79.74 (79.79) | 82.57 (82.65) | **62.82 (63.08)** |
| NetB-Wider | NMom-A | 82.74 (82.90) | 77.29 (77.47) | 80.03 (80.43) | 72.24 (72.32) |
| NetB-Widest | NMom-A | 83.59 (83.60) | 79.31 (79.77) | 81.47 (81.73) | 74.35 (74.52) |
| NetA-Baseline | NMom-A | 83.89 (83.90) | 79.39 (79.49) | 81.49 (81.82) | 74.11 (74.23) |
| NetA-Baseline | NMom-A + WD: 0.0 | 81.13 (81.15) | 79.09 (79.25) | 81.68 (81.90) | 73.95 (74.27) |
| NetA-Baseline | NMom-A + WD: 0.0001 | 84.29 (84.46) | 79.13 (79.74) | 82.22 (82.25) | 72.65 (72.79) |
| NetA-Baseline | NMom-A + WD: 0.0005 | 83.66 (84.60) | 80.48 (80.79) | 82.69 (83.14) | **54.11 (54.15)** |
| NetA-Baseline | NMom-A + DA | 87.16 (87.41) | 83.07 (83.35) | 84.85 (84.89) | **73.95 (74.46)** |

1997; Keskar et al., 2017), and the minimal value in the bowl is relatively greater.[10] Here it is also observed that going into sharp minima starts even in early rounds such as 25th.

**Excessively high training loss of local updates.** The final cause that we consider for the failure cases is excessively high training loss of local updates. For instance, from the left plot of Figure 6, we see that under the Non-IID(2) setting, NetB-Baseline gives much higher training loss than the other models. Here we note that for the NetB-Baseline model, the global average pooling is applied after the last convolutional layer, and the number of neurons in the first fully-connected layer thus becomes $256 \cdot 256$; on the other hand, NetB-Wider and NetB-Widest use the max pooling with stride 4 and 2, which make the number of neurons in that layer become $1024 \cdot 256$ and $4096 \cdot 256$, respectively (see also Appendix A.1 for their details). The experimental results were shown that NetB-Baseline has notably lower test accuracy (see Table 4). We additionally remark that for NetB-Baseline, very high losses are observed under the IID setting, and their values even are greater than in the non-IID case; however, note that one have to be aware that local updates are extremely easy to be overfitted to each training dataset under non-IID data environments, thus the converged training losses being high is more critical than the IID cases.

The middle and the right plot of the figure show the excessive training loss under the non-IID setting when applying the weight decay factor of 0.0005 and the data augmentation, respectively. In the cases of the high level of weight decay, the severe performance degradation appears compared to when the levels are low (i.e., $\leq 0.0001$) as already discussed. In addition, we observed that with Nesterov momentum SGD, the data augmentation yields a diminishing return in test accuracy (i.e., with the data augmentation we can achieve $+3.36\%$ under IID, but $-0.16\%$ is obtained under non-IID(2), compared to when it is not applied); with Adam the degree of the diminishment becomes higher (refer to Table 12 in the appendix). In the data augmentation cases, judging from that the

---

[10]Based on Li et al. (2018), the visualization of loss surface was conducted by $L(\alpha, \beta) = \ell(\theta^* + \alpha\delta + \beta\gamma)$, where $\theta^*$ is a center point of the model parameters, and $\delta$ and $\gamma$ is the orthogonal direction vectors.

Table 5: Summary of the failure cases under the non-IID data setting. PD: parameter divergence.

| Failure Cases | High-level Reasons | Observed Internal Causes | | |
| --- | --- | --- | --- | --- |
| | | Magnitude of PD | Steep Fall Phenomenon | High Local Training Loss |
| Adam-WB | Adam requires a much larger number of model variables than the other optimizers, which indicates the increased number of them stimulated by the data non-IIDness | | ✓ | ✓ |
| Depth++ | The increased number of model parameters could lead to bigger parameter divergence | ✓ | | |
| Width-- | - | ✓ | ✓ | ✓ |
| WD++ | It makes the model parameters in previous iterations to be reflected to the current ones; thus the previous parameter divergence also affects the present | ✓ | | ✓ |
| w/ BN | It makes local updates of learners biased to each data distribution; it could aggravate the parameter divergence | ✓ | ✓ | |
| w/ DA | - | | | ✓ |
| w/ DO | The dropped nodes (or neurons), randomly selected, are different across learners; its impact on the parameter divergence would be much stronger than under the IID setting | ✓ | | |

parameter divergence values are not so different between with and without it, we can identify that the performance degradation stems from the high training loss (see Figures 30 and 31 in the appendix). Here we additionally note that unlike on the CIFAR-10, in the experiments on SVHN it was seen that the generalization effect of the data augmentation is still valid in test accuracy (see Table 12).

## 5 CONCLUSION

In this paper, we explored the effects of various hyperparameter optimization strategies for optimizers, network depth/width, and regularization on federated learning of deep networks. Our primary concern in this study was lied on non-IID data, in which we found that under non-IID data settings many of the probed factors show somewhat different behaviors compared to under the IID setting and vanilla training. To explain this, a concept of the parameter divergence was utilized, and its origin was identified both empirically and theoretically. We also provided the internal reasons of our observations with a number of the experimental cases.

In the meantime, the federated learning has been vigorously studied for decentralized data environments due to its inherent strength, i.e., high communication-efficiency and privacy-preservability. However, so far most of the existing works mainly dealt with only IID data, and the research to address non-IID data has just entered the beginning stage very recently despite its high real-world possibility. Our study, as one of the openings, handles the essential factors in the federated training under the non-IID data environments, and we expect that it will provide refreshing perspectives for upcoming works.

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

# A EXPERIMENTAL DETAILS

## A.1 NETWORK CONFIGURATIONS

In the experiments, we consider CNN architectures, as illustrated in Figure 7. In the network configurations, three groups of $3 \times 3$ convolutional layers are included that have $16 \cdot m$, 128, and 256 output channels, respectively; $n$ denotes the number of the layers in each convolutional group. The first two groups are followed by $3 \times 3$ max pooling with stride 2; the last convolutional layer is followed by either the $3 \times 3$ max pooling with stride $s$ or the global average pooling. In the case of fully-connected layers, we use two types of the stacks: (i) three layers, of which the output sizes are $256 \cdot u$, 256, and 10, respectively; and (ii) a single layer, of which the output size is 10. In addition, we use the ReLU and the softmax activation for the hidden weight layers and the output layer, respectively. Table 6 summarizes the network models used in the experiments.

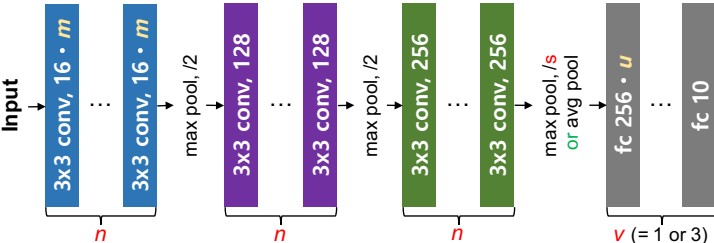

Figure 7: The network configurations.

Table 6: The network models used in the experiments.

|  | $n$ | $m$ | Pooling after the Last Convolutional Layer | $v$ | $u$ |
|---|---|---|---|---|---|
| NetA-Baseline | 1 | 4 | max pool, $/2$ | 3 | 2 |
| NetA-Deeper | 2 | 4 | max pool, $/2$ | 3 | 2 |
| NetA-Deepest | 3 | 4 | max pool, $/2$ | 3 | 2 |
| NetA-Narrowest | 1 | 1 | max pool, $/2$ | 3 | 2 |
| NetA-Narrower | 1 | 2 | max pool, $/2$ | 3 | 2 |
| NetA-Baseline | 1 | 4 | max pool, $/2$ | 3 | 2 |
| NetB-Baseline | 1 | 4 | avg pool | 3 | 1 |
| NetB-Wider | 1 | 4 | max pool, $/4$ | 3 | 1 |
| NetB-Widest | 1 | 4 | max pool, $/2$ | 3 | 1 |
| NetC-Baseline | 1 | 4 | avg pool | 1 | - |
| NetC-Wider | 1 | 4 | max pool, $/4$ | 1 | - |
| NetC-Widest | 1 | 4 | max pool, $/2$ | 1 | - |

## A.2 THE REMAINING CONFIGURATIONS

In the experiments, we initialize the network models to mostly follow the truncated normal distribution with a mean of 0 based on He et al. (2015), however we fix the standard deviation to 0.05 for the first convolutional group and the last fully-connected layer. For training, minibatch stochastic optimization with cross-entropy loss is considered. Specifically, we use pure SGD, Nesterov momentum SGD (Polyak, 1964; Nesterov, 1983), and Adam (Kingma & Ba, 2015) as optimization methods; initial learning rates are set to 0.05, 0.01, and 0.001, respectively for each optimizer. We drop the learning rate by 0.1 at 50% and 75% of the total training iterations, respectively. Regarding the environmental configurations, we predetermine each of learners' local training dataset in a random seed; the training examples are allocated so that they do not overlap between the learners. To report the experimental results, we basically considered to run the trials once, but as for unstable ones in the preliminary tests, we chose the middle results of several runs. In every result plot, the values are plotted at each round.

## B  THEORETICAL ANALYSIS ON THE ORIGIN OF PARAMETER DIVERGENCE

In relation to the federated learning under non-IID data, so far there have been several works for providing theoretical bounds to explain how does the degree of the non-IIDness of decentralized data affect the performance, with respect to its degree (e.g., Zhao et al. (2018); Li et al. (2019b;c); Liu et al. (2019); Stich (2019); Wang et al. (2019); Yu et al. (2019)). Inspired by them, here we further study how does the non-IIDness make the model parameters of each learner diverged.

In this analysis, we consider training deep networks for multi-class classification. Based on the notations in Algorithm 1, the SGD update of learner $k$ at round $t + 1$ is given as

$$\mathbf{w}_k^{t+1} \triangleq \overline{\mathbf{w}}^t - \eta \sum_{\tau=0}^{R-1} \sum_{q \in \mathcal{Q}} p_k(y = q) \mathbb{E}_{\mathbf{x} \in \mathcal{P}_k | y = q} \left[ \nabla_{\mathbf{w}} \log f_q(\mathbf{x}; \mathbf{w}_k^{t+\frac{\tau}{R}}) \right],$$

where $f_q(\mathbf{x}; \mathbf{w})$ is the posterior probability for class $q \in \mathcal{Q}$ ($\mathcal{Q}$ is the label space), obtained from model parameters $\mathbf{w}$ with data examples $(\mathbf{x}, y)$, and $p_k(y = q)$ is the probability that the label of a data example in $\mathcal{P}_k$ is $q$. In this equation, $\mathbf{w}_k^{t+\frac{\tau}{R}}$ is the model parameters after the $\tau$-th local iterations in the round $t + 1$ ($R$ is the number of local iterations of each learner per round). Herein we note that $\mathbf{w}_k^{t+0}$ ($\triangleq \overline{\mathbf{w}}^t$) is the global model parameters received from the server at the round $t + 1$; we use the term to distinguish it from the term $\mathbf{w}_k^t$ (which indicates the local update that has sent back to the server at round $t$).

Then, by the linearity of the gradient, we obtain

$$(\mathbf{d}_q)_k^{t+1} = (\overline{\mathbf{d}}_q)^t - \eta \sum_{\tau=0}^{R-1} p_k(y = q) \nabla_{\mathbf{d}_q} \mathbb{E}_{\mathbf{x} \in \mathcal{P}_k | y = q} \left[ \log f_q(\mathbf{x}; \mathbf{w}_k^{t+\frac{\tau}{R}}) \right],$$

where $\mathbf{d}_q$ denote the neurons, in the (dense) output layer of the model $\mathbf{w}$, that are connected to the output node for class $q$.

**Parameter divergence between $(\mathbf{d}_q)_k^{t+1}$ with the fixed $k$.**  At round $t+1$, suppose that for learner $k$, $\left\| (\mathbf{a}_q)_k^{t+\frac{\tau}{R}} \right\| \leq G, \forall q \in \mathcal{Q}, 0 \leq \tau < R$, and $\left\| (\mathbf{a}_i)_k^{t+\frac{\tau}{R}} - (\mathbf{a}_j)_k^{t+\frac{\tau}{R}} \right\| \leq H, \forall i, j \in \mathcal{Q}, 0 \leq \tau < R$, where $(\mathbf{a}_q)_k^{t+\frac{\tau}{H}} \triangleq \nabla_{\mathbf{d}_q} \mathbb{E}_{\mathbf{x} \in \mathcal{P}_k | y = q} \left[ \log f_q(\mathbf{x}; \mathbf{w}_k^{t+\frac{\tau}{R}}) \right]$. Then, we can get

$$\left\| (\mathbf{d}_i)_k^{t+1} - (\mathbf{d}_j)_k^{t+1} \right\|$$

$$= \left\| (\overline{\mathbf{d}}_i)^t - (\overline{\mathbf{d}}_j)^t - \eta \sum_{\tau=0}^{R-1} \left( p_k(y = i)(\mathbf{a}_i)_k^{t+\frac{\tau}{R}} - p_k(y = j)(\mathbf{a}_j)_k^{t+\frac{\tau}{R}} \right) \right\|$$

$$\leq \left\| (\overline{\mathbf{d}}_i)^t - (\overline{\mathbf{d}}_j)^t \right\| + \eta \sum_{\tau=0}^{R-1} \left\| p_k(y = i)(\mathbf{a}_i)_k^{t+\frac{\tau}{R}} - p_k(y = j)(\mathbf{a}_j)_k^{t+\frac{\tau}{R}} \right\|$$

$$= \left\| (\overline{\mathbf{d}}_i)^t - (\overline{\mathbf{d}}_j)^t \right\| + \eta \sum_{\tau=0}^{R-1} \left\| p_k(y = i)(\mathbf{a}_i)_k^{t+\frac{\tau}{R}} - p_k(y = i)(\mathbf{a}_j)_k^{t+\frac{\tau}{R}} \right.$$

$$\left. + p_k(y = i)(\mathbf{a}_j)_k^{t+\frac{\tau}{R}} - p_k(y = j)(\mathbf{a}_j)_k^{t+\frac{\tau}{R}} \right\|$$

$$\leq \left\| (\overline{\mathbf{d}}_i)^t - (\overline{\mathbf{d}}_j)^t \right\| + \eta p_k(y = i) \sum_{\tau=0}^{R-1} \left\| (\mathbf{a}_i)_k^{t+\frac{\tau}{R}} - (\mathbf{a}_j)_k^{t+\frac{\tau}{R}} \right\|$$

$$+ \eta \left\| p_k(y = i) - p_k(y = j) \right\| \sum_{\tau=0}^{R-1} \left\| (\mathbf{a}_j)_k^{t+\frac{\tau}{R}} \right\|$$

$$\leq \left\| (\overline{\mathbf{d}}_i)^t - (\overline{\mathbf{d}}_j)^t \right\| + \eta R H p_k(y = i) + \eta R G \left\| p_k(y = i) - p_k(y = j) \right\|. \tag{1}$$

From this, we can identify that the parameter difference, $\left\| (\mathbf{d}_i)_k^{t+1} - (\mathbf{d}_j)_k^{t+1} \right\|$ is bounded according to $\left\| p_k(y = i) - p_k(y = j) \right\|$; it corresponds the results in Figure 1.

Table 7: Test accuracy (%) comparison of NetA-Baseline with respect to optimizers. Values outside/inside brackets denote one measured after the whole training and the highest one during the rounds, respectively. PMom: Polyak momentum SGD; NMom: Nesterov momentum SGD; BS: minibatch size.

| Dataset | Method | Vanilla Training | | Federated Learning | |
|---------|--------|------------------|------------------|--------------------|--------------------|
| | | BS: 50 | BS: 500 | IID | Non-IID(2) |
| CIFAR-10 | Pure SGD | 82.47 (82.72) | 78.97 (79.06) | 81.12 (81.36) | 68.92 (69.57) |
| CIFAR-10 | PMom-WB | - | - | 81.56 (81.82) | 74.30 (74.57) |
| CIFAR-10 | PMom-A | - | - | 81.78 (81.83) | 74.82 (75.06) |
| CIFAR-10 | NMom-WB | 83.89 (83.90) | 79.39 (79.49) | 81.56 (81.99) | 74.21 (74.31) |
| CIFAR-10 | NMom-A | 83.89 (83.90) | 79.39 (79.49) | 81.49 (81.82) | 74.11 (74.23) |
| CIFAR-10 | Adam-WB | 81.97 (82.35) | 80.73 (80.76) | 80.99 (81.00) | **67.51 (67.82)** |
| CIFAR-10 | Adam-A | 81.97 (82.35) | 80.73 (80.76) | 81.18 (81.45) | 75.32 (75.45) |
| SVHN | Adam-WB | - | - | 92.82 (93.12) | **88.70 (88.91)** |
| SVHN | Adam-A | - | - | 92.89 (93.28) | 90.72 (90.95) |

**Parameter divergence between** $(\mathbf{d}_q)_k^{t+1}$ **with the fixed** $q$**.** At round $t+1$, suppose that for class $q$, $\left\|(\mathbf{a}_q)_k^{t+\frac{\tau}{R}}\right\| \leq G'$, $\forall k \in \mathcal{K}, 0 \leq \tau < R$, and $\left\|(\mathbf{a}_q)_i^{t+\frac{\tau}{R}} - (\mathbf{a}_q)_j^{t+\frac{\tau}{R}}\right\| \leq H'$, $\forall i, j \in \mathcal{K}, 0 \leq \tau < R$. Then, similar with Equation 1, we can have

$$\left\|(\mathbf{d}_q)_i^{t+1} - (\mathbf{d}_q)_j^{t+1}\right\| \leq \eta R H' p_i(y = q) + \eta R G' \left\|p_i(y = q) - p_j(y = q)\right\|. \tag{2}$$

## C  THE COMPLETE EXPERIMENTAL RESULTS

In this section we provide our complete experimental results. Before the main statement, we first note that in the following figures, $C_{ij}$ denotes the $j$-th convolutional layer in the $i$-th group, and $F_j$ denotes the $j$-th fully-connected layer (in relation, refer to Appendix A.1). In addition, we remind that in this paper "vanilla" training refers to non-distributed training with a single machine, using the whole data examples; for the vanilla training, we trained the networks for 100 epochs.

### C.1  THE EFFECT OF OPTIMIZERS

Here we investigate the effect of optimizers. We importantly note that both momentum SGD and Adam require the additional variables related to momentum as well as weights and biases; the terms *(optimizer name)-A* and *(optimizer name)-WB* are used to refer to the parameter averaging being performed for all the variables and only for weights & biases, respectively.

The experimental results are provided in Table 7 and Figures 10 and 11. From the table, interestingly we can notice that under the non-IID data setting, there exists a huge performance gap between Adam-A and Adam-WB ($\approx 7\%$), unlike the momentum SGD trials. At the initial steps of this study, we had thought that the poor performance of Adam-WB would be from the following: Since Adam requires twice as many momentum variables as momentum SGD, extreme parameter divergence could appear if they are not averaged together with weights and biases. However, unlike our expectations, the parameter divergence values under the Adam-WB was seen to be similar or even smaller than under Adam-A. Nevertheless, we can observe the followings for the non-IID cases: First, the parameter divergence of Adam-WB in $F_3$ is bigger than that of Adam-A at the very early rounds (as we expected), but soon it is abnormally sharply reduced over rounds; this can be considered the steep fall phenomenon. Second, Adam-WB leads to higher training loss of each learner. We guess that these two caused the severe degradation of Adam-WB in test accuracy.

### C.2  THE EFFECT OF NETWORK DEPTH

Here we investigate the effect of network depth. Since deepening networks also indicates that there becomes having more parameters to be averaged in the considered federated learning algorithm, we

Table 8: Test accuracy (%) comparison with respect to network depth. Values outside/inside brackets denote one measured after the whole training and the highest one during the rounds, respectively. Note that ResNet-14 and ResNet-20 have 14 layers (13 convolutional layers + 1 fully-connected layer) and 20 layers (19 convolutional layers + 1 fully-connected layer), respectively; they use the global average pooling after the last convolutional layer, in common with NetC-Baseline. For reference, in our trials with Batch Normalization and data augmentation, the ResNet-20 could achieve test accuracy of 89.18 and 89.07 under Nesterov Momentum SGD and Adam, respectively.

| Dataset | Network | Method | Vanilla Training | | Federated Learning | |
|---|---|---|---|---|---|---|
| | | | BS: 50 | BS: 500 | IID | Non-IID(2) |
| CIFAR-10 | NetA-Baseline | PMom-A | - | - | 81.78 (81.83) | 74.82 (75.06) |
| CIFAR-10 | NetA-Deeper | PMom-A | - | - | 82.93 (83.26) | 74.73 (75.16) |
| CIFAR-10 | NetA-Deepest | PMom-A | - | - | 81.99 (82.75) | **69.77 (70.42)** |
| CIFAR-10 | NetA-Baseline | NMom-A | 83.89 (83.90) | 79.39 (79.49) | 81.49 (81.82) | 74.11 (74.23) |
| CIFAR-10 | NetA-Deeper | NMom-A | 85.94 (86.00) | 81.23 (81.29) | 83.40 (83.63) | 73.67 (73.89) |
| CIFAR-10 | NetA-Deepest | NMom-A | 86.20 (86.33) | 79.48 (79.53) | 83.12 (83.58) | **68.98 (69.64)** |
| SVHN | NetA-Baseline | NMom-A | - | - | 93.02 (93.26) | 89.28 (89.70) |
| SVHN | NetA-Deepest | NMom-A | - | - | 94.42 (94.71) | **89.12 (89.84)** |
| CIFAR-10 | NetC-Baseline | NMom-A | 84.25 (84.50) | 76.70 (76.82) | 80.60 (80.75) | 64.06 (64.55) |
| CIFAR-10 | ResNet-14 | NMom-A | - | - | 81.24 (81.64) | **62.14 (62.47)** |
| CIFAR-10 | ResNet-20 | NMom-A | - | - | 82.30 (82.77) | **59.98 (59.98)** |
| CIFAR-10 | NetC-Baseline | Adam-A | 82.59 (83.42) | 79.71 (80.89) | 81.72 (81.80) | 68.41 (68.75) |
| CIFAR-10 | ResNet-14 | Adam-A | - | - | 79.32 (79.99) | **64.46 (64.92)** |
| CIFAR-10 | ResNet-20 | Adam-A | - | - | 78.40 (79.28) | **63.50 (64.06)** |

had predicted especially under non-IID data settings that depending on their depth, it would yield bigger parameter divergence in the whole model and the consequent diminishing returns compared to under the vanilla training and the IID data setting; the test accuracy results show it as expected (see Table 8).[11] Moreover, it is also seen from Figure 12 that parameter divergence increases also qualitatively (i.e., in a layer level) under the non-IID data setting, as the number of convolutional layers increases. Note that for $C_{21}$ and $C_{31}$, the divergence pattern is resulted as opposed to that of $C_{11}$ and $F_3$; however, the values of $C_{11}$ and $F_3$ would be more impactful as mentioned in Footnote 8.

We additionally remark from the figure that the sharp reduction of parameter divergence (in the convolutional layers) at the very early rounds when using NetA-Deepest indicates the parameter averaging algorithm did not work properly. Correspondingly, the test accuracy values in the early period were seen to be not much different from the initial one.

### C.3    The Effect of Network Width

Following the previous subsection, from now on we investigate the effect of network width. Contrary to the results in the Section C.2, it is seen from Table 9 that widening networks provides positive effects for the considered federated learning algorithm under the non-IID data setting. Especially, one can see that compared to the max pooling trials, while the global average pooling yields higher test accuracy in the vanilla training (with the minibatch size of 50), its performance gets significantly worse under the non-IID data setting (remind that NetB-Baseline and NetC-Baseline use the global average pooling after the last convolutional layer). Focusing on the NetC models, we here make the following observations for the non-IID data setting from Figures 15, 18, and 21: First, the considered federated learning algorithm provides bigger parameter divergence in $F_1$ as its width decreases (note that each input size of $F_1$ is 256, 1024, and 4096 for NetC-Baseline, NetC-Wider, and NetC-Widest, respectively), especially during the beginning rounds (e.g., for the NMom-A case, until about 50 rounds). Unlike in Section C.2, here we can identify that even though the parameter size of is the smallest under the global averaging pooling, it rather yields the biggest qualitative parameter divergence. Second, the steep fall phenomenon appears in $F_1$ for the NetC-

---

[11]Note that for the SVHN cases, the test accuracy of NetA-Deepest is higher than that of NetA-Baseline under the non-IID data setting, but the increase is much smaller than under the IID data setting.

Table 9: Test accuracy (%) comparison with respect to network width. Values outside/inside brackets denote one measured after the whole training and the highest one during the rounds, respectively.

| Dataset | Network | Method | Vanilla Training | | Federated Learning | |
|---|---|---|---|---|---|---|
| | | | BS: 50 | BS: 500 | IID | Non-IID(2) |
| CIFAR-10 | NetA-Narrowest | pure SGD | 81.64 (81.76) | 76.68 (77.06) | 78.78 (78.93) | 67.66 (67.98) |
| CIFAR-10 | NetA-Narrower | pure SGD | 82.46 (82.68) | 78.32 (78.38) | 79.58 (79.72) | 68.93 (69.28) |
| CIFAR-10 | NetA-Baseline | pure SGD | 82.47 (82.72) | 78.97 (79.06) | 81.12 (81.36) | 68.92 (69.57) |
| CIFAR-10 | NetB-Baseline | pure SGD | 84.18 (84.61) | 69.22 (69.22) | 74.58 (74.59) | **54.39 (54.84)** |
| CIFAR-10 | NetB-Wider | pure SGD | 81.41 (81.59) | 76.69 (77.57) | 79.49 (79.81) | 65.06 (65.29) |
| CIFAR-10 | NetB-Widest | pure SGD | 82.60 (82.65) | 79.10 (79.27) | 80.31 (80.37) | 68.77 (69.20) |
| CIFAR-10 | NetC-Baseline | pure SGD | 83.64 (83.69) | 68.46 (68.47) | 72.64 (72.70) | **58.13 (58.51)** |
| CIFAR-10 | NetC-Wider | pure SGD | 79.88 (79.96) | 76.69 (76.87) | 78.08 (78.24) | 68.30 (68.65) |
| CIFAR-10 | NetC-Widest | pure SGD | 82.11 (82.21) | 78.28 (78.34) | 79.94 (79.97) | 70.33 (70.51) |
| CIFAR-10 | NetC-Baseline | PMom-A | - | - | 79.69 (79.84) | **64.42 (64.54)** |
| CIFAR-10 | NetC-Widest | PMom-A | - | - | 80.12 (80.36) | 74.68 (75.18) |
| CIFAR-10 | NetA-Narrowest | NMom-A | 82.03 (82.06) | 77.75 (77.92) | 80.75 (81.02) | 72.57 (72.64) |
| CIFAR-10 | NetA-Narrower | NMom-A | 83.50 (83.50) | 78.11 (78.59) | 81.29 (81.44) | 73.44 (73.60) |
| CIFAR-10 | NetA-Baseline | NMom-A | 83.89 (83.90) | 79.39 (79.49) | 81.49 (81.82) | 74.11 (74.23) |
| CIFAR-10 | NetB-Baseline | NMom-A | 85.05 (85.13) | 79.74 (79.79) | 82.57 (82.65) | **62.82 (63.08)** |
| CIFAR-10 | NetB-Wider | NMom-A | 82.74 (82.90) | 77.29 (77.47) | 80.03 (80.43) | 72.24 (72.32) |
| CIFAR-10 | NetB-Widest | NMom-A | 83.59 (83.60) | 79.31 (79.77) | 81.47 (81.73) | 74.35 (74.52) |
| CIFAR-10 | NetC-Baseline | NMom-A | 84.25 (84.50) | 76.70 (76.82) | 80.60 (80.75) | **64.06 (64.55)** |
| CIFAR-10 | NetC-Wider | NMom-A | 81.48 (81.53) | 77.00 (77.14) | 79.59 (79.85) | 72.61 (72.97) |
| CIFAR-10 | NetC-Widest | NMom-A | 83.16 (83.36) | 78.39 (78.56) | 80.90 (81.14) | 73.64 (73.91) |
| SVHN | NetC-Baseline | NMom-A | - | - | 92.64 (92.85) | **82.69 (83.35)** |
| SVHN | NetC-Widest | NMom-A | - | - | 92.19 (92.50) | 89.55 (90.62) |
| CIFAR-10 | NetA-Narrowest | Adam-A | 80.62 (81.02) | 79.72 (79.72) | 79.82 (79.88) | 75.06 (75.18) |
| CIFAR-10 | NetA-Narrower | Adam-A | 82.23 (82.51) | 80.80 (80.80) | 81.04 (81.09) | 74.88 (74.88) |
| CIFAR-10 | NetA-Baseline | Adam-A | 81.97 (82.35) | 80.73 (80.76) | 81.18 (81.45) | 75.32 (75.45) |
| CIFAR-10 | NetB-Baseline | Adam-A | 83.52 (83.62) | 80.98 (81.61) | 81.68 (82.29) | **71.40 (71.45)** |
| CIFAR-10 | NetB-Wider | Adam-A | 81.20 (81.39) | 79.49 (79.62) | 80.22 (80.39) | 74.42 (74.67) |
| CIFAR-10 | NetB-Widest | Adam-A | 82.08 (82.29) | 80.87 (80.87) | 81.02 (81.20) | 74.93 (75.41) |
| CIFAR-10 | NetC-Baseline | Adam-A | 82.59 (83.42) | 79.71 (80.89) | 81.72 (81.80) | **68.41 (68.75)** |
| CIFAR-10 | NetC-Wider | Adam-A | 80.77 (80.94) | 77.31 (77.48) | 78.81 (79.09) | 75.09 (75.22) |
| CIFAR-10 | NetC-Widest | Adam-A | 81.53 (81.59) | 78.80 (78.92) | 79.91 (80.08) | 76.57 (76.57) |

Baseline case. Third, the global average pooling gives too high training loss of each learner. All the three observations fit well into the failure of the global average pooling.

We additionally note that when using NetC-Baseline, the results under the IID data setting shows very high loss values; this leads to diminishing returns for the pure SGD and the NMom-A cases, compared to the vanilla training results with the minibatch size of 50. However, the corresponding degradation rate is seen to be much higher under the non-IID data setting. This is because the local updates are extremely easy to be overfitted to the training data under the non-IID data setting; thus the converged training losses being high becomes much more critical.

## C.4 THE EFFECT OF WEIGHT DECAY

Here we investigate the effect of weight decay. From Table 10, it is seen that under the non-IID data setting we should apply much smaller weight decay for the considered federated learning algorithm than under the vanilla training or the IID data setting. For its internal reason, Figures 22, 23, and 24 show that under the non-IID data setting, the considered federated learning algorithm not only converges to too high training loss (of each learner) but also causes excessive parameter divergence when the weight decay factor is set to 0.0005. Here we note that if a single iteration is considered for each learner's local update per round, the corresponding parameter divergence will be of course the same without regard to degree of weight decay. However, in our experiments, the great number

Table 10: Test accuracy (%) comparison of NetA-Baseline with respect to weight decay levels. Values outside/inside brackets denote one measured after the whole training and the highest one during the rounds, respectively. WD: weight decay.

| Dataset | Method | WD Factor | Vanilla Training | | Federated Learning | |
|---------|--------|-----------|------------------|------------------|------------------|------------------|
| | | | BS: 50 | BS: 500 | IID | Non-IID(2) |
| CIFAR-10 | pure SGD | 0.0 | 81.33 (81.33) | 78.00 (78.16) | 79.88 (80.09) | 68.85 (69.79) |
| CIFAR-10 | pure SGD | 0.00005 | 82.47 (82.72) | 78.97 (79.06) | 81.12 (81.36) | 68.92 (69.57) |
| CIFAR-10 | pure SGD | 0.0001 | 82.00 (82.19) | 79.58 (79.74) | 81.28 (81.40) | 68.95 (69.03) |
| CIFAR-10 | pure SGD | 0.0005 | 82.39 (82.54) | 78.92 (78.95) | 80.28 (80.28) | **47.43 (47.66)** |
| CIFAR-10 | PMom-A | 0.00005 | - | - | 81.78 (81.83) | 74.82 (75.06) |
| CIFAR-10 | PMom-A | 0.0005 | - | - | 82.67 (83.21) | **55.57 (56.25)** |
| CIFAR-10 | NMom-A | 0.0 | 81.13 (81.15) | 79.09 (79.25) | 81.68 (81.90) | 73.95 (64.27) |
| CIFAR-10 | NMom-A | 0.00005 | 83.89 (83.90) | 79.39 (79.49) | 81.49 (81.82) | 74.11 (74.23) |
| CIFAR-10 | NMom-A | 0.0001 | 84.29 (84.46) | 79.13 (79.74) | 82.22 (82.25) | 72.65 (72.79) |
| CIFAR-10 | NMom-A | 0.0005 | 83.66 (84.60) | 80.48 (80.79) | 82.69 (83.14) | **54.11 (54.15)** |
| SVHN | NMom-A | 0.00005 | - | - | 93.02 (93.26) | 89.28 (89.70) |
| SVHN | NMom-A | 0.0005 | - | - | 93.23 (93.72) | **76.31 (76.82)** |
| CIFAR-10 | Adam-A | 0.0 | 80.70 (80.73) | 80.01 (80.01) | 80.11 (80.19) | 74.77 (74.93) |
| CIFAR-10 | Adam-A | 0.00005 | 81.97 (82.35) | 80.73 (80.76) | 81.18 (81.45) | 75.32 (75.45) |
| CIFAR-10 | Adam-A | 0.0001 | 82.79 (83.19) | 81.27 (81.30) | 81.74 (81.84) | 75.95 (76.02) |
| CIFAR-10 | Adam-A | 0.0005 | 82.93 (84.44) | 81.34 (81.54) | 82.33 (83.06) | **69.75 (69.75)** |

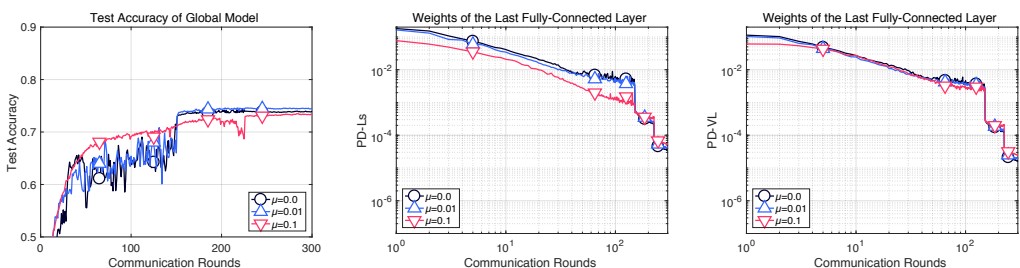

Figure 8: Comparison of test accuracy and parameter divergence under Non-IID(2) setting with respect to FedProx factor $\mu$ in the training of NetA-Baseline on CIFAR-10 (Optimizer: NMom-A).

of local iterations per round (i.e., 100) made a big difference of the divergence values under the non-IID data setting; this eventually yielded the accuracy gap. In addition, we further observe under the non-IID data setting that even with the weight decay factor of 0.0005, the test accuracy increases similarly with its smaller values at very early rounds, in which the norm values of the weights are relatively much smaller.

Moreover we also conducted additional experiments for the related regularization techniques, FedProx (Li et al., 2019b). Under the FedProx, in order to make local updates do not deviate excessively from the current global model parameters, at each round $t$ each learner uses the following surrogate loss function that adds a proximal term to the original objective function: $\ell(\mathbf{w}) + \frac{\mu}{2}\left\|\mathbf{w} - \overline{\mathbf{w}}^{t-1}\right\|^2$. Figure 8 shows the experimental results; as seen from the figure, in our implementation FedProx did not provide dramatic improvement in final accuracy, but we can observe that it could yield not only lower parameter divergence but also faster convergence speed (especially before the first learning rate drop). One can find the corresponding complete results in Figure 25.

## C.5 THE EFFECT OF BATCH NORMALIZATION

Here we investigate the effect of Batch Normalization. For its implementations into the considered federated learning algorithm, we let the server get the proper moving variance by $\frac{1}{K}\sum_{k \in \mathcal{K}} \left(\mathbb{E}\left[\phi^2\right]\right)_k - \mathbb{E}\left[\phi\right]^2$ at each round, by allowing each learner $k$ collect $\left(\mathbb{E}\left[\phi^2\right]\right)_k$ as well

Table 11: Test accuracy (%) comparison of **NetA-Baseline** with/without Batch Normalization/Renormalization. Values outside/inside brackets denote one measured after the whole training and the highest one during the rounds, respectively. BN: Batch Normalization; BRN: Batch Renormalization.

| Dataset | Method | Vanilla Training | | Federated Learning | |
|---|---|---|---|---|---|
| | | BS: 50 | BS: 500 | IID | Non-IID(2) |
| CIFAR-10 | pure SGD | 82.47 (82.72) | 78.97 (79.06) | 81.12 (81.36) | 68.92 (69.57) |
| CIFAR-10 | pure SGD + BN | 85.06 (85.17) | 78.33 (78.70) | 82.91 (83.02) | **57.56 (58.68)** |
| CIFAR-10 | pure SGD + BRN | 85.67 (85.72) | 81.49 (81.65) | 84.00 (84.19) | **66.19 (66.32)** |
| CIFAR-10 | PMom-A | - | - | 81.78 (81.83) | 74.82 (75.06) |
| CIFAR-10 | PMom-A + BN | - | - | 83.74 (83.84) | **52.76 (59.69)** |
| CIFAR-10 | NMom-A | 83.89 (83.90) | 79.39 (79.49) | 81.49 (81.82) | 74.11 (74.23) |
| CIFAR-10 | NMom-A + BN | 85.22 (85.25) | 79.58 (80.14) | 83.54 (83.80) | **50.46 (59.31)** |
| CIFAR-10 | NMom-A + BRN | 86.36 (86.43) | 82.03 (82.46) | 84.24 (84.70) | **70.32 (70.38)** |
| SVHN | NMom-A | - | - | 93.02 (93.26) | 89.28 (89.70) |
| SVHN | NMom-A + BN | - | - | 92.96 (93.34) | **74.50 (79.73)** |
| CIFAR-10 | Adam-A | 81.97 (82.35) | 80.73 (80.76) | 81.18 (81.45) | 75.32 (75.45) |
| CIFAR-10 | Adam-A + BN | 85.76 (85.93) | 83.84 (83.93) | 84.88 (85.17) | **39.31 (47.67)** |
| CIFAR-10 | Adam-A + BRN | 85.86 (86.11) | 83.18 (83.36) | 84.68 (84.91) | **72.49 (72.71)** |

as the existing moving statistics of Batch Normalization ($\phi$ denotes activations). This can be regarded as a federated version of distributed Batch Normazliation methods (e.g., Qin et al. (2018); Zhang et al. (2018)). It is natural to take this strategy especially under the non-IID data setting; otherwise, a huge problem would arise due to bad approximation of the moving statistics. Also, it is additionally remarked that for Batch Renormalization we simply used $\alpha = 0.01$, $r_{max} = 2$, and $d_{max} = 2$ in the experiments (see Ioffe (2017) for the description of the three hyperparameters).

It is seen from Table 11 that under the non-IID data setting, the performance significantly gets worse if Batch Normalization is employed to the baseline; this would be rooted in that the dependence of batch-normalized hidden activations makes each learner's update too overfitted to the distribution of their local training data. The consequent bigger parameter divergence is observed in Figures 26, 27, and 28. On the contrary, Batch Renormalization, by relaxing the dependence, yields a better outcome; although its parameter divergence is seen greater in some layers than under Batch Normalization, it does not lead to the steep fall phenomenon while the Batch Normalization does in $F_3$. Nevertheless, the Batch Renormalization was still not able to exceed the performance of the baseline due to the significant parameter divergence.[12]

## C.6 THE EFFECT OF DATA AUGMENTATION

In the implementation of data augmentation, we used random horizontal flipping, brightness & contrast adjustment, and $24 \times 24$ cropping & resizing in the pipeline. From Table 12, we identify that under the non-IID data setting, the data augmentation yields diminishing returns for the PMom-A, NMom-A, and Adam-A cases on CIFAR-10, compared to under the IID data setting; under Adam-A, especially it gives even a worse outcome. However, it is seen that the corresponding parameter divergence is almost similar between with and without the data augmentation (refer to Figures 30 and 31). Instead, we are able to notice that the diminishing outcomes from the data augmentation had been eventually rooted in local updates' high training losses. Here we note that in the pure SGD case, very high training loss values are found as well under the IID data setting when the data augmentation was applied (see Figure 29); this leads to lower test accuracy compared to the baseline,

---

[12]During the experiments, we observed that Batch Renormalization yields relatively heavy fluctuation of test accuracy and training loss until the first learning rate drop. This is thought to have the effect of setting $r_{max}$ and $d_{max}$ to the fixed value, as discussed in Ioffe (2017); however we guess that the effect of the data non-IIDness might be greater, judging from the degree of the fluctuation under the IID and the non-IID data setting.

Table 12: Test accuracy (%) comparison of NetA-Baseline with/without data augmentation. Values outside/inside brackets denote one measured after the whole training and the highest one during the rounds, respectively. DA: data augmentation.

| Dataset | Method | Vanilla Training | | Federated Learning | |
|---|---|---|---|---|---|
| | | BS: 50 | BS: 500 | IID | Non-IID(2) |
| CIFAR-10 | pure SGD | 82.47 (82.72) | 78.97 (79.06) | 81.12 (81.36) | 68.92 (69.57) |
| CIFAR-10 | pure SGD + DA | 86.79 (86.97) | 77.66 (78.03) | 80.39 (80.48) | 68.33 (68.69) |
| CIFAR-10 | PMom-A | - | - | 81.78 (81.83) | 74.82 (75.06) |
| CIFAR-10 | PMom-A + DA | - | - | 83.87 (83.93) | **72.40 (74.01)** |
| CIFAR-10 | NMom-A | 83.89 (83.90) | 79.39 (79.49) | 81.49 (81.82) | 74.11 (74.23) |
| CIFAR-10 | NMom-A + DA | 87.16 (87.41) | 83.07 (83.35) | 84.85 (84.89) | **73.95 (74.46)** |
| SVHN | NMom-A | - | - | 93.02 (93.26) | 89.28 (89.70) |
| SVHN | NMom-A + DA | - | - | 94.51 (94.66) | 90.61 (91.59) |
| CIFAR-10 | Adam-A | 81.97 (82.35) | 80.73 (80.76) | 81.18 (81.45) | 75.32 (75.45) |
| CIFAR-10 | Adam-A + DA | 83.27 (83.54) | 84.84 (84.96) | 82.63 (82.99) | **72.79 (73.67)** |

Table 13: Test accuracy (%) comparison of NetA-Baseline with/without Dropout. Values outside/inside brackets denote one measured after the whole training and the highest one during the rounds, respectively. DO: Dropout.

| Dataset | Method | Vanilla Training | | Federated Learning | |
|---|---|---|---|---|---|
| | | BS: 50 | BS: 500 | IID | Non-IID(2) |
| CIFAR-10 | pure SGD | 82.47 (82.72) | 78.97 (79.06) | 81.12 (81.36) | 68.92 (69.57) |
| CIFAR-10 | pure SGD + DO | 85.76 (85.79) | 82.48 (82.53) | 83.40 (83.52) | 73.02 (73.04) |
| CIFAR-10 | PMom-A | - | - | 81.78 (81.83) | 74.82 (75.06) |
| CIFAR-10 | PMom-A + DO | - | - | 84.29 (84.38) | **76.83 (76.88)** |
| CIFAR-10 | NMom-A | 83.89 (83.90) | 79.39 (79.49) | 81.49 (81.82) | 74.11 (74.23) |
| CIFAR-10 | NMom-A + DO | 86.55 (86.63) | 84.29 (84.32) | 84.34 (84.53) | **75.80 (75.89)** |
| SVHN | NMom-A | - | - | 93.02 (93.26) | 89.28 (89.70) |
| SVHN | NMom-A + DO | - | - | 94.14 (94.36) | 90.61 (91.45) |
| CIFAR-10 | Adam-A | 81.97 (82.35) | 80.73 (80.76) | 81.18 (81.45) | 75.32 (75.45) |
| CIFAR-10 | Adam-A + DO | 84.27 (84.46) | 83.09 (83.20) | 82.01 (82.11) | 76.63 (76.68) |

similar to under the non-IID cases. Also, it is additionally noted that unlike on the CIFAR-10, in the experiments on SVHN it was observed that the generalization effect of the data augmentation is still valid in test accuracy.

## C.7 THE EFFECT OF DROPOUT

In the experiments, we employed Dropout with the rates 0.2 and 0.5 for convolutional layers and fully-connected layers, respectively. The results show that under the non-IID data setting, the Dropout provides greater parameter divergence compared to the baselines, especially in $F_3$ (see Figures 32, 33, and 34); this leads to diminishing returns for the PMom-A and NMom-A cases on CIFAR-10, compared to under the IID data setting. However, we can observe from Table 13 that the effect of the Dropout is still maintained positive for the rest of the cases.

## C.8 THE EXPERIMENT RESULTS UNDER UNBALANCED DATA SETTINGS

As remarked in (McMahan et al., 2017), since the federated learning do not require centralizing local data, data unbalancedness (i.e., each learner has various numbers of local data examples) would be also naturally assumed in the federated learning along with non-IIDness. In relation, we also conducted the experiments under the unbalanced cases. Table 14 summarizes the considered unbalanced data settings; they were constructed similarly to (Li et al., 2019b) so that the number of data examples per learner follows a power law.

The experimental results under the unbalanced settings are summarized in Table 15. From the table, it is observed that our findings in Section 4.1 are still valid under the unbalanced data settings. In addition, we can also see that for the unbalanced cases, the performance under Non-IID(2) setting is worse mostly than that of balanced cases while they show similar values under the IID data setting; this indicates that the negative impact of data unbalancedness is not as great as that of the non-IIDness, but it becomes much bigger when the two are combined.

Table 14: Configuration of unbalanced data settings. #Cls/L: the number of classes in each learner's local training dataset; #Exs/L: the number of training examples in each local dataset.

| Data Setting | #Learners | #Cls/L | #Exs/L | |
| --- | --- | --- | --- | --- |
| | | | Mean | Std. |
| IID | 10 | 10 | 5000 | 2905 |
| Non-IID(2) | 10 | 2 | 5000 | 2905 |

Table 15: Test accuracy (%) comparison on CIFAR-10 under the balanced and the unbalanced data settings. Values outside/inside brackets denote one measured after the whole training and the highest one during the rounds, respectively.

| Network | Method | Balanced Data Setting | | Unbalanced Data Setting | |
| --- | --- | --- | --- | --- | --- |
| | | IID | Non-IID(2) | IID | Non-IID(2) |
| NetA-Baseline | Adam-WB | 80.99 (81.00) | **67.51 (67.82)** | 80.03 (80.27) | **56.52 (59.62)** |
| NetA-Baseline | Adam-A | 81.18 (81.45) | 75.32 (75.45) | 81.02 (81.14) | 73.12 (73.19) |
| NetA-Baseline | NMom-A | 81.49 (81.82) | 74.11 (74.23) | 81.31 (81.49) | 72.59 (72.86) |
| NetA-Deepest | NMom-A | 83.12 (83.58) | **68.98 (69.64)** | 83.67 (83.68) | **69.10 (69.23)** |
| NetC-Baseline | NMom-A | 80.60 (80.75) | **64.06 (64.55)** | 81.00 (81.10) | **62.23 (62.30)** |
| NetC-Widest | NMom-A | 80.90 (81.14) | 73.64 (73.91) | 80.67 (80.76) | 71.84 (72.34) |
| NetA-Baseline | NMom-A | 81.49 (81.82) | 74.11 (74.23) | 81.31 (81.49) | 72.59 (72.86) |
| NetA-Baseline | NMom-A + WD: 0.0005 | 82.69 (83.14) | **54.11 (54.15)** | 82.76 (83.13) | **47.85 (47.85)** |
| NetA-Baseline | NMom-A + BN | 83.54 (83.80) | **50.46 (59.31)** | 83.38 (83.54) | **45.92 (50.95)** |
| NetA-Baseline | NMom-A + DA | 84.85 (84.89) | **73.95 (74.46)** | 84.16 (84.19) | **73.60 (74.06)** |
| NetA-Baseline | NMom-A + DO | 84.34 (84.53) | **75.80 (75.89)** | 84.25 (84.38) | **68.83 (70.33)** |

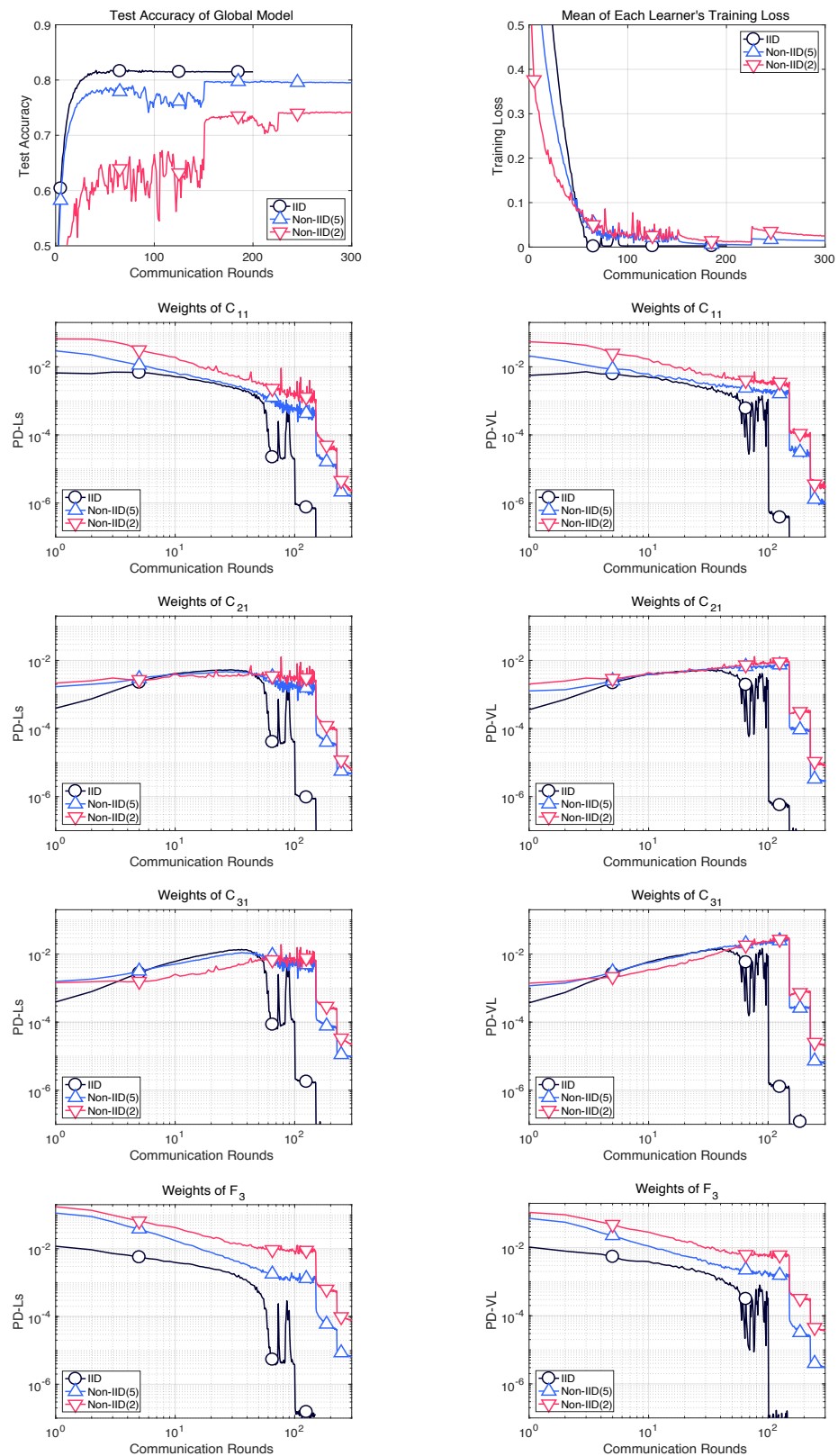

Figure 9: Behavior comparison with respect to the degree of data non-IIDness in the training of NetA-Baseline on CIFAR-10 (Optimizer: NMom-A).

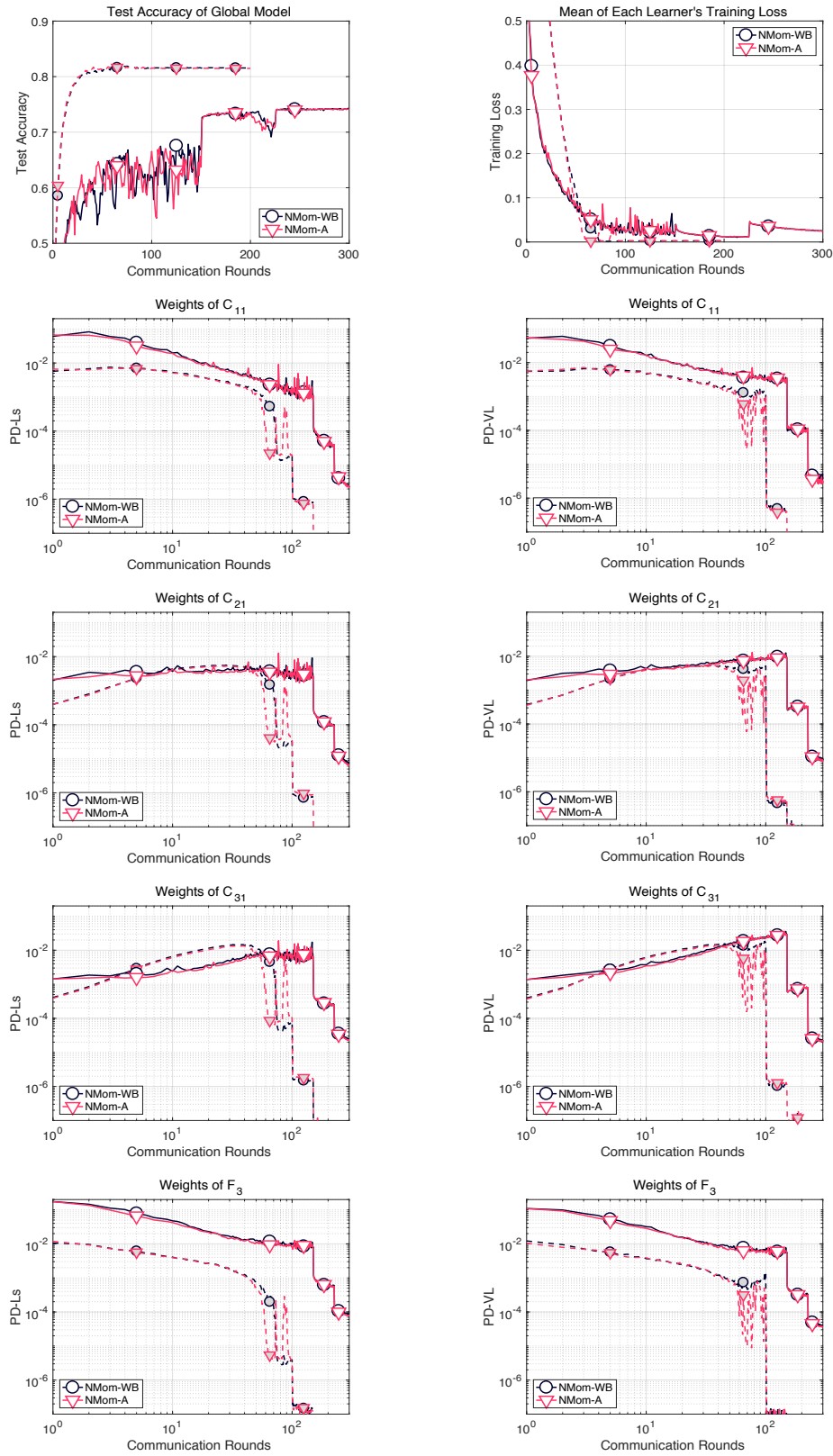

Figure 10: Behavior comparison between NMom-WB and Nmom-A in the training of NetA-Baseline on CIFAR-10. Dotted and solid lines indicate the results under IID and Non-IID(2) setting, respectively.

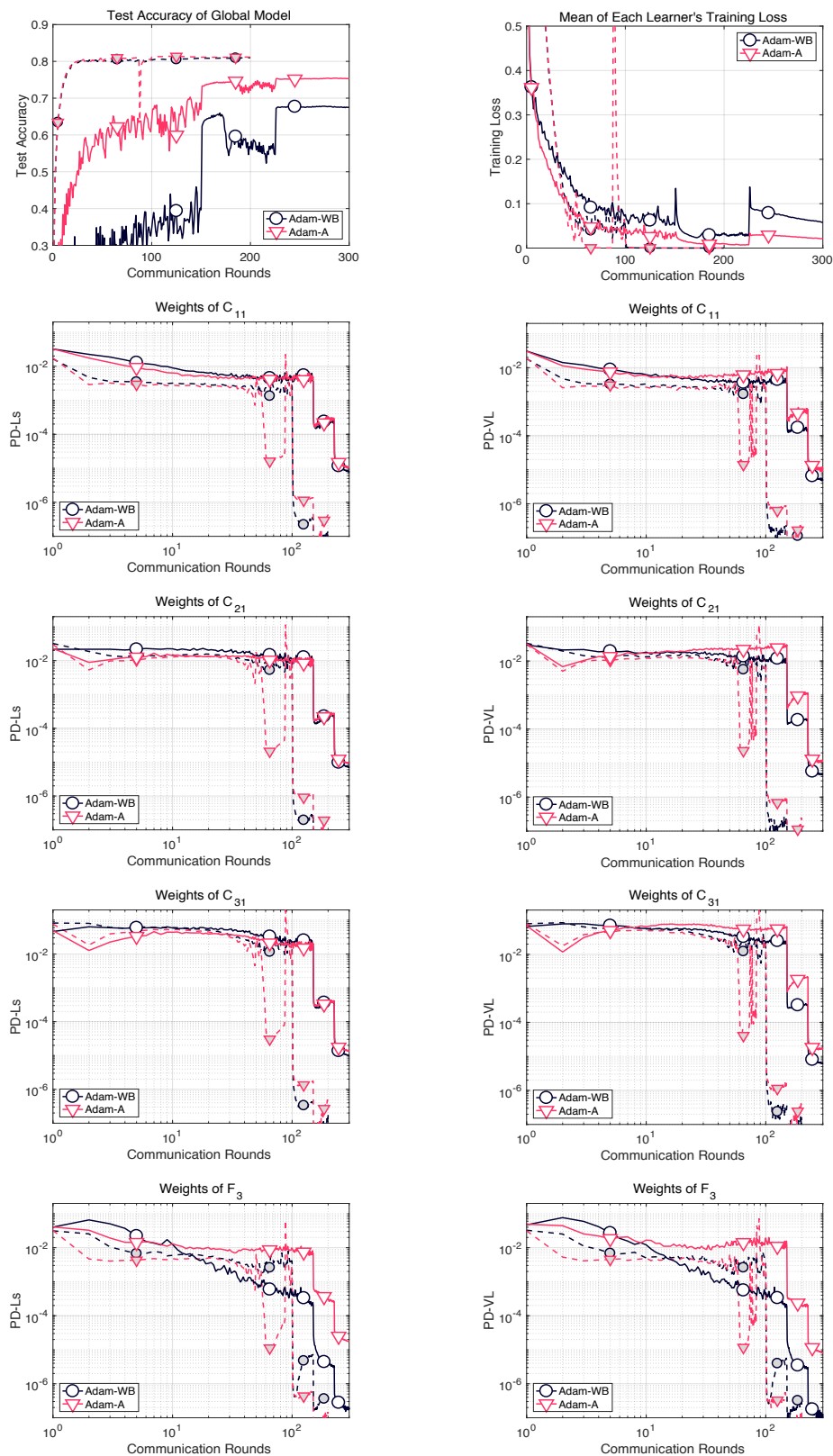

Figure 11: Behavior comparison between Adam-WB and Adam-A in the training of NetA-Baseline on CIFAR-10. Dotted and solid lines indicate the results under IID and Non-IID(2) setting, respectively.

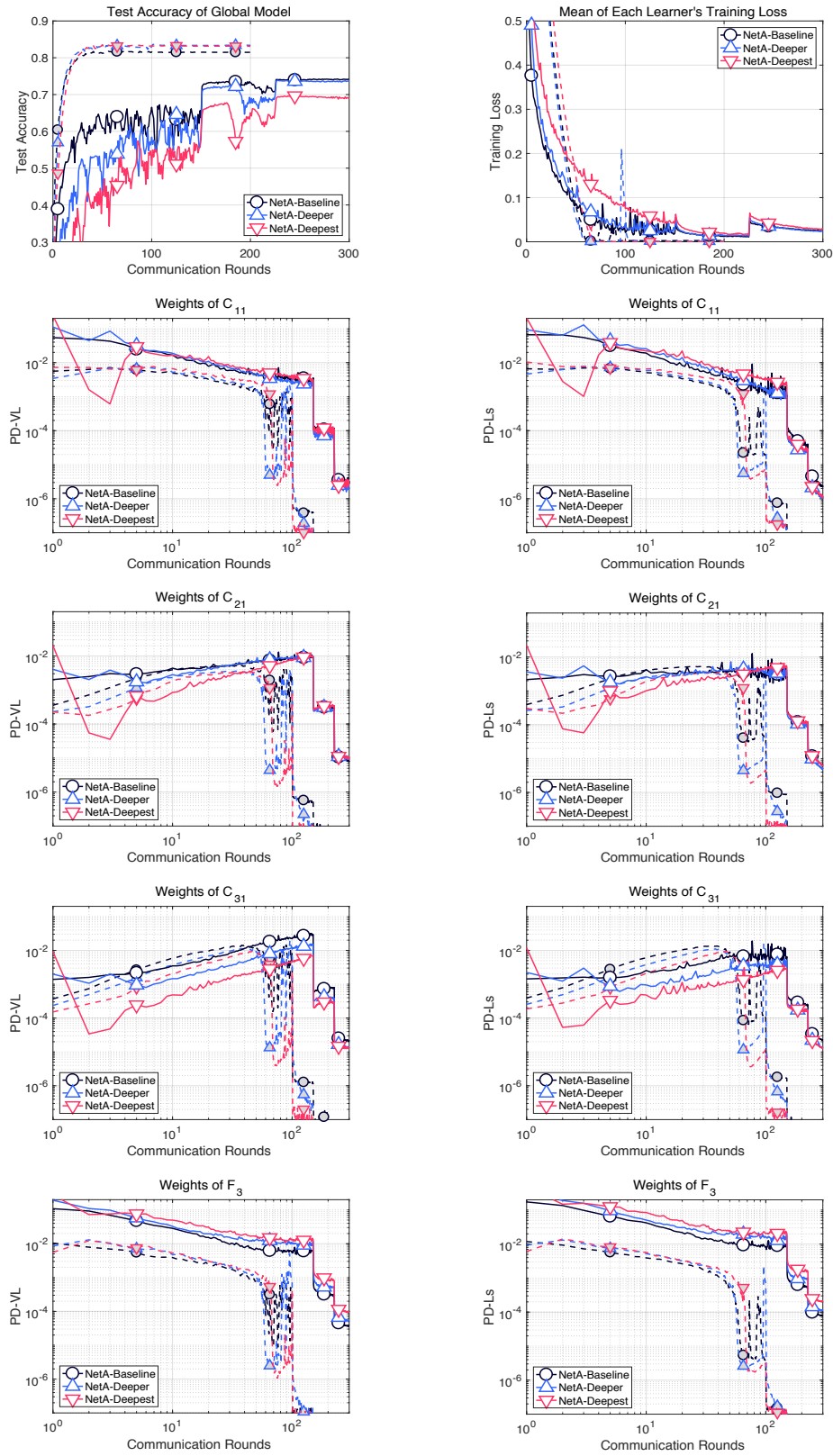

Figure 12: Behavior comparison of the NetA models with respect to network depth on CIFAR-10 (Optimizer: NMom-A). Dotted and solid lines indicate the results under IID and Non-IID(2) setting, respectively.

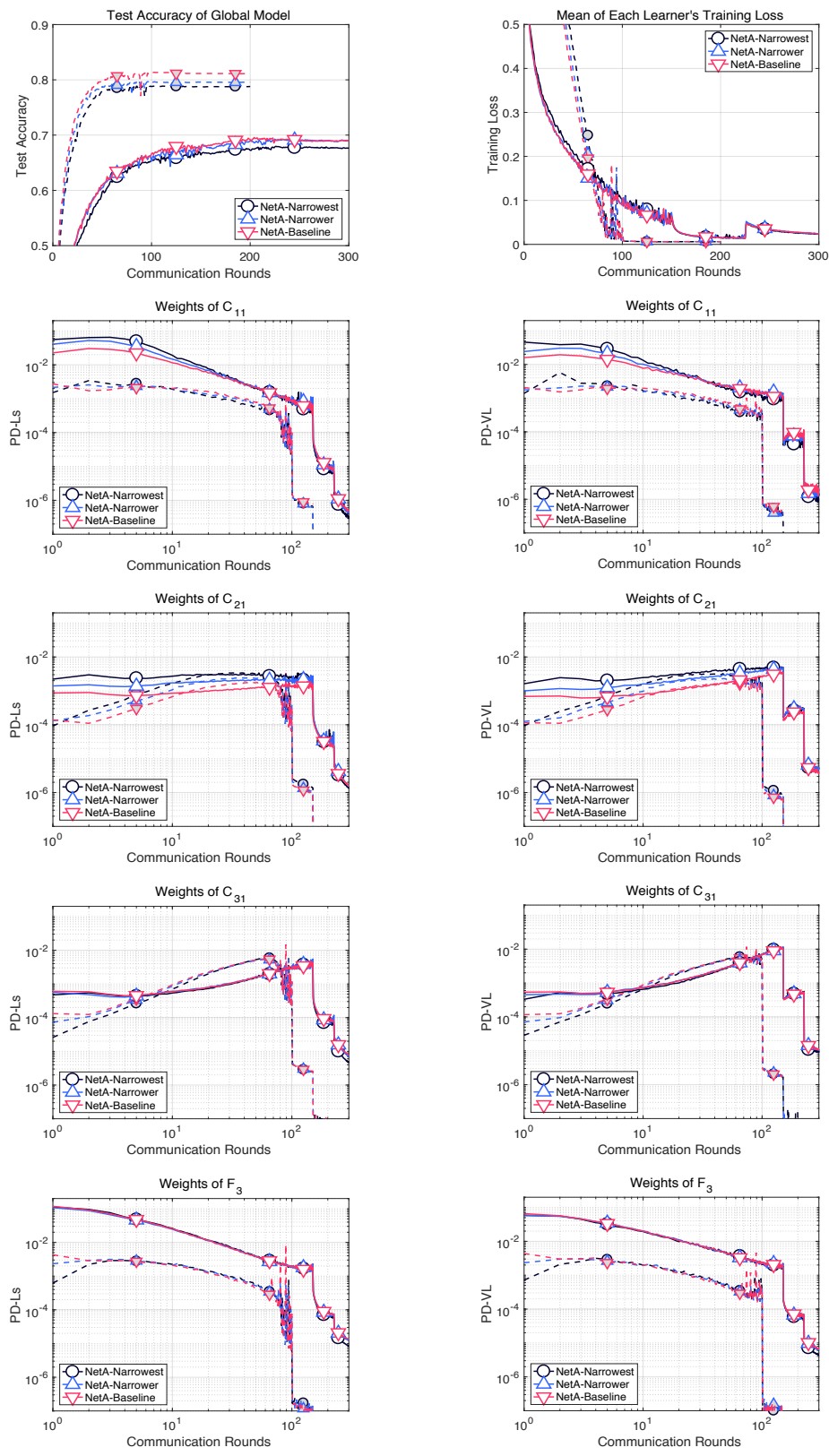

Figure 13: Behavior comparison of the NetA models with respect to network width on CIFAR-10 (Optimizer: pure SGD). Dotted and solid lines indicate the results under IID and Non-IID(2) setting, respectively.

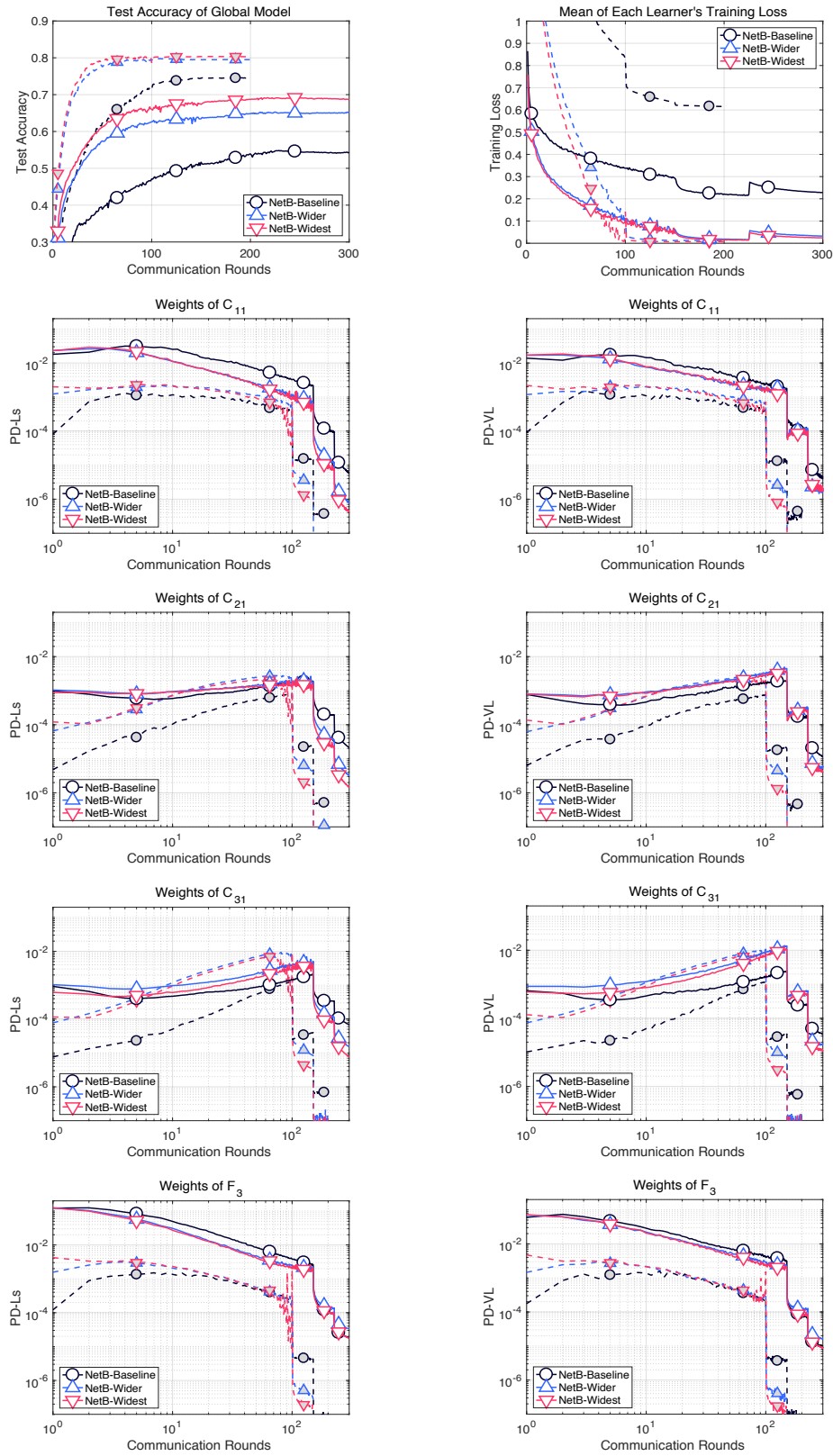

Figure 14: Behavior comparison of the NetB models with respect to network width on CIFAR-10 (Optimizer: pure SGD). Dotted and solid lines indicate the results under IID and Non-IID(2) setting, respectively.

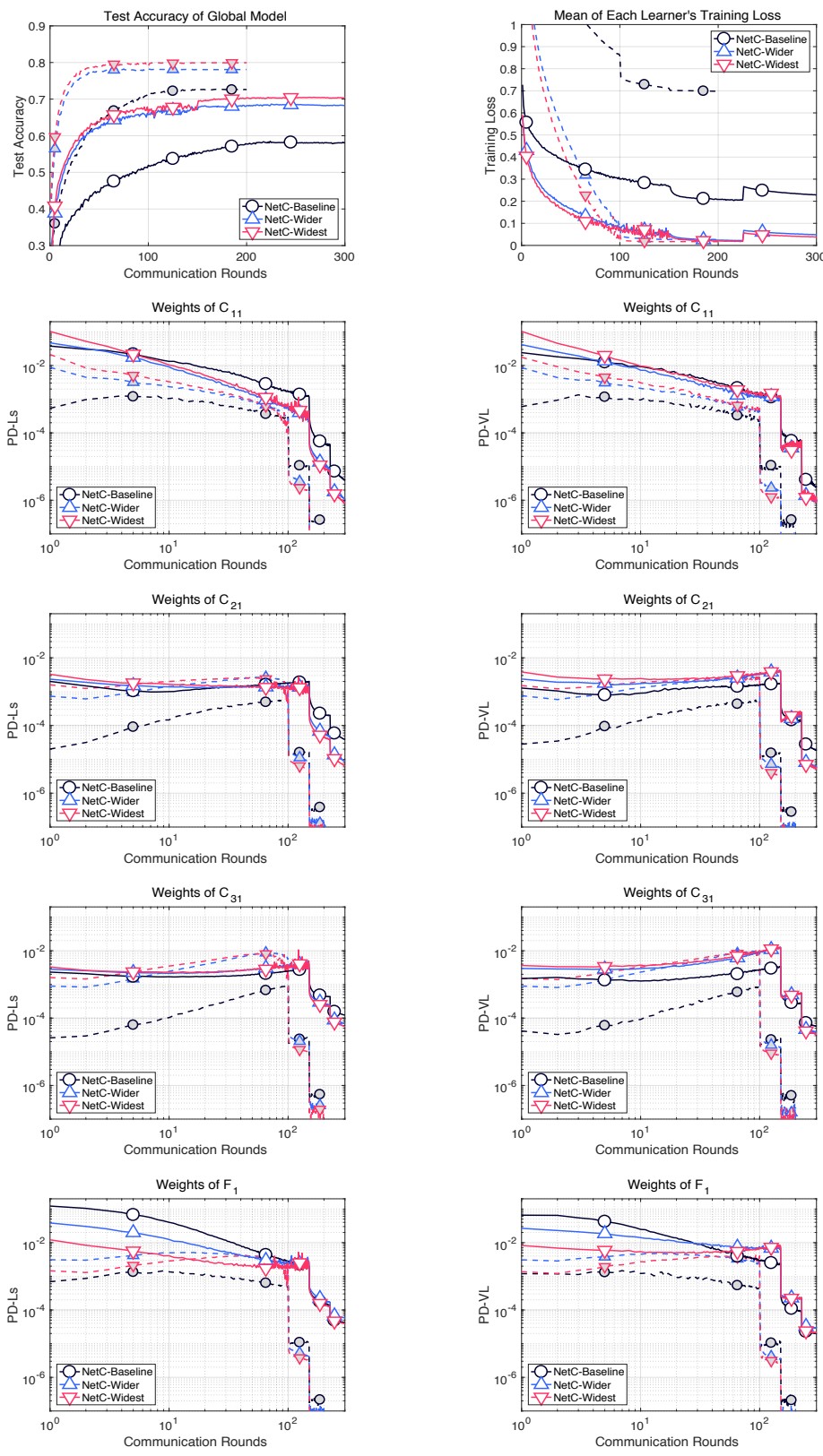

Figure 15: Behavior comparison of the NetC models with respect to network width on CIFAR-10 (Optimizer: pure SGD). Dotted and solid lines indicate the results under IID and Non-IID(2) setting, respectively.

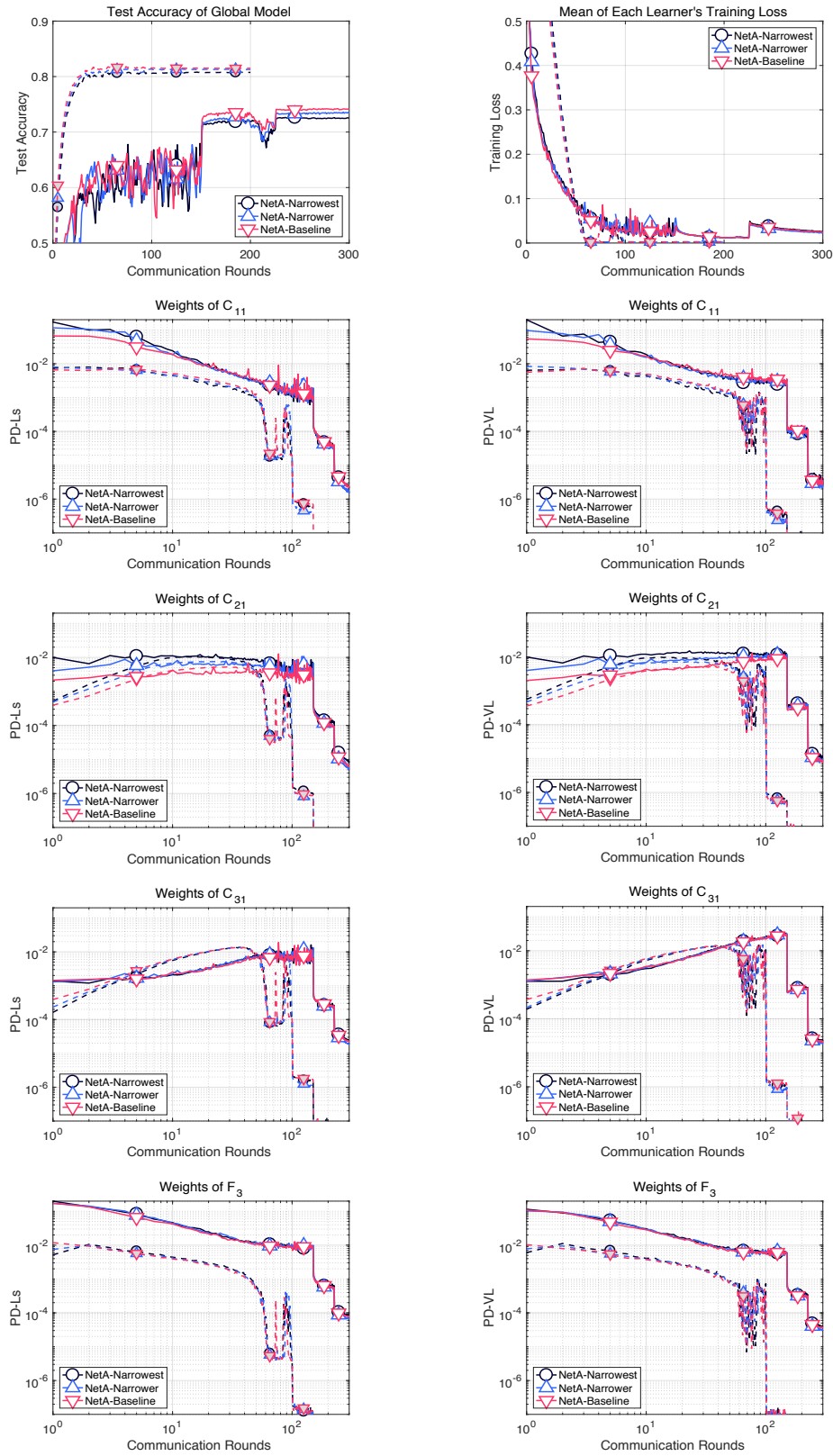

Figure 16: Behavior comparison of the NetA models with respect to network width on CIFAR-10 (Optimizer: NMom-A). Dotted and solid lines indicate the results under IID and Non-IID(2) setting, respectively.

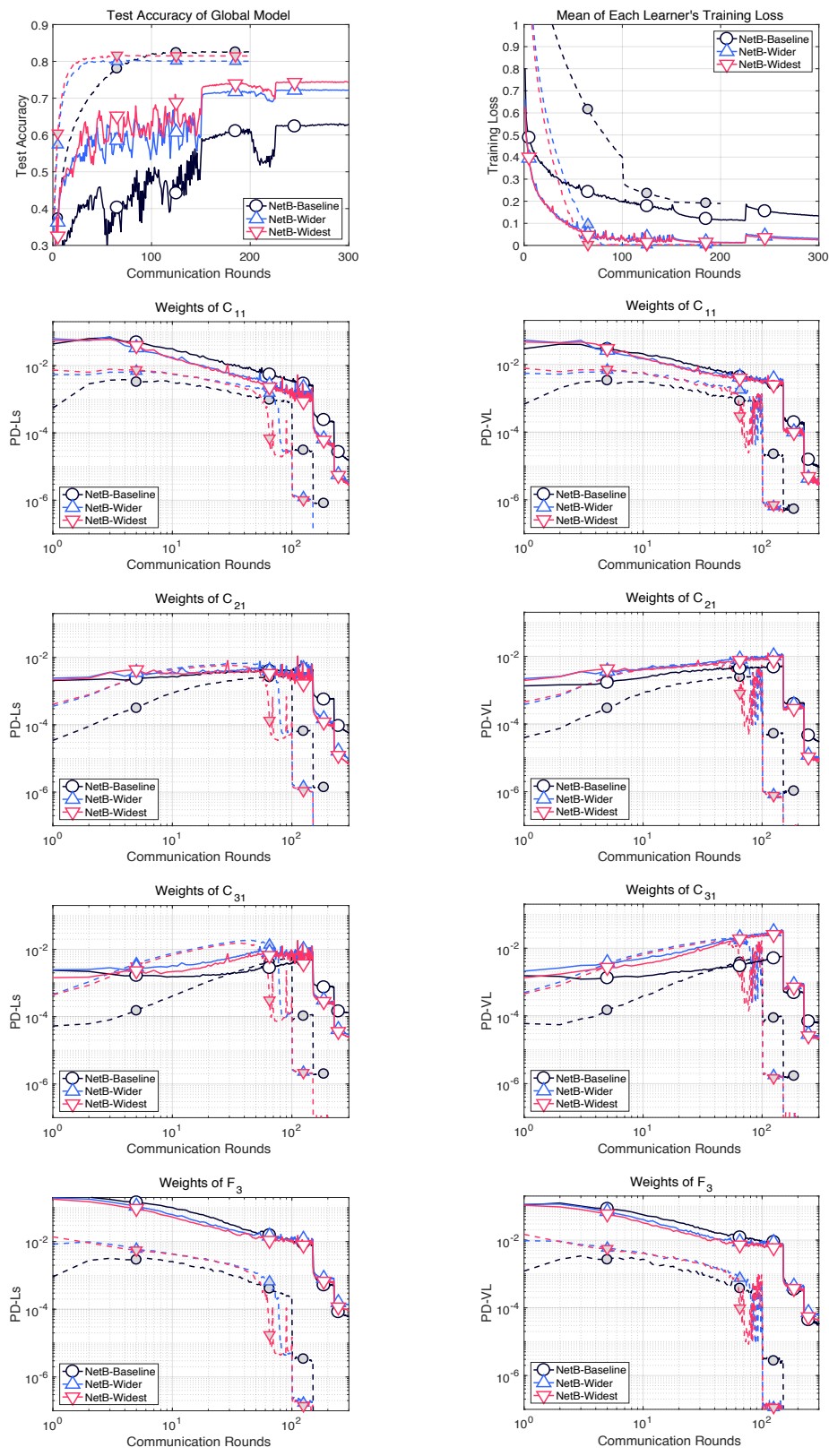

Figure 17: Behavior comparison of the NetB models with respect to network width on CIFAR-10 (Optimizer: NMom-A). Dotted and solid lines indicate the results under IID and Non-IID(2) setting, respectively.

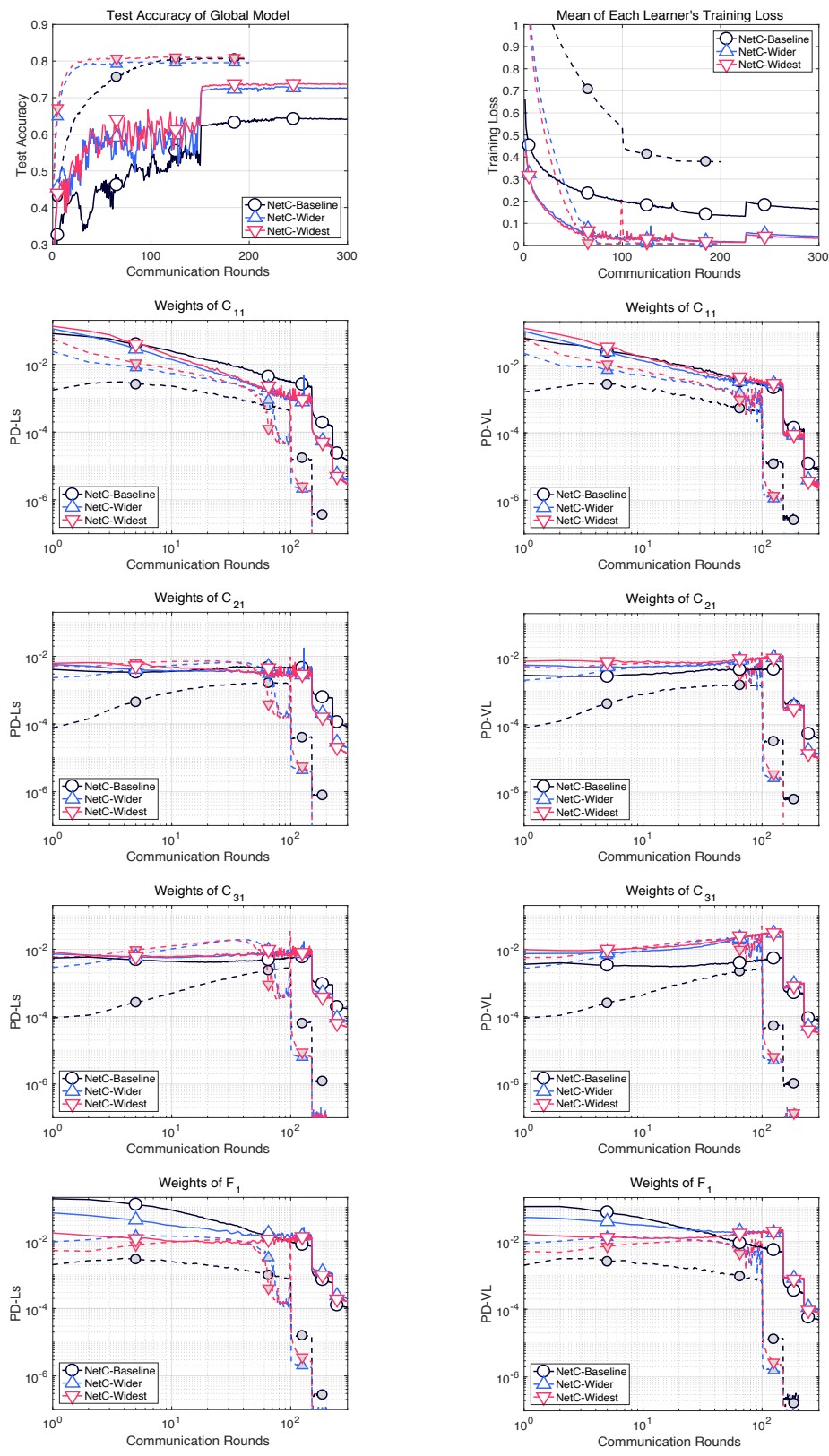

Figure 18: Behavior comparison of the NetC models with respect to network width on CIFAR-10 (Optimizer: NMom-A). Dotted and solid lines indicate the results under IID and Non-IID(2) setting, respectively.

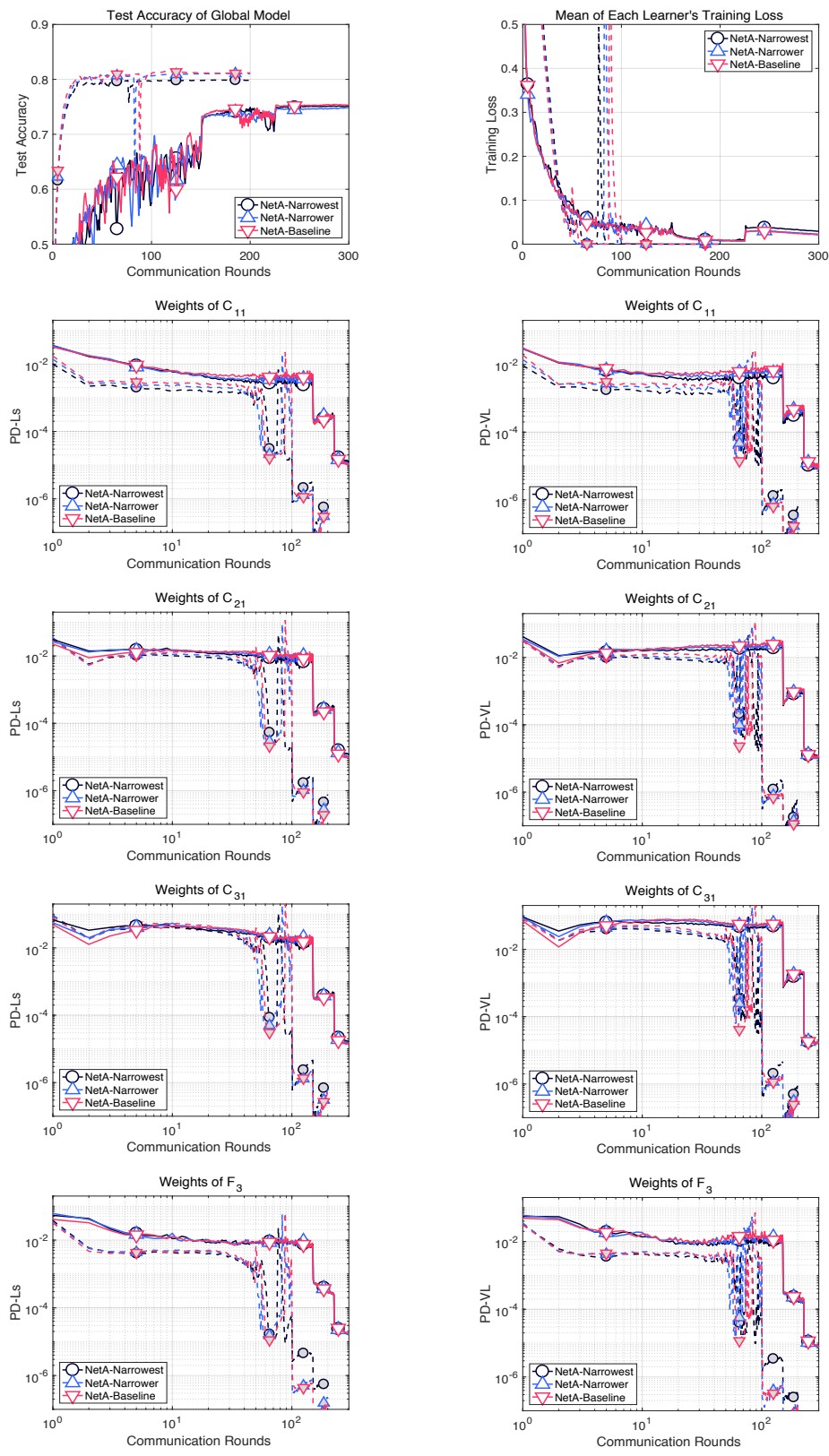

Figure 19: Behavior comparison of the NetA models with respect to network width on CIFAR-10 (Optimizer: Adam-A). Dotted and solid lines indicate the results under IID and Non-IID(2) setting, respectively.

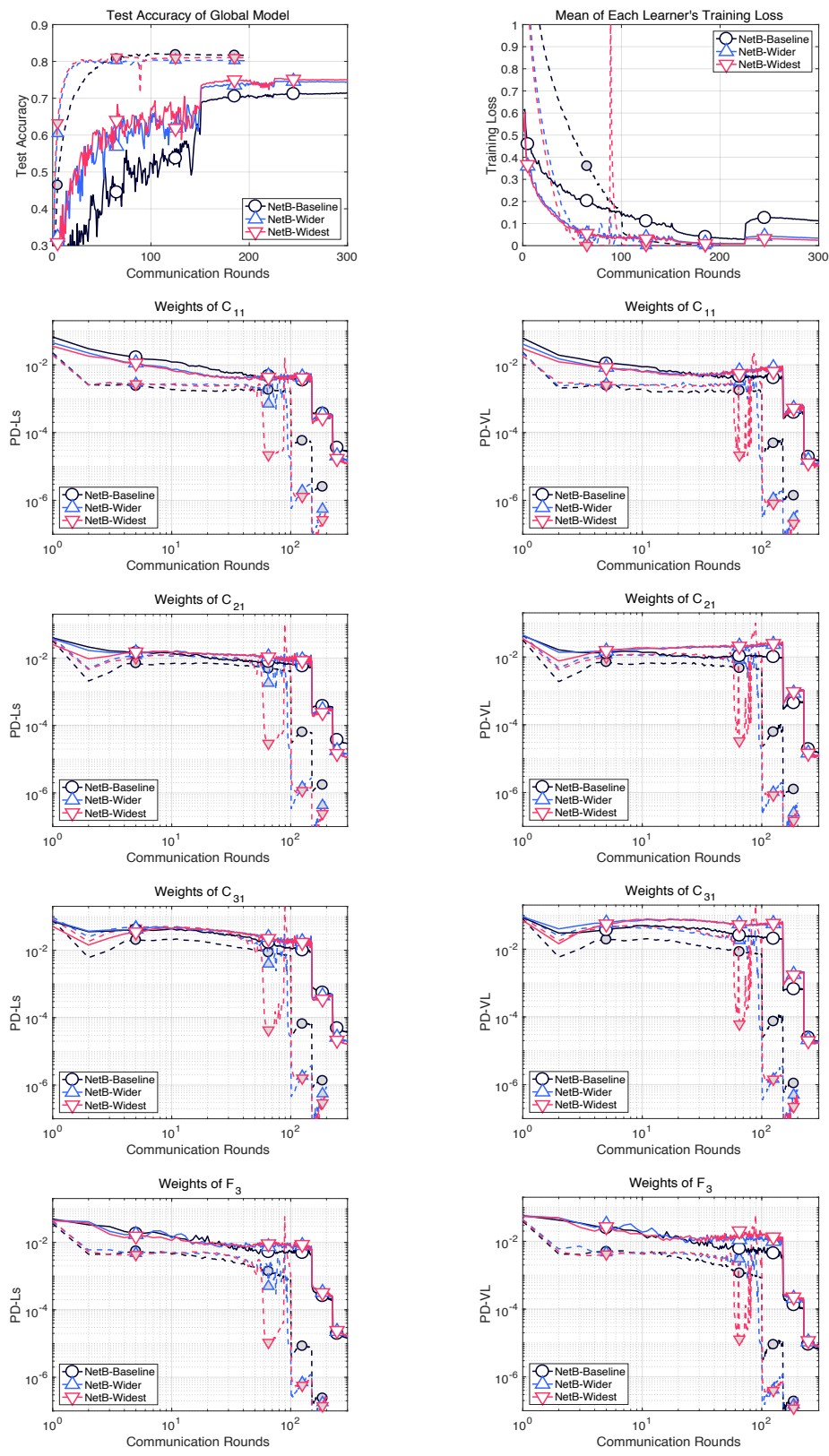

Figure 20: Behavior comparison of the NetB models with respect to network width on CIFAR-10 (Optimizer: Adam-A). Dotted and solid lines indicate the results under IID and Non-IID(2) setting, respectively.

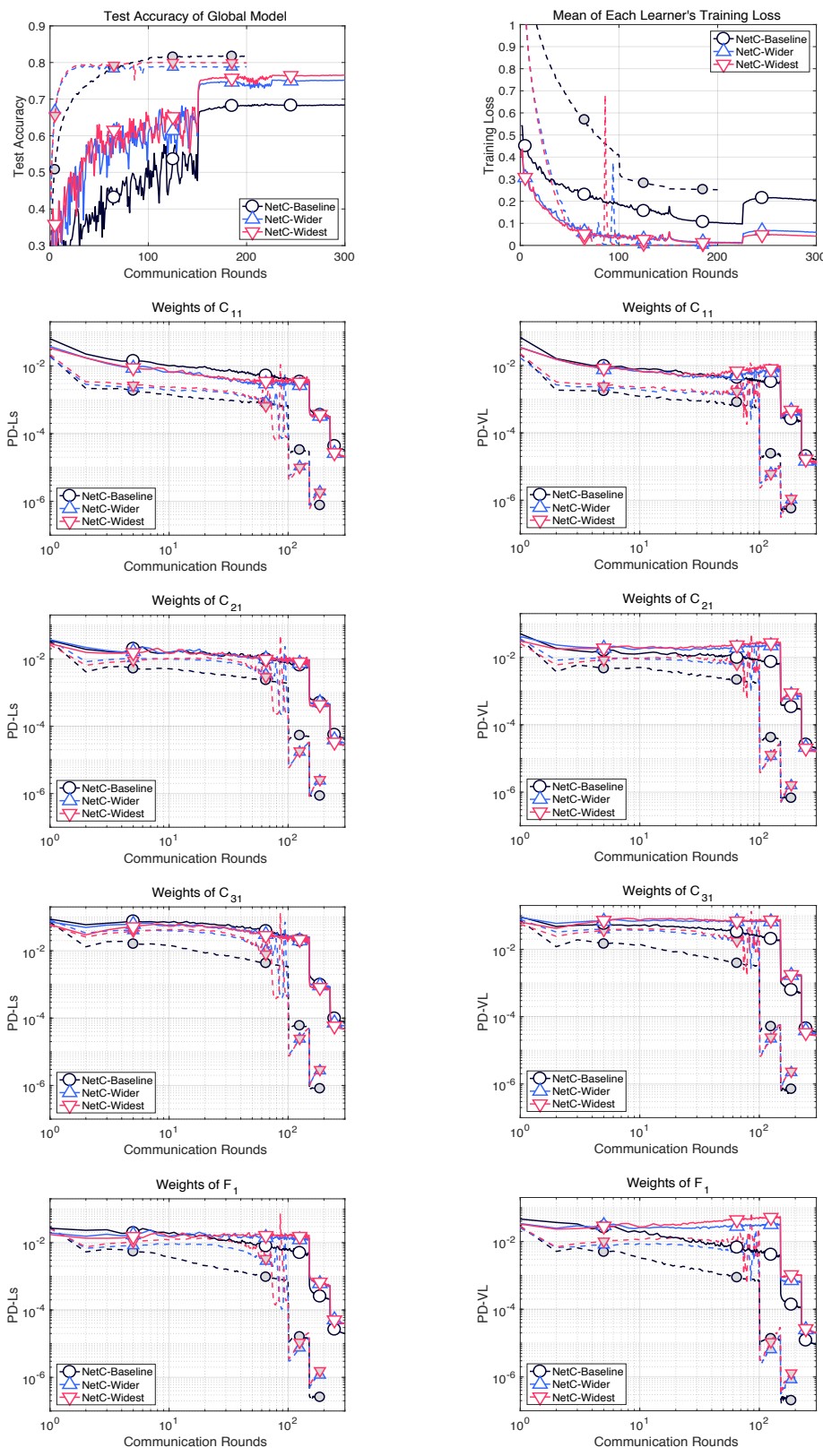

Figure 21: Behavior comparison of the NetC models with respect to network width on CIFAR-10 (Optimizer: Adam-A). Dotted and solid lines indicate the results under IID and Non-IID(2) setting, respectively.

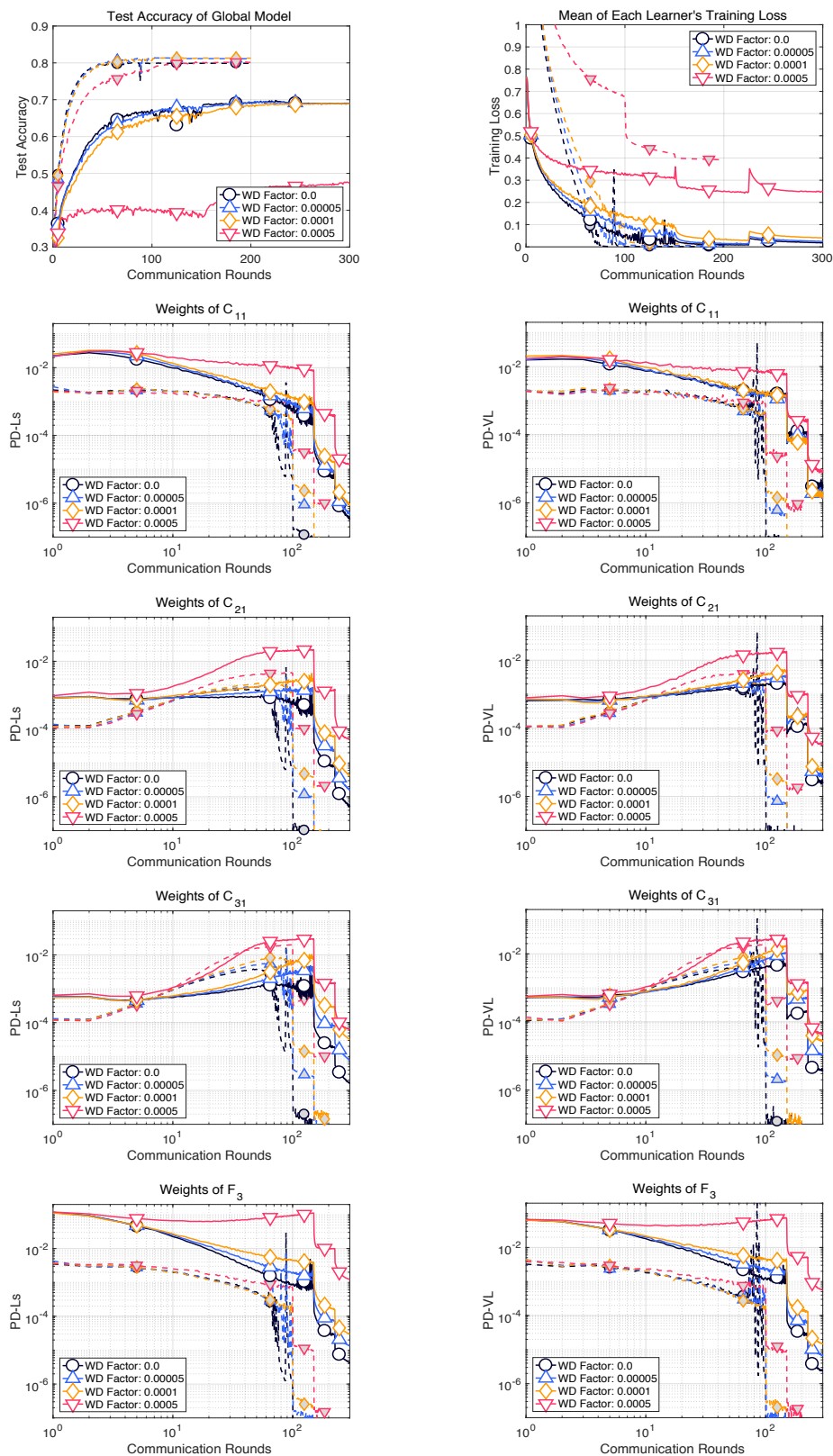

Figure 22: Behavior comparison with respect to weight decay levels in the training of NetA-Baseline on CIFAR-10 (Optimizer: pure SGD). Dotted and solid lines indicate the results under IID and Non-IID(2) setting, respectively.

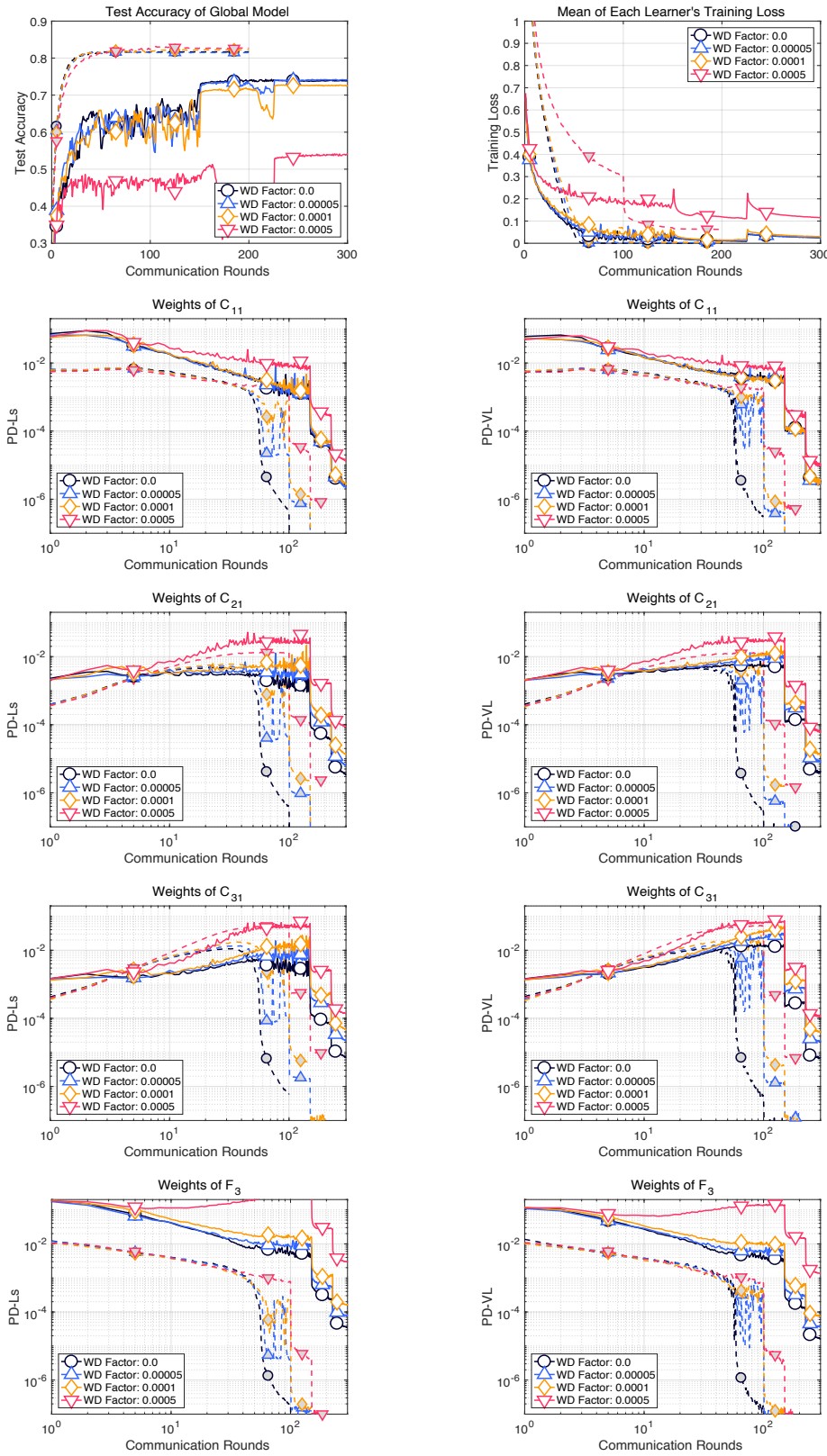

Figure 23: Behavior comparison with respect to weight decay levels in the training of NetA-Baseline on CIFAR-10 (Optimizer: NMom-A). Dotted and solid lines indicate the results under IID and Non-IID(2) setting, respectively.

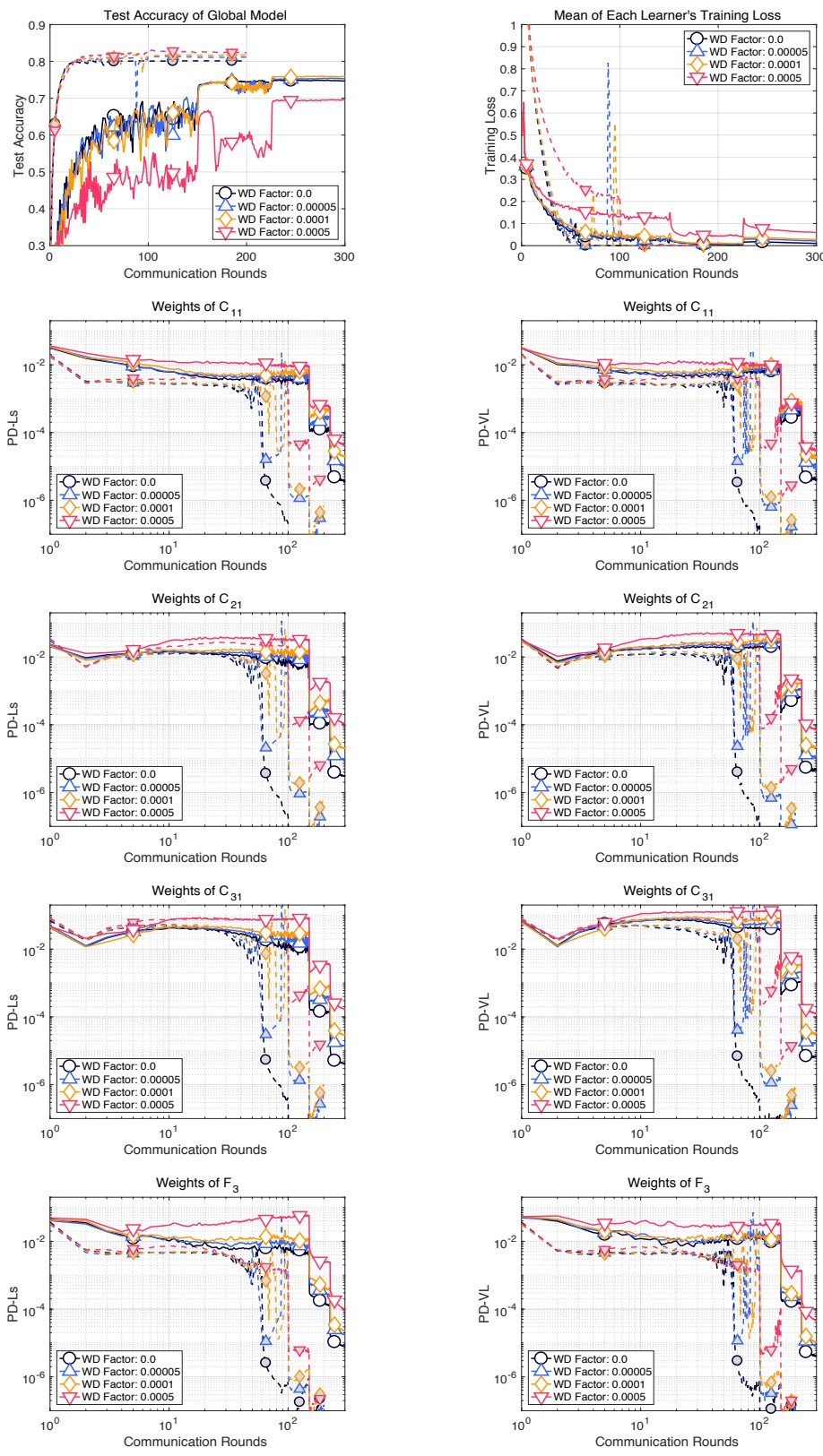

Figure 24: Behavior comparison with respect to weight decay levels in the training of NetA-Baseline on CIFAR-10 (Optimizer: Adam-A). Dotted and solid lines indicate the results under IID and Non-IID(2) setting, respectively.

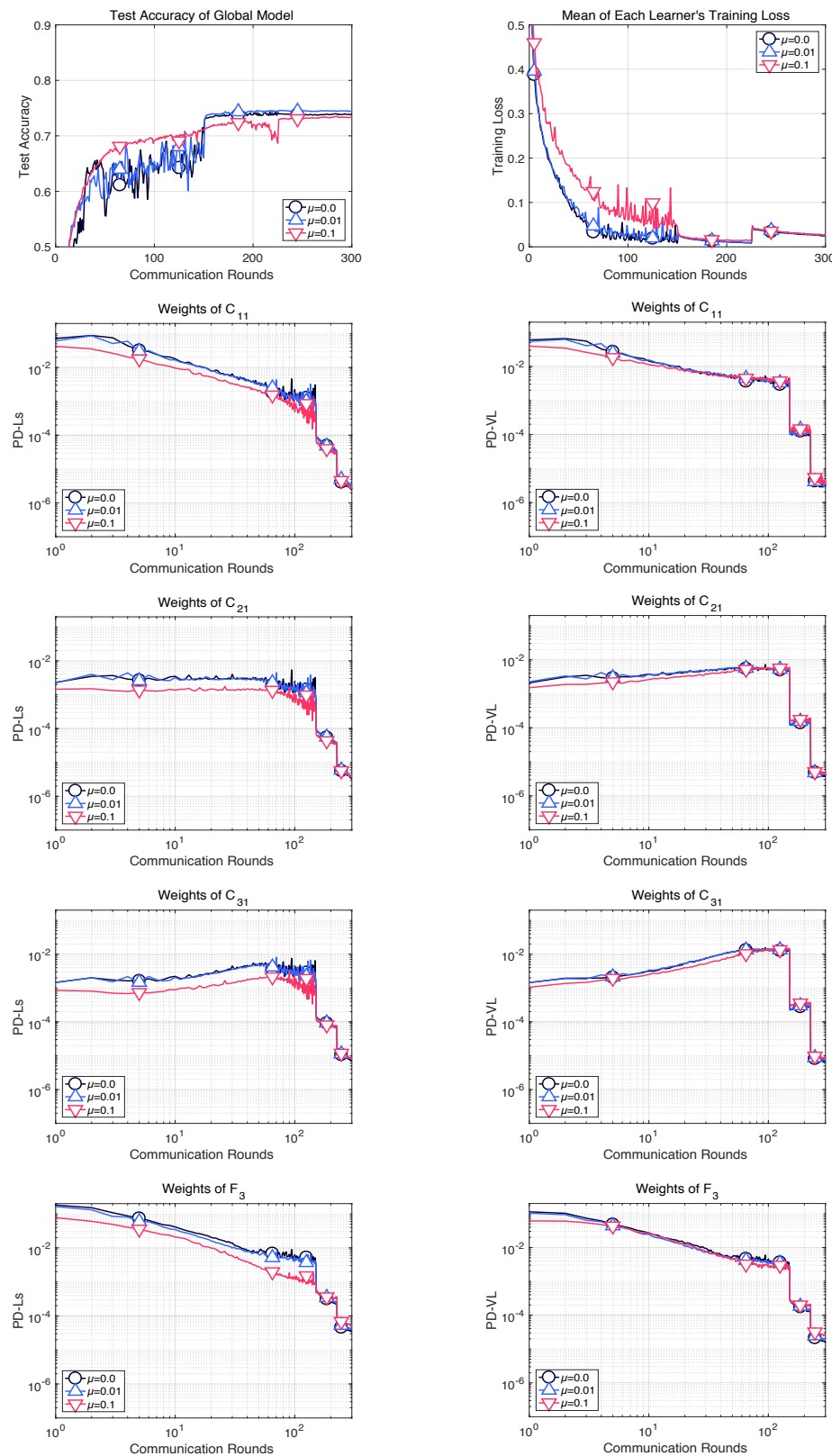

Figure 25: Behavior comparison under Non-IID(2) setting with respect to FedProx factor $\mu$ in the training of NetA-Baseline on CIFAR-10 (Optimizer: NMom-A).

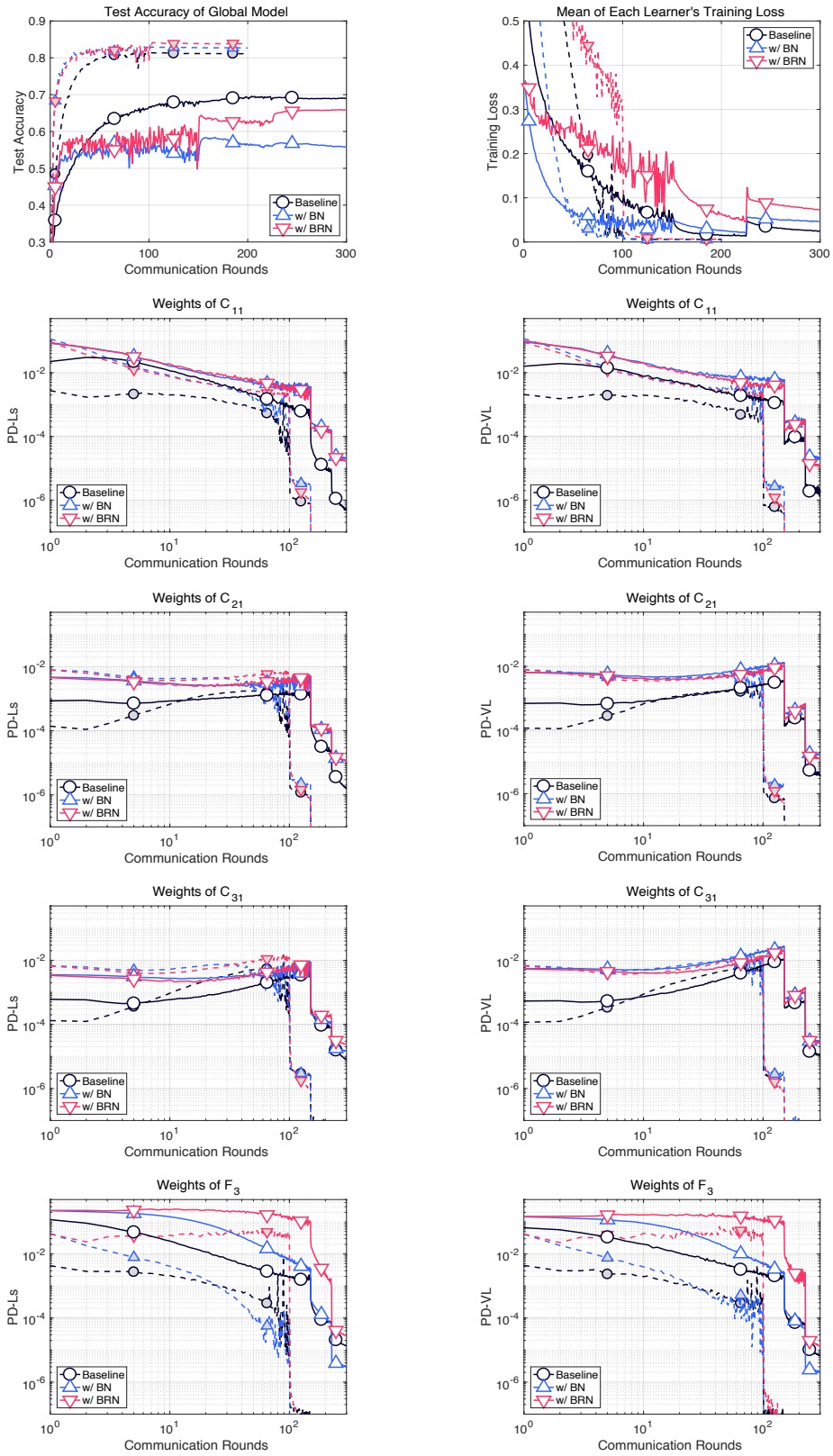

Figure 26: Behavior comparison with/without Batch Normalization/Renormalization in the training of NetA-Baseline on CIFAR-10 (Optimizer: pure SGD). Dotted and solid lines indicate the results under IID and Non-IID(2) setting, respectively.

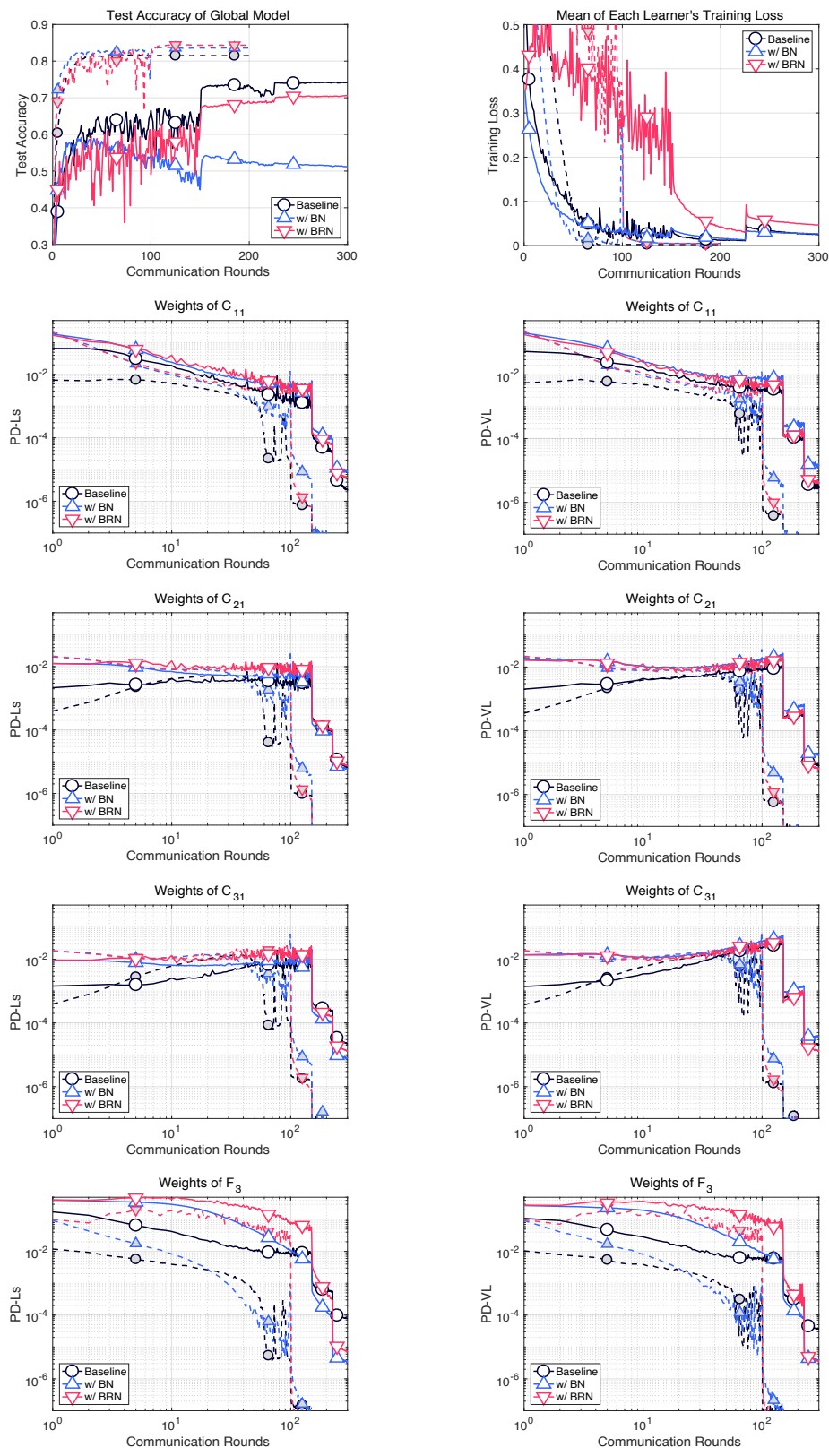

Figure 27: Behavior comparison with/without Batch Normalization/Renormalization in the training of NetA-Baseline on CIFAR-10 (Optimizer: NMom-A). Dotted and solid lines indicate the results under IID and Non-IID(2) setting, respectively.

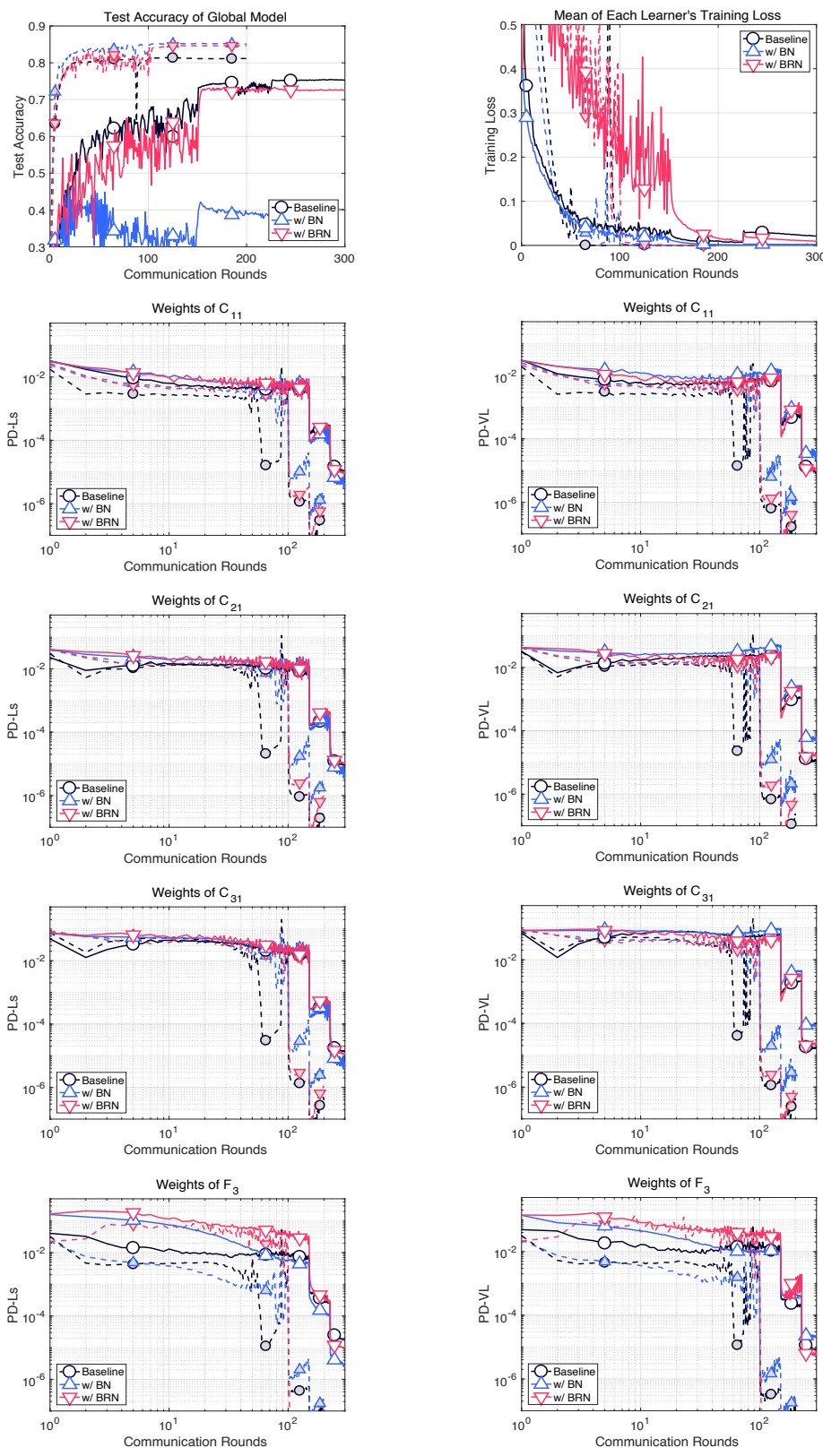

Figure 28: Behavior comparison with/without Batch Normalization/Renormalization in the training of NetA-Baseline on CIFAR-10 (Optimizer: Adam-A). Dotted and solid lines indicate the results under IID and Non-IID(2) setting, respectively.

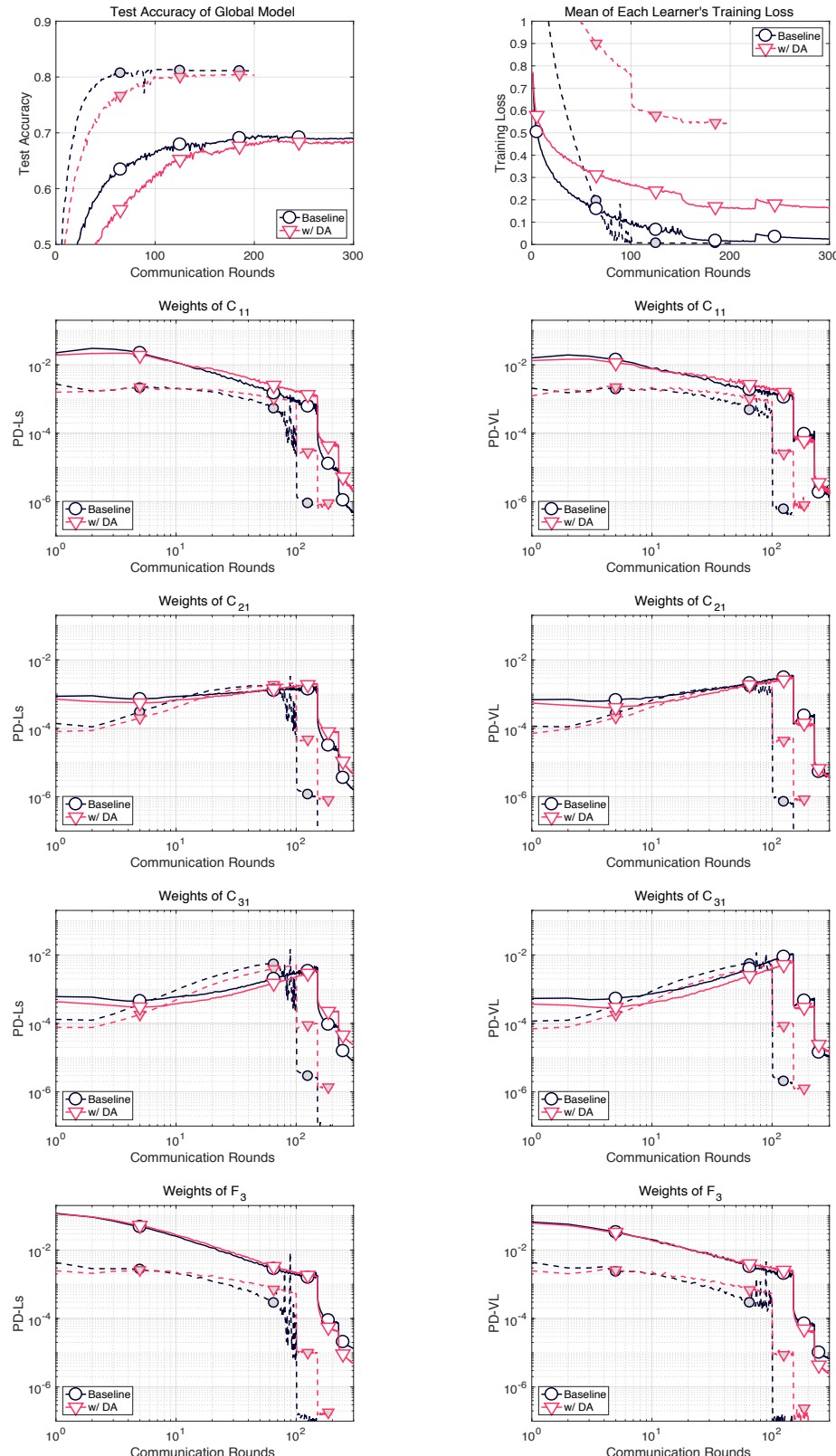

Figure 29: Behavior comparison with/without data augmentation in the training of NetA-Baseline on CIFAR-10 (Optimizer: pure SGD). Dotted and solid lines indicate the results under IID and Non-IID(2) setting, respectively.

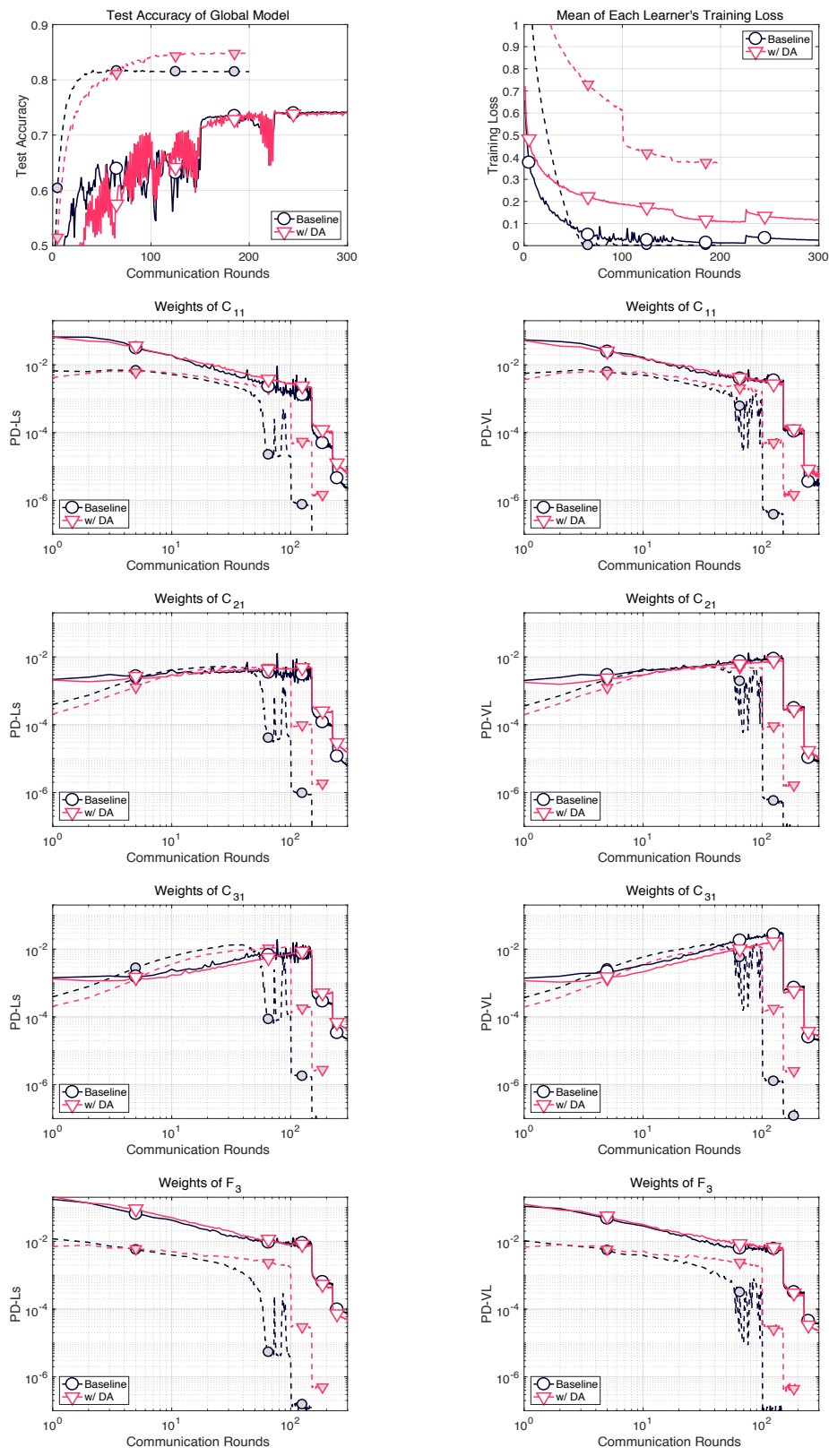

Figure 30: Behavior comparison with/without data augmentation in the training of NetA-Baseline on CIFAR-10 (Optimizer: NMom-A). Dotted and solid lines indicate the results under IID and Non-IID(2) setting, respectively.

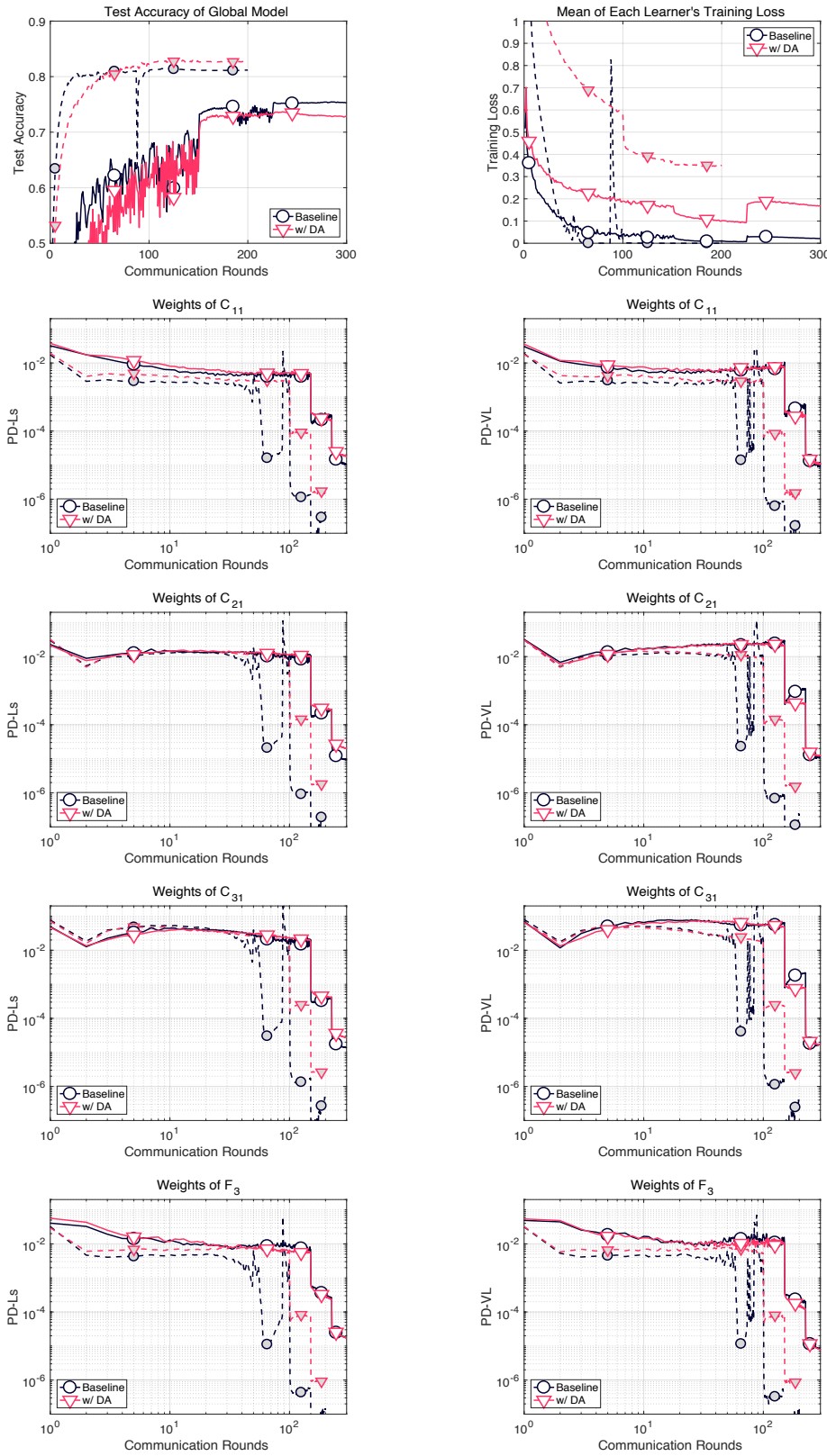

Figure 31: Behavior comparison with/without data augmentation in the training of NetA-Baseline on CIFAR-10 (Optimizer: Adam-A). Dotted and solid lines indicate the results under IID and Non-IID(2) setting, respectively.

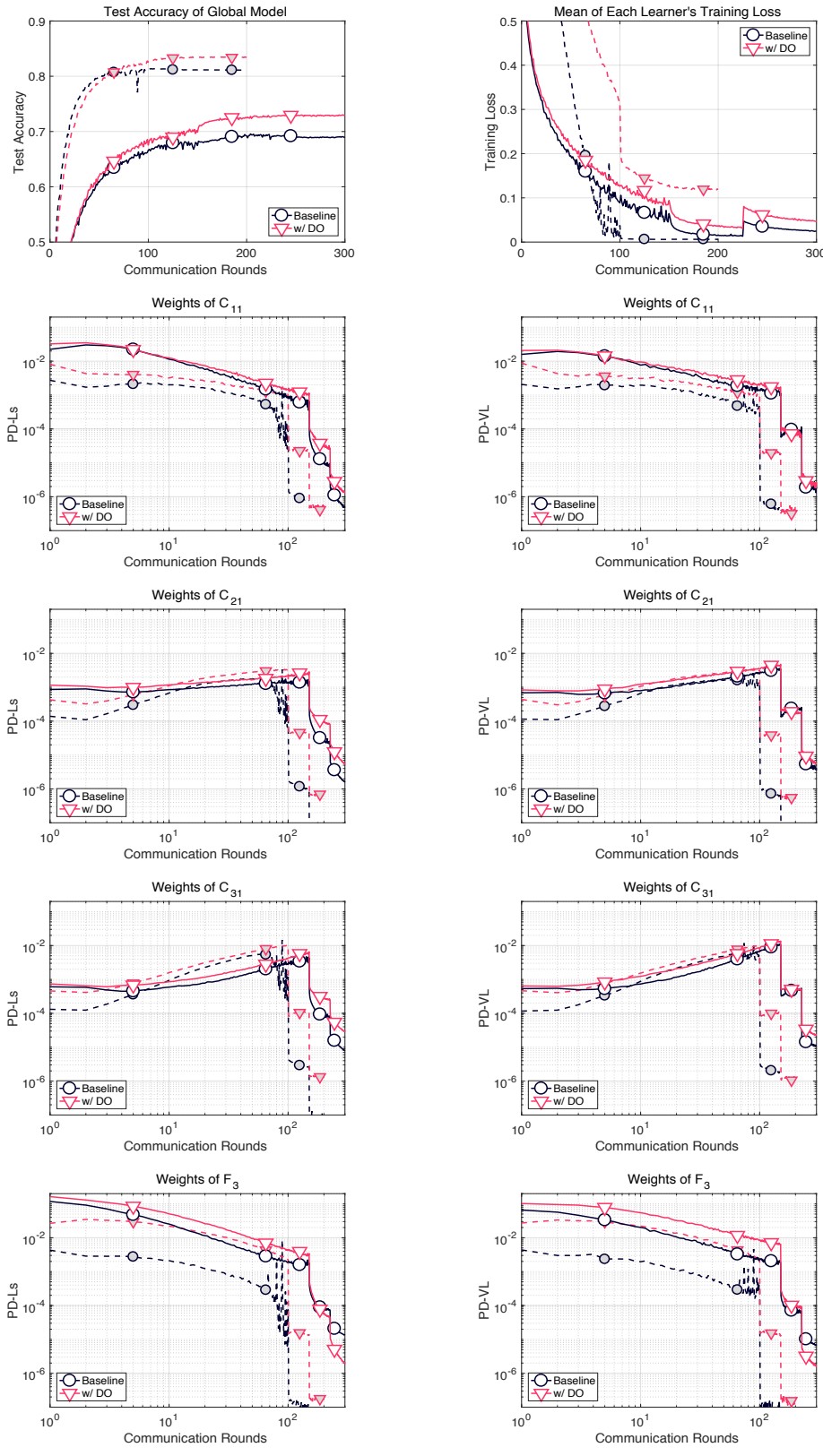

Figure 32: Behavior comparison with/without Dropout in the training of NetA-Baseline on CIFAR-10 (Optimizer: pure SGD). Dotted and solid lines indicate the results under IID and Non-IID(2) setting, respectively.

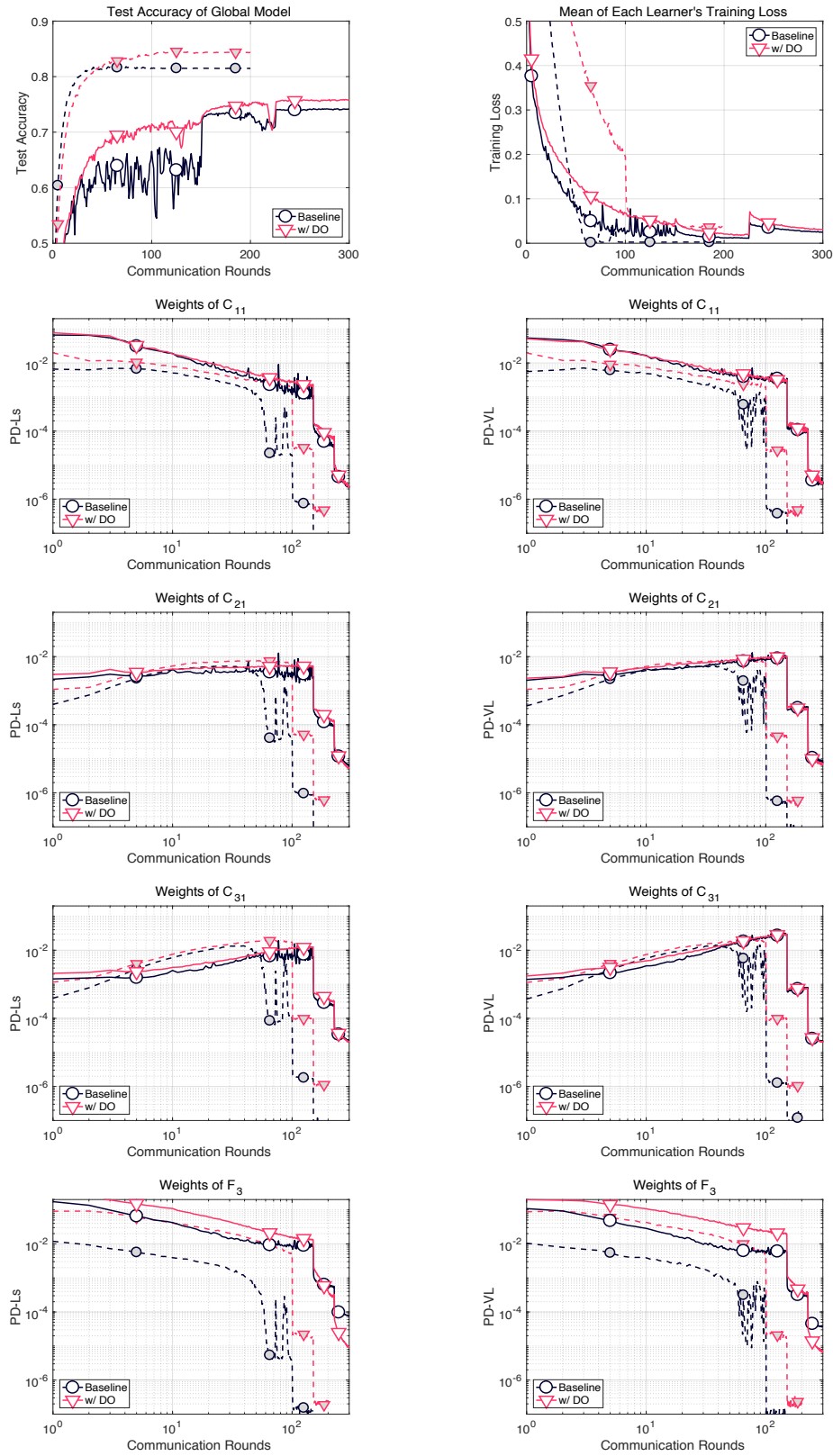

Figure 33: Behavior comparison with/without Dropout in the training of NetA-Baseline on CIFAR-10 (Optimizer: NMom-A). Dotted and solid lines indicate the results under IID and Non-IID(2) setting, respectively.

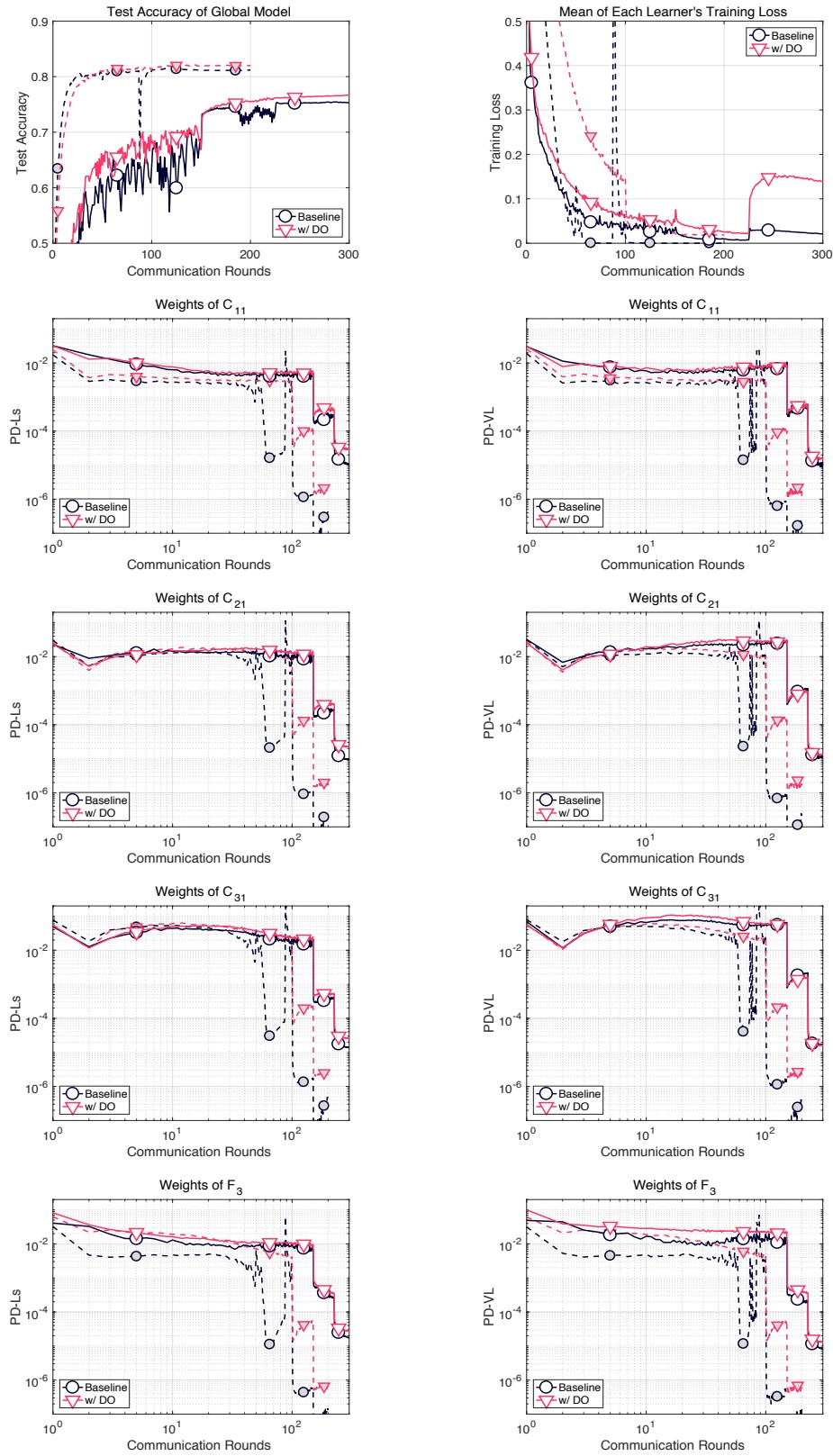

Figure 34: Behavior comparison with/without Dropout in the training of NetA-Baseline on CIFAR-10 (Optimizer: Adam-A). Dotted and solid lines indicate the results under IID and Non-IID(2) setting, respectively.

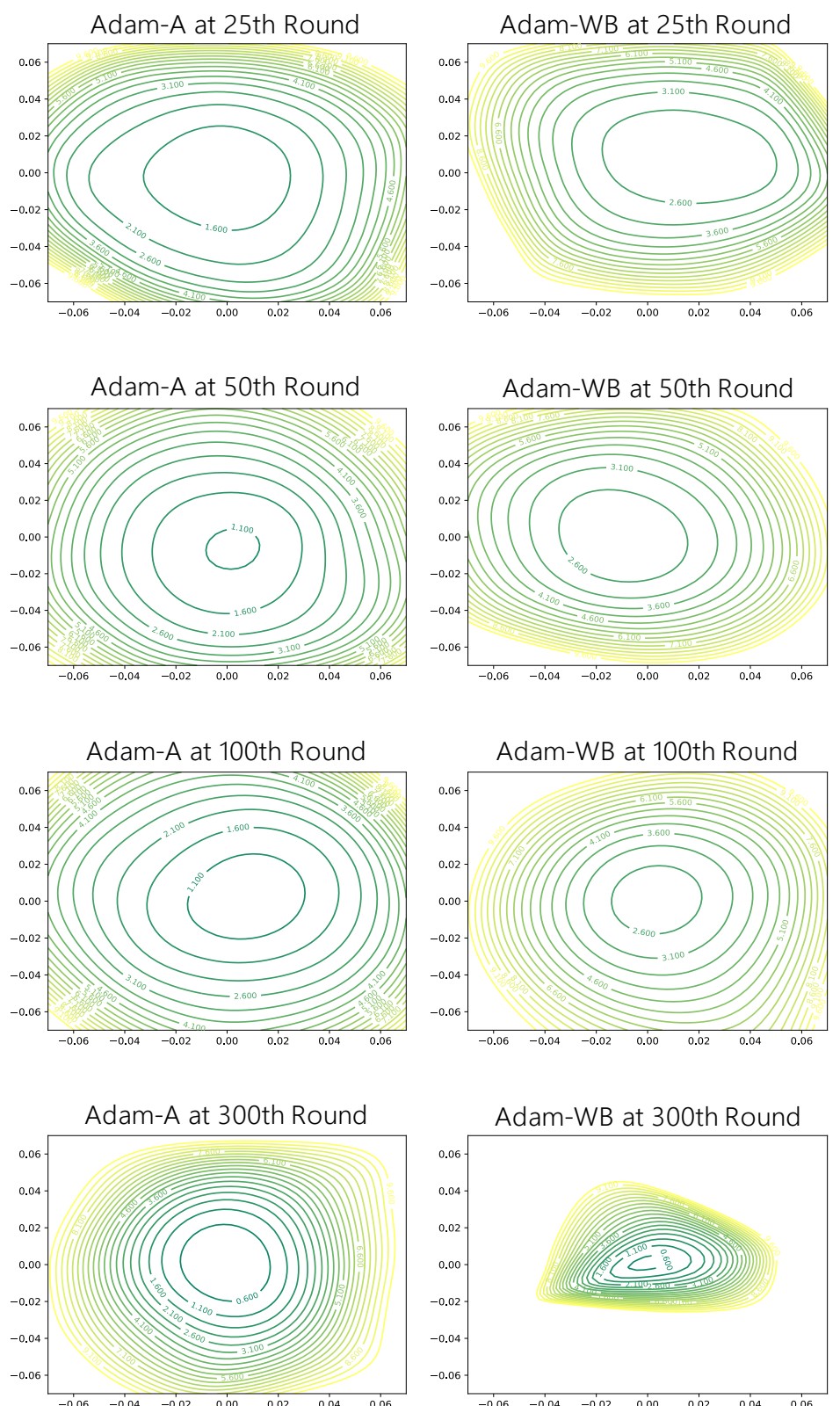

Figure 35: Loss surface of the global model parameters with Adam-WB and with Adam-A under Non-IID(2) setting.

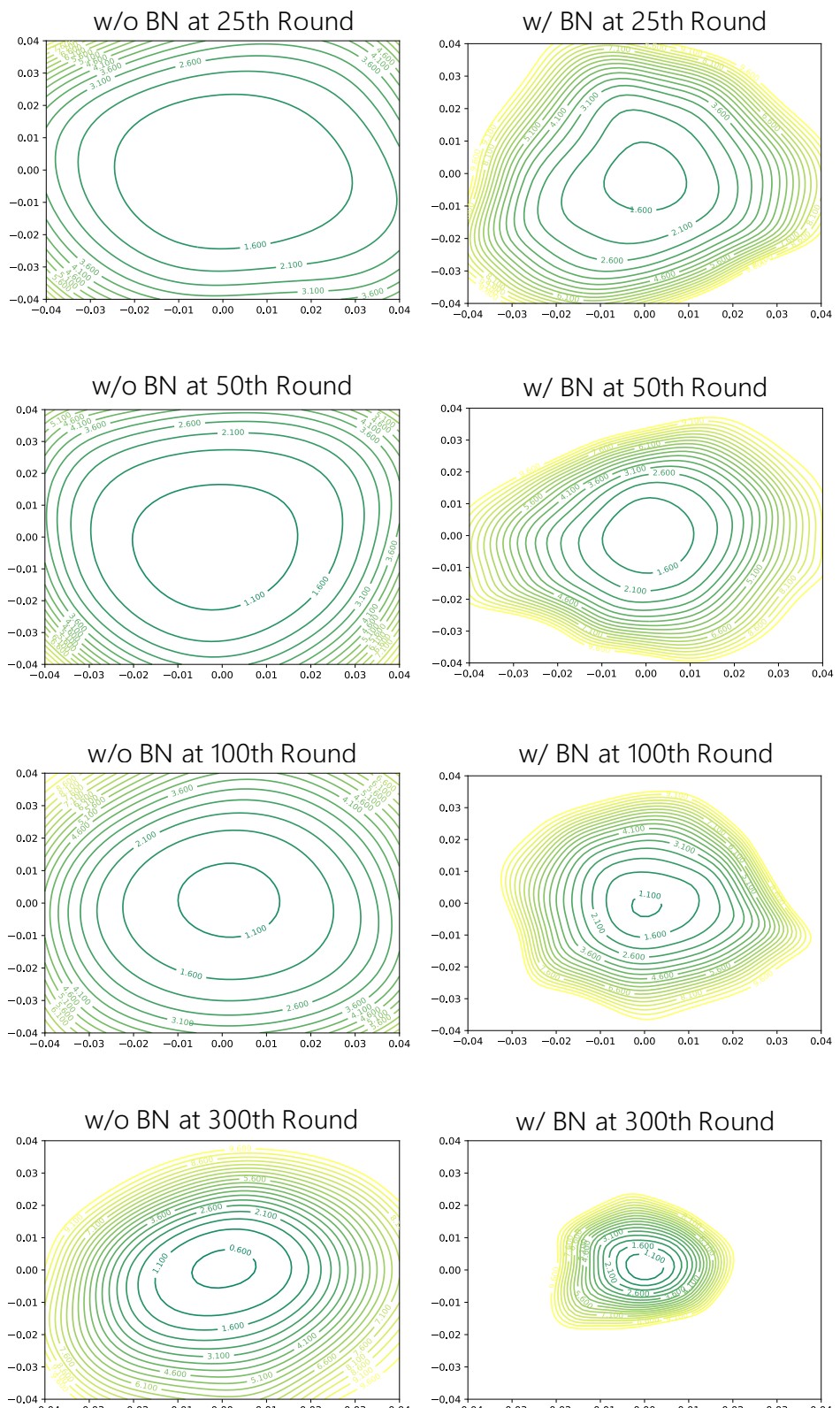

Figure 36: Loss surface of the global model parameters with/without Batch Normalization under Non-IID(2) setting.

