# OpenReview forum: "On Federated Learning of Deep Networks from Non-IID Data: Parameter Divergence and the Effects of Hyperparametric Methods"
_ICLR.cc/2020/Conference — Reject_

### Official Review · AnonReviewer1 · 2019-10-23
**Official Blind Review #1**

**Rating:** 3

**Review:**

In this paper, the authors empirically investigate parameter divergence of local updates in federated learning with non-IID data. The authors study the effects of optimizers, network depth/width, and regularization techniques, and provide some observations. In overall, I think this paper study an important problem in federated learning.

However, there are some weakness in this paper:

1. The paper is nearly pure empirical. There is no theoretical analysis supporting the observations proposed in Section 4.1, which weaken the contribution of this paper.

2. This paper only raises some issues in federated learning with non-IID data, and discusses the potential causes. No suggestions or potential solutions is proposed in this paper, which weaken the contribution of this paper.

3. Since this is nearly a pure empirical paper, I hope the authors can make the experiments thorough. However, there are some experiments I expect to see but not yet included in this paper:

    3.1. The authors only studies Nesterov momentum in this paper. However, in practice, it is more common to use Polyak momentum. I hope the authors can also study FL SGD with Polyak momentum in this paper.

    3.2. In this paper, the authors assume that different workers has the same number of local data samples (in Definition 1). However, due to the heterogeneous setting, it is very likely that different workers have different numbers of local data samples, which could be another source of divergence. Furthermore, different numbers of local data samples also results in different numbers of local steps, which may also cause divergence.

    3.3. [1] proposes a regularization mechanism (FedProx) to deal with the heterogeneity. Instead of studying weight decay, it is more reasonable to study the regularization technique proposed by [1].


4. There are some missing details (maybe they are already in the paper but I didn't find them):

    4.1. What is the definition of Adam-A and Adam-WB? And, what are the differences between Adam-A, Adam-WB, and vanilla  Adam? (and also, what is the "A" in NMom-A?)

    4.2. When using Adam in federated learning, how are the variables synchronized? Note that for Adam, there are 3 sets of variables: model parameters, 1st moment, and 2nd moment. Due to the local updates, all the 3 sets of variables are not synchronized. When the authors use Adam in FL, did they only synchronize/average the model parameter and ignore the 1st and 2nd moments, or did they synchronize all the 3 sets of variables?


----------------
Reference

[1]  Li, Tian et al. “Federated Optimization for Heterogeneous Networks.” (2018).

**Experience Assessment:**

I have published one or two papers in this area.

**Review Assessment: Checking Correctness Of Derivations And Theory:**

N/A

**Review Assessment: Checking Correctness Of Experiments:**

I assessed the sensibility of the experiments.

**Review Assessment: Thoroughness In Paper Reading:**

I read the paper at least twice and used my best judgement in assessing the paper.

---

> ### Author Response · Authors · 2019-11-15
> **Author response to reviewer #1 (1/3)**
>
>
> We first appreciate the valuable comments. We carefully looked through all the comments; the following describes our answers.
>
> 1. The paper is nearly pure empirical. There is no theoretical analysis supporting the observations proposed in Section 4.1, which weaken the contribution of this paper.
> 2. This paper only raises some issues in federated learning with non-IID data, and discusses the potential causes. No suggestions or potential solutions is proposed in this paper, which weaken the contribution of this paper.
>  =====(Answer for Question 1 and 2)=====
> We appreciate the valuable comments, and we admit your concerns.
> Nevertheless, we believe that focusing on federated learning with non-IID data, our work provides the meaningful exploratory analysis breaking the existing common wisdom about the considered hyperparameter optimization methods. In relation, here we intend to emphasize our contributions.
>
> Our distinct contributions can be highlighted as follows:
>
> **Regarding Section 3:
> In many previous literatures, e.g., (Zhao et al., 2018), parameter divergence is regarded as a direct response to learners’ local data being non-IID sampled from the population distribution. In relation, it was reported that as the probabilistic distance (e.g., earth mover’s distance) of learners’ local data becomes farther away from the population distribution, bigger parameter divergence might appear; this is correlated with the degradation of performance such as test accuracy (please refer to Section 3.2 of (Zhao et al., 2018)). Also, we added our analysis of the relationship among the three factors (i.e., probabilistic distance, parameter divergence, and performance) in the rebuttal period; the relevant description can be found in Section 3 of the revised version of this paper.
>
> Regarding the parameter divergence, our distinct contribution can be summarized in two-fold:
> First, for the first time we identified the mechanism by which data non-IIDness affects the parameter divergence: “if data distributions in each local dataset are highly skewed and heterogeneous over classes, subsets of neurons, which have especially big magnitudes of the gradients in back propagation, become significantly different across learners; this leads to inordinate parameter divergence between them”. It has been analyzed in both empirical and theoretical way.
> Second, many of the related literatures usually handle the parameter difference of each learner’s local model parameters from one computed with the population distribution (this philosophy is connected to the definition of PD-VL); meanwhile, in our study we also considered the parameter diversity between the local updates as well (this is connected to the definition of PD-Ls). The reason of probing parameter divergence being important is that the federated learning are performed based on iterative parameter averaging. That is, investigating how local updates are diverged can give a clue whether the subsequent parameter averaging yields positive returns; the proposed divergence metrics provide two ways for it.
>
> **Regarding Section 4.1:
> In this study, we focused on the well-known hyperparameter optimization strategies (i.e., hyperparametric strategies) to improve learning performance: (i) using momentum SGD or Adam than pure SGD, (ii) network deepening/widening (until a proper level), (iii) Batch Normalization, (iv) weight decay, (v) data augmentation, and (vi) Dropout. Their positive effects have been reported in a variety of literatures; practically, they are being broadly used in deep net training. Also in our experiments, the hyperparametric methods yielded better outcome under vanilla training (i.e., non-distributed training) and under the considered federated learning algorithm with the IID decentralized data setting.
> However, under the non-IID data setting, we newly identified that the hyperparametric methods could rather give negative/diminishing effects on performance of the federated learning algorithm; we believe that these findings can be highly impactful to the upcoming works or industrial implementations.

---

> > ### Author Response · Authors · 2019-11-15
> > **Author response to reviewer #1 (2/3)**
> >
> >
> > **Regarding Section 4.2:
> > In many previous literatures, e.g., (Zhao et al., 2018), inordinate magnitude of parameter divergence is regarded as a direct response to learners’ local data being non-IID sampled from the population distribution; thus they explained that the consequent parameter averaging with the highly diverged local updates could lead to bad solutions far from the global optimum. Likewise, in our experiments, for many of the failure cases under the non-IID data setting, we observed that the inordinate magnitude of parameter divergence could become one of the internal causes of the diminishing returns.
> > However, under the non-IID data setting, some of the failure cases have been observed where the test accuracy is still low but the parameter divergence values decrease (rapidly) over rounds; as the round goes, even the values were sometimes seen to be lower than those of the comparison targets. For the failure cases, we concluded that these (unexpected abnormal) sudden drop of parameter divergence values indicate going into poor local minima (or saddles); this can be supported by the behaviors that test accuracy increases plausibly at very early rounds, but the growth rate quickly stagnates and eventually becomes much lower than the comparison targets.
> > In relation, we provided Figure 5 (in the revised version of this paper) as the evidence of the steep fall phenomenon; as depicted in the figure, the loss landscapes of the failure cases (i.e., “Adam-WB” and “w/ BN” under the non-IID setting) show sharper minima and the minimal value in the bowl is relatively greater. Here “sharp” minima is broadly known to lead to poorer generalization ability (Hochreiter & Schmidhuber, 1997; Keskar et al., 2017); it is observed from the figure that going into a sharp minima happens even in early rounds (e.g., 25th).
> > It is expected that the discovery of these steep fall phenomena provides a new insight into the relationship between test accuracy and parameter divergence; we believe that the steep fall phenomenon should be considered as the cause of diminishing returns of the federated learning with non-IID data, along with the inordinate magnitude of parameter divergence.
> >
> > (Zhao et al., 2018) Yue Zhao, Meng Li, Liangzhen Lai, Naveen Suda, Damon Civin, and Vikas Chandra. Federated learning with non-IID data. arXiv preprint arXiv: 1806.00582, 2018.
> > (Hochreiter & Schmidhuber, 1997) Sepp Hochreiter and Jurgen Schmidhuber. Flat minima. Neural Computation, 9(1), 1997.
> > (Keskar et al., 2017) Nitish Shirish Keskar, Dheevatsa Mudigere, Jorge Nocedal, Mikhail Smelyanskiy, and Ping Tak Peter Tang. On large-batch training for deep learning: Generalization gap and sharp minima. In ICLR, 2017.
> >
> >
> > 3. Since this is nearly a pure empirical paper, I hope the authors can make the experiments thorough. However, there are some experiments I expect to see but not yet included in this paper:
> >
> > 3.1. The authors only studies Nesterov momentum in this paper. However, in practice, it is more common to use Polyak momentum. I hope the authors can also study FL SGD with Polyak momentum in this paper.
> > =====(Answer)=====
> > We appreciate the valuable suggestion. We conducted the corresponding experiments; for the details, please see Tables 7-13 in the revised version of the paper.
> >
> > 3.2. In this paper, the authors assume that different workers has the same number of local data samples (in Definition 1). However, due to the heterogeneous setting, it is very likely that different workers have different numbers of local data samples, which could be another source of divergence. Furthermore, different numbers of local data samples also results in different numbers of local steps, which may also cause divergence.
> > =====(Answer)=====
> > As you point out, since the federated learning do not require centralizing local data, data unbalancedness (i.e., each learner has various numbers of local data examples) would be also naturally assumed in the federated learning along with the data non-IIDness (McMahan et al., 2017). We appreciate the valuable suggestion. We conducted the corresponding experiments; for the details, please see Appendix C.8 in the revised version of the paper.
> >
> > (McMahan et al., 2017) H. Brendan McMahan, Eider Moore, Daniel Ramage, Seth Hampson, and Blaise Agu ̈era y Arcas. Communication-efficient learning of deep networks from decentralized data. In AISTATS, 2017.
> >
> > 3.3. [1] proposes a regularization mechanism (FedProx) to deal with the heterogeneity. Instead of studying weight decay, it is more reasonable to study the regularization technique proposed by [1].
> > =====(Answer)=====
> > We appreciate the valuable suggestion. We conducted the corresponding experiments; for the details, please see Appendix C.4 in the revised version of the paper.

---

> > > ### Author Response · Authors · 2019-11-15
> > > **Author response to reviewer #1 (3/3)**
> > >
> > >
> > > 4. There are some missing details (maybe they are already in the paper but I didn't find them):
> > >
> > > 4.1. What is the definition of Adam-A and Adam-WB? And, what are the differences between Adam-A, Adam-WB, and vanilla Adam? (and also, what is the "A" in NMom-A?)
> > > =====(Answer)=====
> > > We apologize for your confusion. As mentioned at the second paragraph of “Steep fall phenomenon” of Section 4.2 (in the original version of the paper),
> > > “(optimizer name)-A”: under the certain optimizer, the parameter averaging being performed for all the variables;
> > > “(optimizer name)-WB”: under the certain optimizer, the parameter averaging being performed only for weights & biases.
> > > Therefore, we conducted an analysis of Adam-A vs Adam-WB; and NMom-A vs NMom-WB (NMom: Nesterov momentum SGD optimizer). In addition, as mentioned in Footnote 3 and the first paragraph of Appendix C, “vanilla” training refers to non-distributed training with a single machine, using the whole training examples; for the vanilla training, we trained the networks for 100 epochs.
> > > Please note that in the revised version, the location of the mention about the definition of “(optimizer name)-A” and “(optimizer name)-WB” has been changed to inside Section 4.1.
> > >
> > > 4.2. When using Adam in federated learning, how are the variables synchronized? Note that for Adam, there are 3 sets of variables: model parameters, 1st moment, and 2nd moment. Due to the local updates, all the 3 sets of variables are not synchronized. When the authors use Adam in FL, did they only synchronize/average the model parameter and ignore the 1st and 2nd moments, or did they synchronize all the 3 sets of variables?
> > > =====(Answer)=====
> > > As we stated above, we experimented with both the “(optimizer name)-A” and “(optimizer name)-WB” cases. To the best of our knowledge, so far there have been no studies about Adam to synchronize all the 3 sets of variables under federated learning. However, in the momentum SGD case, there have been some literatures; for instance, (Lin et al., 2018) presented methods with “Local Momentum”, “Global Momentum”, and “Hybrid Momentum”. In our experiments, “Adam-A” and “NMom-A” take the simple averaging strategy for all the 3 sets (i.e., model parameters, 1st moment, and 2nd moment for Adam) and 2 sets (i.e., model parameters and the momentum for momentum SGD) of variables, respectively; it has the similar philosophy with the “Local Momentum” method. One can see from Table 4 in (Lin et al., 2018) that the simple averaging strategy can yield still competitive results compared to “Global Momentum” or “Hybrid Momentum” method. This answer was reflected in Footnote 7 of the revised version of the paper. Please also refer to Table 7 in the appendix of the revised version the paper.
> > >
> > > (Lin et al., 2018) Tao Lin, Sebastian U. Stich, Kumar Kshitij Patel, and Martin Jaggi. Don’t use large mini-batches, use local SGD. arXiv preprint arXiv: 1808.07217, 2018.

---

### Official Review · AnonReviewer3 · 2019-10-23
**Official Blind Review #3**

**Rating:** 3

**Review:**

The paper experimentally studies the reasons for the slow convergence of the Federated Averaging algorithm when the data are non-iid distributed between workers in the multiclass-classification case. Paper performs extensive experimental study and observes that the main reasons for failure are connected to (i) the parameter divergence during the local steps, (ii) steep-fall phenomena when parameters on different nodes are getting close fast, and to the (iii) high training loss.

My score is weak reject. The paper provides extensive but unclear experimental results. Improving presentation would significantly improve the paper. For example, why in experimental and theoretical study different parameter divergence metrics were used, etc (see below), why different networks use different optimizers.
Moreover, provided experimental comparison might be unfair. The learning rate is constant throughout all of the experiments, depending only on the optimizer, but not on the neural network architecture. This can affect the final results.

Concerns and questions that should be addressed:
1. The initial learning rates were not tuned properly. It is set to be the same for different neural network topologies, which might significantly affect the results. What did the choice of initial learning rates is based on?

2. Why the parameter divergence metric in Definition 1 is not the same as in the theoretical study (Appendix B)? What is the intuition behind Definition 1?

3. Why the divergence of parameters is considered only at the last layer? It seems to hide many important interactions in the other layers.

4. Some important experimental details --- should be added:
   - At which moment the parameter divergence is computed in the plots? Is it computed at the end of the local iterations right before synchronization?
   - How the training loss was computed in the plots? before or after synchronization? on the local only or the global data?
   - Which batch size was used?
   - Improve the figure caption to detail the experimental setup. (e.g. in fig 3. the network architecture was mentioned only for one of the figures, include which optimized was used, etc)

5. In experiments on Fig. 2. and Fig.3 (middle) what is the accuracy for IID baseline? Is the observed phenomena connected to the poor network architecture or to the non-iid data?

6. In table 5 of the appendix, why experiments use Adam optimizer, but not Momentum SGD as in the main paper to compare the performance of ResNet14 and ResNet20?

7. Better re-prase the definition of the steep fall phenomena, now it is not very clear: in the IID setting parameter divergence values are also sometimes reducing sharply; in the network width study parameters divergence doesn’t experience sudden drop. Also, how does this phenomena (and parameter divergence too) connects to the training loss?

8. Why for different experiments different baseline models are used? (NetA, NetB, NetC)


Other minor comments:
- Appendix B, first equation on page 13. (d_q)^t -> (d_q)^t_k; The size of gradient \nabla_w [E ...] is different from the size of (d_q)_k. They cannot be added together.
- page 7, last sentence of the first paragraph: what is the accuracy achieved with Batch Renormalization? Why the reason for accuracy gap is “significant parameter divergence”? on fig. 3 “parameter divergence” is smaller than for the baseline.
- Why the name of the section on page 7 is “excessively high training loss of local updates” if later it is stated that it is actually smaller than for the IID case?
- Defenition 1, line 4: “the then” -> “the”
- section 3: “A pleasant level of parameter divergence can help to improve generalization” -> where was it shown?
- section 4.2, paragraph 2: what is meant by “hyperparametric methods”?
- section 4.2, paragraph 3: “quantitative increase in a layer level” -> not clear what does it mean.
- page 4, effect of optimizers: what do you refer to as “all model parameters”?
- page 5, last paragraph: Hinton et al... -> (Hinton et al…). Use \citet(\citep) instead of \cite.
- why Dropout yields bigger parameter divergence if on Fig 2, right it actually helps?
- Last line of the page 5. Where was this observed?


**Experience Assessment:**

I have read many papers in this area.

**Review Assessment: Checking Correctness Of Derivations And Theory:**

N/A

**Review Assessment: Checking Correctness Of Experiments:**

I assessed the sensibility of the experiments.

**Review Assessment: Thoroughness In Paper Reading:**

I read the paper at least twice and used my best judgement in assessing the paper.

---

> ### Author Response · Authors · 2019-11-15
> **Author response to reviewer #3 (1/6)**
>
>
> We first appreciate the valuable comments. We carefully looked through all the comments; the following describes our answers.
>
> 1. The initial learning rates were not tuned properly. It is set to be the same for different neural network topologies, which might significantly affect the results. What did the choice of initial learning rates is based on?
> =====(Answer)=====
> As you remark, the initial learning rates were set to be the same for different model architectures. Therefore, the best results might not have been obtained with regard to the learning rates. Nevertheless, the choice of the initial learning rates was conducted based on the follows:
> (i) Based on the results of Appendix B as well as the intuitive thoughts, (especially before the first learning rate drop) learning rates may highly affect the values of the parameter divergence. Therefore, we set the initial learning rates the same for the compared cases (e.g., NetA-Baseline vs NetA-Deeper vs NetA-Deepest) so that the corresponding parameter divergence values could be compared under the same conditions.
> (ii) In addition, one of the main objective in the paper is to show that the considered hyper parameter optimization strategies (which have been reported that they yield better outcome under “vanilla” training or under the federated learning with IID data) could rather result in the diminishing returns under non-IID data setting.
> As described in Tables 7-13 of Appendix C (in the revised version of the paper), we can see that under “vanilla training” (especially for batch size: 50) and under the federated learning with the IID data setting, most of the results are shown to be similar with what we already know (e.g., the advantages of deeper network architectures, global average pooling, Batch Normalization, and so on). However, under the federated learning with the Non-IID(2) data setting, we can see that some of the hyperparameter optmization methods rather yield the highly conflicted results (i.e., the diminishing returns).
> Therefore, in summary, our setting of the initial learning rates could be rather far from the best results; nevertheless, from Tables 7-13 the results can be interpreted as still valid (since the results under “vanilla training” and under the federated learning with the IID data setting follow the similar trends to those well known). In addition, we believe that our setting also provides the fair comparison of parameter divergence.
>
>
> 2. Why the parameter divergence metric in Definition 1 is not the same as in the theoretical study (Appendix B)? What is the intuition behind Definition 1?
> =====(Answer)=====
> We first remark that PD-Ls in Definition 1 and ||(d_q)^(t+1)_i - (d_q)^(t+1)_j|| are related. In the case of Figure 1 (and Appendix B), we used the same network architecture and training methods. Manipulated variables here is only data distributions (i.e., IID, Non-IID(2), and Non-IID(1)). Therefore, ||(d_q)^(t+1)_i - (d_q)^(t+1)_j|| can be validly utilized. However, in most of our experiments, we compared the different network architectures (e.g., NetC-Baseline, NetC-Wider, and NetC-Widest) or the effects of the different training settings (e.g., various weight decay factors) together in a set. Therefore, for instance, in the case of NetC-Baseline, NetC-Wider, and NetC-Widest, the number of neurons in the output layer becomes different (i.e., 2560, 10240, and 40960, respectively); in the case of various weight decay factors, the degree to which the model parameters from the previous iteration are reflected in the current parameters highly depends on the factor values. Therefore, we thought that we need a normalized (qualitative) metric rather than simply considering the magnitude of parameter (weight) differences; consequently, instead of the euclidean distance, we used cosine distance-based metrics in Definition 1. This answer was reflected in the third paragraph of Section 3 in the revised version of the paper.

---

> > ### Author Response · Authors · 2019-11-15
> > **Author response to reviewer #3 (2/6)**
> >
> >
> > 3. Why the divergence of parameters is considered only at the last layer? It seems to hide many important interactions in the other layers.
> > =====(Answer)=====
> > At first, please remind that regarding each experimental trial, in Figures 9-34 (of the revised version of the paper) we provide the PD-Ls and the PD-VL graphs for each four layers. From the figures of the experimental results in Appendix C, we can identify that in most cases the parameter divergence values of the first convolutional layer and the last fully-connected layer would be more dominant than those of the other layers, judging from their difference of magnitude between under the IID and the non-IID data setting (please also note that log scale is used for the y-axis). We additionally remark that the results of other related studies also show the dominance of the first convolutional layer and the last fully-connected layer (e.g., see Figure 2 in Zhao et al. (2018)). Therefore, our discussion here was primarily described based on the results of the first convolutional layer and the last fully-connected layer. This answer was reflected in Footnote 8 of the revised version of the paper.
> >
> > (Zhao et al., 2018) Yue Zhao, Meng Li, Liangzhen Lai, Naveen Suda, Damon Civin, and Vikas Chandra. Federated learning with non-IID data. arXiv preprint arXiv: 1806.00582, 2018.
> >
> >
> > 4. Some important experimental details --- should be added:
> >
> > We apologize for your confusion, and we made them clear in the revised version.
> >
> > - At which moment the parameter divergence is computed in the plots? Is it computed at the end of the local iterations right before synchronization?
> > =====(Answer)=====
> > According to the definition of w^t_k in Algorithm 1, the values of PD-VL and PD-Ls are computed at the end of the local iterations right before synchronization.
> >
> > - How the training loss was computed in the plots? before or after synchronization? on the local only or the global data?
> > =====(Answer)=====
> > Training loss values in the plots such as Figure 6 (of the revised version of the paper) are mean of each learner’s training loss before each synchronization; each learner’s training loss values were calculated on their local data. In the revised version of the paper, we added this information at the caption of Figure 6.
> >
> > - Which batch size was used?
> > =====(Answer)=====
> > As stated in Table 1 (of the original version of the paper), the minibatch size was set to be 50 for the considered federated learning algorithm. => Please note that in the revised version, the location of the mention about the minibatch size has been changed to inside the text in “Environmental configuration” of Section 2.2. In addition, in Tables 7-13 (of the revised version of the paper), the results under vanilla training include both when the minibatch size is 50 and 500.
> >
> > - Improve the figure caption to detail the experimental setup. (e.g. in fig 3. the network architecture was mentioned only for one of the figures, include which optimized was used, etc)
> > =====(Answer)=====
> > As stated in “Baseline network model”, we used NetA-Baseline as our baseline network architecture; without the specific mention of the network architecture, the NetA-Baseline network is considered for Figures 3-6 in the revised version of the paper. To be more clear, we provided the corresponding mention again through Tables 2-4.
> > In addition, we clarified the remaining setups in the first paragraph of Section 4.2 in the revised version of the paper as follows:
> > “Note that our discussion in this subsection is mostly made from the results under Nesterov momentum SGD and on CIFAR-10; the complete results including other optimizers (e.g., pure SGD, Polyak momentum SGD, and Adam) and datasets (e.g., SVHN) are given in Appendix C.”

---

> > > ### Author Response · Authors · 2019-11-15
> > > **Author response to reviewer #3 (3/6)**
> > >
> > >
> > > 5. In experiments on Fig. 2. and Fig.3 (middle) what is the accuracy for IID baseline? Is the observed phenomena connected to the poor network architecture or to the non-iid data?
> > > =====(Answer)=====
> > > We apologize for your confusion. Each test accuracy values (under both IID and Non-IID(2) setting) is found in the tables in Appendix C. In the revised version of the paper, we also provided the corresponding tables in Section 4.2 (see Tables 2-4).
> > >
> > > In this study, we focused on the well-known hyperparameter optimization strategies (i.e., hyperparametric strategies) to improve learning performance: (i) using momentum SGD or Adam than pure SGD, (ii) network deepening/widening (until a proper level), (iii) weight decay, (iv) Batch Normalization, (v) data augmentation, and (vi) Dropout. Their positive effects have been reported in a variety of literatures; practically, they are being broadly used in deep net training to improve learning performance. Also in our experiments, the hyperparametric methods yielded better outcome under vanilla training (i.e., non-distributed training) and under the considered federated learning algorithm with the IID decentralized data setting.
> > > However, under the non-IID data setting, we newly identified that the hyperparametric methods could rather give negative/diminishing effects on performance of the federated learning algorithm; we believe that these findings can be highly impactful to the upcoming works or industrial implementations.
> > >
> > >
> > > 6. In table 5 of the appendix, why experiments use Adam optimizer, but not Momentum SGD as in the main paper to compare the performance of ResNet14 and ResNet20?
> > > =====(Answer)=====
> > > We appreciate the valuable suggestion. We conducted the corresponding experiments; for the details, please see Table 8 in the appendix of the revised version of the paper. We additionally note that regarding the ResNet results, there was some error in the original version of the paper; we corrected this in the revised version.

---

> > > > ### Author Response · Authors · 2019-11-15
> > > > **Author response to reviewer #3 (4/6)**
> > > >
> > > >
> > > > 7. Better re-prase the definition of the steep fall phenomena, now it is not very clear: in the IID setting parameter divergence values are also sometimes reducing sharply; in the network width study parameters divergence doesn’t experience sudden drop. Also, how does this phenomena (and parameter divergence too) connects to the training loss?
> > > > =====(Answer)=====
> > > > We appreciate the thankful suggestion. As you point out, in the original version of this paper, the definition of the steep fall phenomenon might be described somewhat ambiguously.
> > > > Instead of simply describing sudden drop of the parameter divergence values in the last fully-connected layer, the philosophy behind the steep fall phenomenon is as follows:
> > > > In many previous literatures, e.g., (Zhao et al., 2018), inordinate magnitude of parameter divergence is regarded as a direct response to learners’ local data being non-IID sampled from the population distribution; thus they explained that the consequent parameter averaging with the highly diverged local updates could lead to bad solutions far from the global optimum. Likewise, in our experiments, for many of the failure cases under the non-IID data setting, we observed that the inordinate magnitude of parameter divergence could become one of the internal causes of the diminishing returns.
> > > > However, under the non-IID data setting, some of the failure cases have been observed where the test accuracy is still low but the parameter divergence values decrease (rapidly) over rounds; as the round goes, even the values were sometimes seen to be lower than those of the comparison targets. For the failure cases, we concluded that these (unexpected abnormal) sudden drop of parameter divergence values indicate going into poor local minima (or saddles); this can be supported by the behaviors that test accuracy increases plausibly at very early rounds, but the growth rate quickly stagnates and eventually becomes much lower than the comparison targets.
> > > > In relation, we provided Figure 5 (in the revised version of this paper) as the evidence of the steep fall phenomenon; as depicted in the figure, the loss landscapes of the failure cases (i.e., “Adam-WB” and “w/ BN” under the non-IID setting) show sharper minima and the minimal value in the bowl is relatively greater. Here “sharp” minima is broadly known to lead to poorer generalization ability (Hochreiter & Schmidhuber, 1997; Keskar et al., 2017); it is observed from the figure that going into a sharp minima happens even in early rounds (e.g., 25th).
> > > > It is expected that the discovery of these steep fall phenomena provides a new insight into the relationship between test accuracy and parameter divergence; we believe that the steep fall phenomenon should be considered as the cause of diminishing returns of the federated learning with non-IID data, along with the inordinate magnitude of parameter divergence.
> > > > This answer was reflected in “Steep fall phenomenon” of Section 4.2 in the revised version of the paper.
> > > >
> > > > (Zhao et al., 2018) Yue Zhao, Meng Li, Liangzhen Lai, Naveen Suda, Damon Civin, and Vikas Chandra. Federated learning with non-IID data. arXiv preprint arXiv: 1806.00582, 2018.
> > > > (Hochreiter & Schmidhuber, 1997) Sepp Hochreiter and Jurgen Schmidhuber. Flat minima. Neural Computation, 9(1), 1997.
> > > > (Keskar et al., 2017) Nitish Shirish Keskar, Dheevatsa Mudigere, Jorge Nocedal, Mikhail Smelyanskiy, and Ping Tak Peter Tang. On large-batch training for deep learning: Generalization gap and sharp minima. In ICLR, 2017.

---

> > > > > ### Author Response · Authors · 2019-11-15
> > > > > **Author response to reviewer #3 (5/6)**
> > > > >
> > > > >
> > > > > 8. Why for different experiments different baseline models are used? (NetA, NetB, NetC)
> > > > > =====(Answer)=====
> > > > > Basically, we considered NetA-Baseline as our baseline network architecture; it was used in the investigation of the effects of (i) optimizers, (ii) weight decay, (iii) Batch Normalization, (iv) data augmentation, and (v) Dropout.
> > > > > In order to study the effects of network depth, we also used its two variants, i.e., NetA-Deeper and NetA-Deepest.
> > > > > Also, in order to study the effects of network width in relation of convolutional layers, we also used its other two variants, i.e., NetA-Narrower and NetA-Narrowest. While the first convolutional layer of NetA-Baseline has 64 output channels, those of NetA-Narrower and NetA-Narrowest have 16 and 32 output channels, respectively.
> > > > > In addition, regarding network width in relation of fully-connected layers, we also wanted to investigate the effects of the global average pooling. Therefore, we used two baseline networks (i.e., NetB-Baseline and NetC-Baseline), of which the number of fully-connected layers is 3 and 1; they also use the global average pooling after the last convolutional layer. One might regard the NetB-Baseline and the NetC-Baseline as a VGG-type and a ResNet-type fully-connected layers.
> > > > > We also then constructed its max pooling variants, i.e., NetB-Wider and NetB-Widest; and NetC-Wider and NetC-Widest.
> > > > > We additionally remark that the NetC-Baseline network can be regarded as a shallow ResNet-type network, and it was compared with ResNet-14 and ResNet-20 in our study (see Table 8 in the appendix of the revised version of the paper).
> > > > >
> > > > >
> > > > > - Appendix B, first equation on page 13. (d_q)^t -> (d_q)^t_k; The size of gradient \nabla_w [E ...] is different from the size of (d_q)_k. They cannot be added together.
> > > > > =====(Answer)===== We appreciate the thankful comment; we corrected this in the revised version of the paper. Please see Appendix B.
> > > > >
> > > > > - page 7, last sentence of the first paragraph: what is the accuracy achieved with Batch Renormalization?
> > > > > =====(Answer)===== We apologize for your confusion. In the revised version of the paper, it can be found at Table 3.
> > > > >
> > > > >   Why the reason for accuracy gap is “significant parameter divergence”? on fig. 3 “parameter divergence” is smaller than for the baseline.
> > > > > =====(Answer)===== We clarified this in the revised version of the paper as follows:
> > > > > “Batch Normalization yields not only big parameter divergence (especially before the first learning rate drop) but also the steep fall phenomenon; the corresponding test accuracy was seen to be very low (see Table 3).”
> > > > >
> > > > > - Why the name of the section on page 7 is “excessively high training loss of local updates” if later it is stated that it is actually smaller than for the IID case?
> > > > > =====(Answer)===== This is because the comparison target of “excessively high” here is the baseline cases under the non-IID data setting. Also, as we remarked in the paper, please additionally note that the training loss being high is much more critical under non-IID data setting than under IID cases; this is because local updates are extremely easy to be overfitted to each training dataset under non-IID data environments.
> > > > >
> > > > > - section 3: “A pleasant level of parameter divergence can help to improve generalization” -> where was it shown?
> > > > > =====(Answer)===== We appreciate the valuable comment; In the revision of the paper, we corrected/clarified the sentence as follows:
> > > > > “A pleasant level of parameter divergence could rather imply exploiting rich decentralized data”
> > > > >
> > > > > - section 4.2, paragraph 2: what is meant by “hyperparametric methods”?
> > > > > =====(Answer)===== We apologize for your confusion; in our paper, “hyperparametric methods” is used interchangeably with “hyperparameter optimization methods”. We specified this at Footnote 2 in the revised version of the paper.
> > > > >
> > > > > - section 4.2, paragraph 3: “quantitative increase in a layer level” -> not clear what does it mean.
> > > > > =====(Answer)===== Our parameter divergence metrics make normalized (qualitative) measures possible since they used cosine distance instead of Euclidean distance. Please refer to also the third paragraph in Section 3 of the revised version of the paper.

---

> > > > > > ### Author Response · Authors · 2019-11-15
> > > > > > **Author response to reviewer #3 (6/6)**
> > > > > >
> > > > > >
> > > > > > - page 4, effect of optimizers: what do you refer to as “all model parameters”?
> > > > > > =====(Answer)===== We apologize for your confusion; we clarified this in the revised version of the paper as follows:
> > > > > > “Effects of optimizers. Unlike non-adaptive optimizers such as pure SGD and momentum SGD (Polyak, 1964; Nesterov, 1983), Adam (Kingma & Ba, 2015) could give poor performance from non-IID data if the parameter averaging is performed only for weights and biases, compared to all the model variables (including the first and second moment) being averaged.”
> > > > > >
> > > > > > - why Dropout yields bigger parameter divergence if on Fig 2, right it actually helps?
> > > > > > =====(Answer)===== At the initial steps of this study, we had expected that the dropped nodes (or neurons), randomly selected, are different across learners, and its impact on 􏰑the parameter divergence would be much stronger than under the IID setting; the experimental results were also shown that the Dropout yield bigger parameter divergence. However, it was observed that the generalization effect of the Dropout could be still valid in test accuracy in some cases. Regarding this, we expect that the positive effects of the Dropout become weaker as difficulty level of learning tasks goes to be higher (e.g., CIFAR-100).
> > > > > >
> > > > > > - Last line of the page 5. Where was this observed?
> > > > > > =====(Answer)===== We apologize for your confusion. In the revised version of the paper, it can be found at Table 13 of the appendix; we specified this.

---

### Official Review · AnonReviewer2 · 2019-10-24
**Official Blind Review #2**

**Rating:** 1

**Review:**

Summary:
The paper presents an empirical study of causes of parameter divergence in federated learning. Federated learning is the setting where parameter updates (e.g. gradients) are computed separately on possibly non-IID subsamples of the data and then aggregated by averaging. The paper examines the effects of choice of optimizer, network width and depth, batch normalization, weight decay, and data augmentation on the amount of parameter divergence. Divergence is defined as the average cosine distance between pairs of locally-updated weights, or between locally updated weights and weights trained with IID data. The paper generally concludes that regularization methods like BN and weight decay have an adverse effect in the federated setting, the deepening the network has an adverse effect while widening it might be beneficial, and that adaptive optimizers like Adam can perform poorly if their internal statistics are not aggregated.

I recommend that the paper be rejected. The main shortcoming of the paper is the lack of rigororous statistical analysis to support its conclusions. The paper contains a lot of raw data, but the discussion mainly highlights trends that the authors seem to have observed in the results, without quantifying the relative sizes of effects, how consistent they are across experimental conditions, etc. The writing is also quite unclear, to the point that I often didn't understand exactly what argument was being made.

Details / Questions:
The main problem is the lack of quantitative analysis of the trends the paper identifies. For example, regarding "Effects of Batch Normalization", there seem to be two claims made:
1. Batch normalization makes things worse (somehow) in the federated setting
2. Batch re-normalization still makes things worse, but not as much
How are these effects quantified? How large are they? Do they hold across all datasets, architectures, and optimizers considered? Ideally there would be a table summarizing each experimental manipulation, its effect on performance, whether that effect is significant, etc. Of course this requires some care because the paper is doing an exploratory analysis and there are many hypotheses to test; a good reference is [1].

The paper also relies heavily on parameter divergence as a measure of performance in federated learning, but I see no evidence presented that parameter divergence is predictive of test accuracy (which is presumably what we actually care about). Intuitively I can see how it might be related, but since divergence is basically being used as a proxy for accuracy, it is vital to show convincingly that the two are related. What do we gain by analyzing parameter divergence rather than simply comparing test accuracy?

Regarding the "steep fall phenomenon": The paper seems to present this as an indicator that a manipulation performs poorly in the federated setting. But, isn't it a good thing if parameter divergence goes down? Why does specifically a sudden, sharp decrease in divergence indicate a problem?

Finally, some improvements might be made to the experiment setup. For one, the case of completely-disjoint label sets in different local learners seems extreme to me. Wouldn't at least partial overlap be more common in practice? (This is not my area so I don't know). Experimenting with different degrees of overlap would be useful. As for network architectures, it would be valuable to look at a greater variety of standard architecture styles (e.g. ResNet, Inception, etc). I realize there are some experiments with ResNet, but the focus is mainly on the single-path VGG-like architecture. I do realize this is a lot of experiments to do.

Minor points:
* In the setting described as "IID" in Table 1 is not, the subsampled for each learner are not IID subsamples of the full dataset because they are class-balanced (if I'm understanding correctly)

References:
[1] Demšar, J. (2006). Statistical comparisons of classifiers over multiple data sets. Journal of Machine Learning Research, 7(Jan), 1-30.

**Experience Assessment:**

I do not know much about this area.

**Review Assessment: Checking Correctness Of Derivations And Theory:**

N/A

**Review Assessment: Checking Correctness Of Experiments:**

I assessed the sensibility of the experiments.

**Review Assessment: Thoroughness In Paper Reading:**

I read the paper at least twice and used my best judgement in assessing the paper.

---

> ### Author Response · Authors · 2019-11-15
> **Author response to reviewer #2 (1/3)**
>
>
> We first appreciate the valuable comments. We carefully looked through all the comments; the following describes our answers.
>
> 1. Regarding the lack of quantitative analysis of the trends the paper identifies
> =====(Answer)=====
> We appreciate the valuable comments. As you point out, we admit that the paper lacks a quantitive analysis of the findings.
> However, please remind that even under “vanilla” training, it is not easy to generally quantify the gains of the considered hyperparameter optimization methods since they highly depend on the training dataset or the remaining training strategies. Therefore, we were afraid to conclude the general quantification of the effects of the methods.
> Instead, by also providing the results under “vanilla” training and the federated learning with IID data, we tried to emphasize the negative effects of the hyperparametric methods; we think our results show the severity of performance degradation of each method, even indirectly.
>
>
> 2. Regarding the relationship between test accuracy and parameter divergence
> =====(Answer)=====
> In many previous literatures, e.g., (Zhao et al., 2018), parameter divergence is regarded as a direct response to learners’ local data being non-IID sampled from the population distribution. In relation, it was reported that as the probabilistic distance (e.g., earth mover’s distance) of learners’ local data becomes farther away from the population distribution, bigger parameter divergence might appear; this is correlated with the degradation of performance such as test accuracy (please refer to Section 3.2 of (Zhao et al., 2018)).
> Also, we added our analysis of the relationship among the three factors (i.e., probabilistic distance, parameter divergence, and performance) in the rebuttal period; the relevant description can be found in Section 3 of the revised version of this paper.
>
> The reason of probing parameter divergence being important is that the federated learning are performed based on iterative parameter averaging. That is, investigating how local updates are diverged can give a clue whether the subsequent parameter averaging yields positive returns; the proposed divergence metrics provide two ways for it.
>
> (Zhao et al., 2018) Yue Zhao, Meng Li, Liangzhen Lai, Naveen Suda, Damon Civin, and Vikas Chandra. Federated learning with non-IID data. arXiv preprint arXiv: 1806.00582, 2018.

---

> > ### Author Response · Authors · 2019-11-15
> > **Author response to reviewer #2 (2/3)**
> >
> >
> > 3. Regarding steep fall phenomenon
> > =====(Answer)=====
> > We apologize for your confusion; the philosophy behind the steep fall phenomenon is as follows:
> > In many previous literatures, e.g., (Zhao et al., 2018), inordinate magnitude of parameter divergence is regarded as a direct response to learners’ local data being non-IID sampled from the population distribution; thus they explained that the consequent parameter averaging with the highly diverged local updates could lead to bad solutions far from the global optimum. Likewise, in our experiments, for many of the failure cases under the non-IID data setting, we observed that the inordinate magnitude of parameter divergence could become one of the internal causes of the diminishing returns.
> > However, under the non-IID data setting, some of the failure cases have been observed where the test accuracy is still low but the parameter divergence values decrease (rapidly) over rounds; as the round goes, even the values were sometimes seen to be lower than those of the comparison targets.
> > For the failure cases, we concluded that these (unexpected abnormal) sudden drop of parameter divergence values indicate going into poor local minima (or saddles); this can be supported by the behaviors that test accuracy increases plausibly at very early rounds, but the growth rate quickly stagnates and eventually becomes much lower than the comparison targets.
> > In relation, we provided Figure 5 (in the revised version of this paper) as the evidence of the steep fall phenomenon; as depicted in the figure, the loss landscapes of the failure cases (i.e., “Adam-WB” and “w/ BN” under the non-IID setting) show sharper minima and the minimal value in the bowl is relatively greater. Here “sharp” minima is broadly known to lead to poorer generalization ability (Hochreiter & Schmidhuber, 1997; Keskar et al., 2017); it is observed from the figure that going into a sharp minima happens even in early rounds (e.g., 25th).
> > It is expected that the discovery of these steep fall phenomena provides a new insight into the relationship between test accuracy and parameter divergence; we believe that the steep fall phenomenon should be considered as the cause of diminishing returns of the federated learning with non-IID data, along with the inordinate magnitude of parameter divergence.
> >
> > (Zhao et al., 2018) Yue Zhao, Meng Li, Liangzhen Lai, Naveen Suda, Damon Civin, and Vikas Chandra. Federated learning with non-IID data. arXiv preprint arXiv: 1806.00582, 2018.
> > (Hochreiter & Schmidhuber, 1997) Sepp Hochreiter and Jurgen Schmidhuber. Flat minima. Neural Computation, 9(1), 1997.
> > (Keskar et al., 2017) Nitish Shirish Keskar, Dheevatsa Mudigere, Jorge Nocedal, Mikhail Smelyanskiy, and Ping Tak Peter Tang. On large-batch training for deep learning: Generalization gap and sharp minima. In ICLR, 2017.

---

> > > ### Author Response · Authors · 2019-11-15
> > > **Author response to reviewer #2 (3/3)**
> > >
> > >
> > > 5. In the setting described as "IID" in Table 1 is not, the subsampled for each learner are not IID subsamples of the full dataset because they are class-balanced (if I'm understanding correctly)
> > > =====(Answer)=====
> > > As you point out, in our “IID” data setting, we cannot say that each learner’s data examples are practically IID sampled.
> > > In order to each learner’s data examples being practically IID sampled, we should have conducted the followings: (i) shuffling the full dataset, and (ii) partitioning the data examples into learners in that order.
> > > However, we did (i) sorting the full dataset by class, and (ii) partitioning the data examples into learners to be class-balanced.
> > > Nevertheless, the CIFAR-10 dataset consists of 50000 training data examples of 5000 images each for 10 classes; thus, statistically we believe that our “IID” data setting can be regarded as one of the ideal IID settings.
> > > Note that, for the SVHN dataset (consisting of 73257 training data examples), we reconstructed the full dataset to have 50000 training data examples of 5000 images each for 10 classes.

---

### Author Response · Authors · 2019-09-30
**Some typos**

We found some typos in the paper:

In “Inordinate magnitude of parameter divergence” of Section 4.2,

(1) Regarding the second sentence from the behind of the second paragraph, we correct this sentence to “Since the NetA-Deeper and NetA-Deepest have twice and three times as many model parameters as NetA-Baseline, it can be expected enough that the deeper models yield bigger parameter divergence in the whole model; but our results also show its qualitative increase in a layer level.”

(2) Regarding the last sentence of the third paragraph, we correct this sentence to “We additionally observe for the non-IID cases that even with the weight decay factor of 0.0005, the parameter divergence values are similar to those with the smaller factors at very early rounds in which the norms of the weights are relatively very small.”

(3) Regarding the first sentence in the fourth paragraph, we correct this sentence to “In addition, it is observed from the right plot of the figure that Dropout (Hinton et al., 2012; Srivatava et al. 2014) also yields bigger parameter divergence under the non-IID data setting.”

(4) Regarding the last sentence in the fourth paragraph, we correct this sentence to “The corresponding test accuracy was seen to be a diminishing return with the momentum SGD optimizer (i.e., using Dropout we can achieve +2.85% under IID, but only +1.69% is obtained under non-IID(2), compared to when it is not applied); however, it was observed that the generalization effect of the Dropout is still valid in test accuracy for the pure SGD and the Adam (refer to also Table 10 in the appendix).”

In Appendix B,

(5) Regarding the last sentence, we correct this sentence to “Then, similar with Equation (1), we can have (the equation).”

We apologize for the inconvenience.

---

### Author Response · Authors · 2019-11-15
**Summary of our distinct contributions (1/2)**

We believe that focusing on federated learning with non-IID data, our work provides the meaningful exploratory analysis breaking the existing common wisdom about the considered hyperparameter optimization methods. In relation, here we intend to emphasize our contributions.

Our distinct contributions can be highlighted as follows:

**Regarding Section 3:
In many previous literatures, e.g., (Zhao et al., 2018), parameter divergence is regarded as a direct response to learners’ local data being non-IID sampled from the population distribution. In relation, it was reported that as the probabilistic distance (e.g., earth mover’s distance) of learners’ local data becomes farther away from the population distribution, bigger parameter divergence might appear; this is correlated with the degradation of performance such as test accuracy (please refer to Section 3.2 of (Zhao et al., 2018)). Also, we added our analysis of the relationship among the three factors (i.e., probabilistic distance, parameter divergence, and performance) in the rebuttal period; the relevant description can be found in Section 3 of the revised version of this paper.

Regarding the parameter divergence, our distinct contribution can be summarized in two-fold:
First, for the first time we identified the mechanism by which data non-IIDness affects the parameter divergence: “if data distributions in each local dataset are highly skewed and heterogeneous over classes, subsets of neurons, which have especially big magnitudes of the gradients in back propagation, become significantly different across learners; this leads to inordinate parameter divergence between them”. It has been analyzed in both empirical and theoretical way.
Second, many of the related literatures usually handle the parameter difference of each learner’s local model parameters from one computed with the population distribution (this philosophy is connected to the definition of PD-VL); meanwhile, in our study we also considered the parameter diversity between the local updates as well (this is connected to the definition of PD-Ls). The reason of probing parameter divergence being important is that the federated learning are performed based on iterative parameter averaging. That is, investigating how local updates are diverged can give a clue whether the subsequent parameter averaging yields positive returns; the proposed divergence metrics provide two ways for it.

**Regarding Section 4.1:
In this study, we focused on the well-known hyperparameter optimization strategies (i.e., hyperparametric strategies) to improve learning performance: (i) using momentum SGD or Adam than pure SGD, (ii) network deepening/widening (until a proper level), (iii) Batch Normalization, (iv) weight decay, (v) data augmentation, and (vi) Dropout. Their positive effects have been reported in a variety of literatures; practically, they are being broadly used in deep net training. Also in our experiments, the hyperparametric methods yielded better outcome under vanilla training (i.e., non-distributed training) and under the considered federated learning algorithm with the IID decentralized data setting.
However, under the non-IID data setting, we newly identified that the hyperparametric methods could rather give negative/diminishing effects on performance of the federated learning algorithm; we believe that these findings can be highly impactful to the upcoming works or industrial implementations.

---

> ### Author Response · Authors · 2019-11-15
> **Summary of our distinct contributions (2/2)**
>
>
> **Regarding Section 4.2:
> In many previous literatures, e.g., (Zhao et al., 2018), inordinate magnitude of parameter divergence is regarded as a direct response to learners’ local data being non-IID sampled from the population distribution; thus they explained that the consequent parameter averaging with the highly diverged local updates could lead to bad solutions far from the global optimum. Likewise, in our experiments, for many of the failure cases under the non-IID data setting, we observed that the inordinate magnitude of parameter divergence could become one of the internal causes of the diminishing returns.
> However, under the non-IID data setting, some of the failure cases have been observed where the test accuracy is still low but the parameter divergence values decrease (rapidly) over rounds; as the round goes, even the values were sometimes seen to be lower than those of the comparison targets. For the failure cases, we concluded that these (unexpected abnormal) sudden drop of parameter divergence values indicate going into poor local minima (or saddles); this can be supported by the behaviors that test accuracy increases plausibly at very early rounds, but the growth rate quickly stagnates and eventually becomes much lower than the comparison targets.
> In relation, we provided Figure 5 (in the revised version of this paper) as the evidence of the steep fall phenomenon; as depicted in the figure, the loss landscapes of the failure cases (i.e., “Adam-WB” and “w/ BN” under the non-IID setting) show sharper minima and the minimal value in the bowl is relatively greater. Here “sharp” minima is broadly known to lead to poorer generalization ability (Hochreiter & Schmidhuber, 1997; Keskar et al., 2017); it is observed from the figure that going into a sharp minima happens even in early rounds (e.g., 25th).
> It is expected that the discovery of these steep fall phenomena provides a new insight into the relationship between test accuracy and parameter divergence; we believe that the steep fall phenomenon should be considered as the cause of diminishing returns of the federated learning with non-IID data, along with the inordinate magnitude of parameter divergence.
>
> (Zhao et al., 2018) Yue Zhao, Meng Li, Liangzhen Lai, Naveen Suda, Damon Civin, and Vikas Chandra. Federated learning with non-IID data. arXiv preprint arXiv: 1806.00582, 2018.
> (Hochreiter & Schmidhuber, 1997) Sepp Hochreiter and Jurgen Schmidhuber. Flat minima. Neural Computation, 9(1), 1997.
> (Keskar et al., 2017) Nitish Shirish Keskar, Dheevatsa Mudigere, Jorge Nocedal, Mikhail Smelyanskiy, and Ping Tak Peter Tang. On large-batch training for deep learning: Generalization gap and sharp minima. In ICLR, 2017.

---

### Author Response · Authors · 2019-11-15
**We posted the revised version of the paper**

Thanks to the valuable comments of reviewers, we were able to improve our paper better.

In the revision, we aimed to improve the clarity by addressing the concerns/questions of reviewers.

In the process we strengthened the description of the reason why the proposed parameter divergence metrics are important, and we added the new content to establish the relationship among probabilistic distance, parameter divergence, and performance. We also provide the additional experiment results about (i) ResNet with Nesterov momentum SGD, (ii) Polyak momentum SGD, (iii) unbalanced non-IID data settings, and (iv) FedProx (Li et al., 2019).

(Li et al., 2019) Tian Li, Anit Kumar Sahu, Manzil Zaheer, Maziar Sanjabi, Ameet Talwalkar, and Virginia Smith. Federated optimization for heterogeneous networks. In ICML Workshop, 2019.

How and where the reviewers’ comments have been addressed in the revised version is described in the responses to each comment.

Best regards,

Authors

---

### Decision · Program_Chairs · 2019-12-19

**Decision:**

Reject

**Comment:**

This paper studies the problem of federated learning for non-i.i.d. data, and looks at the hyperparameter optimization in this setting. As the reviewers have noted, this is a purely empirical paper. There are certain aspects of the experiments that need further discussion, especially the learning rate selection for different architectures. That said, the submission may not be ready for publication at its current stage.